# Evaluation and optimization of ICOS atmosphere station data as part of the labeling process

Camille Yver-Kwok[1], Carole Philippon[1], Peter Bergamaschi[2], Tobias Biermann[3], Francescopiero Calzolari[4], Huilin Chen[5], Sebastien Conil[6], Paolo Cristofanelli[4], Marc Delmotte[1], Juha Hatakka[7], Michal Heliasz[3], Ove Hermansen[8], Kateřina Komínková[9], Dagmar Kubistin[10], Nicolas Kumps[11], Olivier Laurent[1], Tuomas Laurila[7], Irene Lehner[3], Janne Levula[12], Matthias Lindauer[10], Morgan Lopez[1], Ivan Mammarella[12], Giovanni Manca[2], Per Marklund[13], Jean-Marc Metzger[14], Meelis Mölder[15], Stephen M. Platt[9], Michel Ramonet[1], Leonard Rivier[1], Bert Scheeren[5], Mahesh Kumar Sha[11], Paul Smith[13], Martin Steinbacher[16], Gabriela Vítková[9], and Simon Wyss[16]

[1]Laboratoire des Sciences du Climat et de l'Environnement (LSCE-IPSL), CEA-CNRS-UVSQ, Université Paris-Saclay, F-91191 Gif-sur-Yvette, France
[2]European Commission Joint Research Centre (JRC), Via E. Fermi 2749, 21027 Ispra, Italy
[3]Centre for Environmental and Climate Research, Lund University, Sölvegatan 37, SE-223 62, Lund, Sweden
[4]National Research Concil of Italy, Institute of Atmospheric Sciences and Climate, Via Gobett 101, 40129 Bologna, Italy
[5]Centre for Isotope Research (CIO), Energy and Sustainability Research Institute Groningen (ESRIG), University of Groningen, Groningen, the Netherlands
[6]DRD/OPE, Andra, Bure, 55290, France
[7]Finnish Meteorological Institute, P.O.Box 503, FI-00101 Helsinki, Finland
[8]NILU, Norwegian Institute for Air Research, Kjeller, Norway
[9]Global Change Research Institute of the Czech Academy of Sciences, Brno, Czech Republic.
[10]Meteorologisches Observatorium Hohenpeissenberg, Deutscher Wetterdienst (DWD), 82383 Hohenpeissenberg, Germany
[11]Royal Belgian Institute for Space Aeronomy (BIRA-IASB), Brussels, Belgium
[12]Institute for Atmospheric and Earth System Research/Physics, Faculty of Sciences, University of Helsinki, Helsinki, Finland
[13]Department of Forest Ecology and Management, Swedish University of Agricultural Sciences, Umea, Sweden
[14]Unité Mixte de Service (UMS 3365), Observatoire des Sciences de l'Univers à La Réunion (OSU-R), Université de la Réunion, Saint-Denis de la Réunion (FR)
[15]Department of Physical Geography and Ecosystem Science, Lund University,Sölvegatan 12 SE-223 62, Lund, Sweden
[16]Empa, Laboratory for Air Pollution/Environmental Technology,Dübendorf, Switzerland

**Correspondence:** Camille Yver-Kwok (camille.yver@lsce.ipsl.fr)

**Abstract.** The Integrated Carbon Observation System (ICOS) is a pan-European research infrastructure which provides harmonized and high precision scientific data on the carbon cycle and the greenhouse gas budget. All stations have to undergo a rigorous assessment before being labeled, i.e. receiving approval to join the network. In this paper, we present the labeling process for the ICOS atmosphere network through the 23 stations that have been labeled between November 2017 and November 2019. We describe the labeling steps as well as the quality controls used to verify that the ICOS data ($CO_2$, $CH_4$, CO and meteorological measurements) attain the expected quality level defined within ICOS. To ensure the quality of the greenhouse gas data, three to four calibration gases and two target gases are measured, one target two to three times a day, the other gases twice a month. The data are verified on a weekly basis and tests on the station sampling lines are performed twice a year.

From these high-quality data, we conclude that regular calibrations of the $CO_2$, $CH_4$ and CO analyzers used here (twice a month) are important in particular for carbon monoxide (CO) due to the analyzer's variability and that reducing the number of calibration injections (from four to three) in a calibration sequence is possible, saving gas and extending the calibration gas lifespan. We also show that currently, the on-site water vapor correction test does not deliver quantitative results possibly due to environmental factors. Thus the use of a drying system is strongly recommended. Finally, the mandatory regular intake line tests are shown to be useful to detect artifacts and leaks as shown here via three different examples at the stations.

## 1 Introduction

Precise greenhouse gas monitoring began in 1957 at the South Pole and in 1958 at the Mauna Loa observatory (Keeling, 1960; Brown and Keeling, 1965; Pales and Keeling, 1965). Over these 60 years of data, $CO_2$ levels have risen by about 100 ppm in the atmosphere. $CO_2$ and other greenhouses gases are a major source of climate forcing (IPCC et al., 2014) and following Mauna Loa measurements, several monitoring networks (Prinn et al., 2018; Andrews et al., 2014; Fang et al., 2014; Ramonet et al., 2010) and coordinating programs (WMO, 2014) have been developed over time to monitor the increasing mixing ratios in different parts of the world, and quantify the relative roles of the biospheric, oceanic fluxes and anthropogenic emissions. Initially, the goal was to measure greenhouse gases at background stations to get data representative of large scales. Later, more and more regional stations and networks have been established in order to get more information on regional to local fluxes. Indeed, this is especially relevant in the context of monitoring and verifying the international climate agreements (Bergamaschi et al., 2018).

The Integrated Carbon Observation System (ICOS) is a pan-European research infrastructure (https://www.icos-ri.eu) which provides highly compatible, harmonized and high precision scientific data on the carbon cycle and greenhouse gas budget. It consists of three monitoring networks: atmospheric observations, flux measurements within and above ecosystems and measurements of $CO_2$ partial pressure in seawater. Its implementation included a preparatory phase (2008–2013, EU FP7 project reference 211574) and a demonstration experiment until the end of 2015 when ICOS officially started as a legal entity. ICOS was first designed to serve as a backbone network to monitor fluxes away from main anthropogenic sources. There, the concentration gradients between European sites is typically of only a few ppm on seasonal time scales. These small atmospheric signals combined with atmospheric transport models are used to deduce surface fluxes. For these atmospheric inversions a high precision and integrated network is mandatory. As a precise example, Ramonet et al. (2020) show that a strong drought in Europe like the one seen in summer 2018 produces an atmospheric signal of only 1 to 2ppm.

During the preparatory phase and the demonstration experiment, standard operating procedures for testing the instruments and measuring air in the most precise and unbiased way were defined. Data management plans were created and required IT infrastructure such as databases and the quality control software tools were developed. In addition to the monitoring networks and the Head Office which is the organizational hub of the entire ICOS research infrastructure, central facilities have been built to support the production of high-quality data. This ensures traceability, quality assurance/quality control (QA/QC), instrument testing, data handling and network support, with the aim of standardizing operations and measurement protocols. The central

facilities are grouped as follows: the Flask and Calibration Laboratory (CAL-FCL, Jena, Germany) for greenhouse gas flask and cylinder calibration (linking the ICOS data to the WMO calibration scales), the Central Radiocarbon Laboratory (CAL-CRL, Heidelberg, Germany) for radiocarbon analysis, next to three thematic centers for atmosphere, ecosystem and oceans.

The thematic centers are responsible for data processing, instrument testing and developing protocols in collaboration with station Principal Investigators (PI's). Regular Monitoring Station Assembly meetings (MSA) facilitate discussion of technical and scientific matters.

The Atmosphere Thematic Center (ATC, https://icos-atc.lsce.ipsl.fr/) is divided into three components; the metrology laboratory (MLab) responsible for instrument evaluation, protocol definition and PI support, the data unit responsible for data processing, code development and graphical tools for PI's, both located in Gif-sur-Yvette, France and finally the MobileLab in Helsinki, Finland, tasked with the audit of the stations during and after the labeling process.

One very important task for the ATC is ensuring that the stations reach the quality objectives defined within ICOS, based on the compatibility goals of the WMO (WMO, 2018) and detailed in the ICOS Atmosphere Station specifications (ERIC, 2020). To do so, a so-called "labeling process" has been developed to firstly assess the relevance of a new measurement site, as well as the adequacy of the human and logistical resources available with the ICOS requirements. Afterwards, an evaluation of the first months of measurement is carried out, verifying compliance with the ICOS protocols. The Carbon Portal (https://www.icos-cp.eu/, Lund, Sweden), which is responsible for storage and dissemination of data and elaborated products (such as inversion results or emission maps) is associated in the labeling process via PID/DOI attribution for the data and the provision of a web interface to gather important information needed for the labeling. The labeling process is very useful for new stations coming into the network to ensure proper setting, good measurement practice and in the end be able to reach the precision and stability requirement of ICOS. For the end-user, the labeling process guarantees high quality observations with full metadata description and traceable data processing.

In this paper, we present the labeling process for the ICOS atmosphere network and illustrate it through the twenty-three stations that have been labeled between November 2017 (first stations labeled) and November 2019. First, we describe the protocol that a station must follow to be labeled. Then, we detail the different metrics and elements that are analyzed during the labeling process to validate the quality level. Afterwards, we present the 23 labeled atmosphere stations and in a third part, we discuss results and findings from these stations as seen during the labeling process.

## 2   Protocol and metrics of the labeling process

To be labeled, an ICOS atmosphere station has to follow the guidelines and requirements defined in the ICOS Atmosphere Station specifications (ERIC (2020), hereafter referred as "the AS specifications") and the labeling document (ATC-GN-LA-PR-1.0_Step2info.pdf, available on the ATC website, section Documents, Public documents, Labelling or at https://box.lsce.ipsl.fr/index.php/s/uvnKhrEinB2Adw9/download?path=%2FLabelling&files=ATC-GN-LA-PR-1.0_Step2info.pdf). The AS specifications are discussed and updated if necessary every six months at the Monitoring Station Assembly meetings that include the PI's of all the ICOS stations and representatives of the central facilities. The goal of these specifications is to allow each site to

reach the performances required by the ICOS atmosphere data quality objectives, which are principally adhering to the WMO guidelines (WMO, 2018) for greenhouse gas observations but are elaborated on more in the AS specifications and presented in section 2.4 below.

The labeling process of atmosphere stations has been defined as a three-step process: Step 1: Evaluation of the station location and infrastructure, Step 2: Station performances, Step 3: Official and formal ICOS data labeling by the ICOS General Assembly (composed of representatives of the Member and Observer countries of ICOS and meeting twice a year).

To pass Step 1, stations must submit information about the infrastructure of their site, its location and its proximity to anthropogenic sources like cities or main roads. This is done through the Carbon Portal interface (https://meta.icos-cp.eu/labeling). The ATC uses these data to compile a report and issue recommendations to the ICOS Head Office who will then approve or reject the application. Usually, if there are some problematic points, the ATC first contacts the PI to see if improvements can be done to meet the requirements or ask for additional documents. ICOS atmosphere is mainly focused on tall tower sites, measuring regional signals but accepts a limited number of a high altitude and coastal sites (ERIC, 2020).

Once Step 1 is approved, the station can be built, equipped and set up to fulfill the AS specifications. Once the near-real time data flow to the ATC database is established (Hazan et al., 2016), stations can apply for Step 2. The time lap between Step 1 and Step 2 can vary greatly depending on the site. Indeed, in case of already existing stations, they are entering ICOS with already running instruments and historical datasets and need only small changes such as getting the calibration cylinders from the CAL-FCL and modifying some procedures to have their data processed into the database before beginning Step 2. Others will have the whole construction of the tall tower, shelter and installation of lines to achieve first.

During Step 2, a phase of measurement optimization begins: the initial test period. This is done in close collaboration between the station PI and the ATC through routine sessions of data evaluation (usually every month). This period typically lasts 4 to 6 months to gather data to evaluate their quality. The period may be prolonged if needed. If data meeting the AS specifications are available prior to the Step 2 application, the initial test period can be shorter.

During the initial test period, the requirements detailed hereafter are asked from the station PI in order to be able to analyze all the data in a uniform way for all sites.

## 2.1 General requirements

ICOS atmosphere network aims to provide high precision measurements of greenhouse gases, and in priority $CO_2$ and $CH_4$ which represent the main anthropogenic GHG. In-situ measurements of $N_2O$, the third most important contributor to the additional radiative forcing, was not required in the initial phase of ICOS due to the difficulty to find at that time, reliable instruments able to provide the expected precision (Lebegue et al., 2016). This gas is expected to become mandatory in the near future of ICOS. Flask sampling is required at Class 1 stations for quality control of in-situ measurements, and to provide additional trace gases measurements like $N_2O$, $H_2$, $CO_2$ isotopes (Levin et al., 2020). Other parameters are required in order to support the interpretation of the GHG variabilities, like CO as a tracer of combustions, and meteorological parameters to characterize the local winds, vertical stability along tall towers and weather conditions (pressure, temperature, relative humidity). The eddy covariance fluxes have been selected as well with the idea to characterize the local surface fluxes either

from biogenic and/or anthropogenic activities and to monitor possible long term changes around the ICOS sites. So far this parameter is not required for the labeling process due to logistical difficulties to install such measurements at several atmosphere sites.

At the beginning of the initial test period, a station must provide at minimum continuous in-situ greenhouse gas data to the database on a daily basis, and by the end, meteorological parameters (wind speed, wind direction, atmospheric temperature, relative humidity and pressure) and additional diagnostic data (room temperature, instrument and flushing pump flow rates). Table 1 shows the list of all mandatory parameters that should be provided by the stations depending on the station class. Furthermore, it also provides a list of recommended parameters. A Class 1 station will provide more parameters than Class 2, but both stations must meet the same level of data quality. Presently, the MSA has decided that labeling should not be contingent upon two Class 1 parameters: the boundary layer height and the measure of greenhouse gases and $\delta^{14}CO_2$ values from flask sampling. Indeed, for these two parameters, the technologies, hardware and software are still in development or in need of improvement (Feist et al., 2015; Levin et al., 2020; Poltera et al., 2017). As soon as the MSA decides to approve a technology, it should however be added to the station as soon as possible. Indeed, flask sampling is an additional quality control tool as well as a way to sample species that cannot be yet measured continuously while boundary layer heights would help improve models. For all the stations presented here, we focused on $CO_2$ and $CH_4$ continuous measurements for all sites and on CO measurements for Class 1 sites. The other species measured by some instruments such as $N_2O$ were not assessed as not mandatory.

The instruments providing the data must be ICOS compliant as defined in the AS specifications. The list of accepted analyzers is regularly updated to keep up with new technologies that are continuously tested at the ATC. In the case of the GHG analyzers, all instruments operated in the network are tested at the ICOS ATC MLab following the procedure described in Yver Kwok et al. (2015). Their intrinsic performances are evaluated as well as their sensitivities to atmospheric pressure, instrument inlet pressure, ambient temperature, other species and water vapor. In the case of water vapor, a specific correction is determined for each instrument. A test report is produced systematically to provide the specific analyzer status in regards to its compliance with the specifications. It is important to characterize the instrument performances under well defined and controlled conditions at the ATC MLab since they will be used as a reference for the evaluation of field performances. The initial test of the analyzers also allows to verify if the performances of the instruments are consistent with the specifications provided by the manufacturer. Over the past years, few instruments were sent back to the manufacturer due to poor performance. Others parts such as pressure regulators, gas distribution systems, type of cylinders are also defined in the AS specifications.

## 2.2 Greenhouse gas calibration requirement

For consistency and efficiency purpose over the network, a common calibration strategy has to be followed. During the initial test period, the general philosophy is to carry out frequent calibrations and quality control measurements, with the aim of determining their optimal frequencies and durations. Presently, the calibration strategy for the initial period is as follow: 3 to 4 cylinders (filled with natural dry air, for which values have been assigned at the CAL-FCL and are traceable to the WMO scales, https://www.icos-cal.eu/static/images/docs/ICOS-FCL_QC-Report_2017_v1.3.pdf) are each measured four times for

30 minutes one after the other every 15 days leading to a total six to eight hours of calibration measure. Depending on the stability of one calibration to the other, ATC will recommend if the frequency can be reduced but in any case at least one calibration sequence per month is required. Cylinder numbers and positions on the sampling system at which they are connected as well as the sequence of injections have to be entered into the ATC configuration software. An automatic quality control of

raw measurements (Hazan et al., 2016) is performed on the calibration data based on a check of instrumental parameters such as temperature and pressure of the analyzer cavity, to ensure the instrument is working properly. For example, the typical accepted range for CRDS cavity pressure is 139.8 to 140.2 Torr. Then a flushing period, whose duration is configured via the ATC configuration tool, is automatically filtered out. From the validated measurements, one-minute averages are calculated, then the injection means for each of the calibration gases. The different levels of data aggregation (minute, injections, cycles)

are automatically checked, by comparing them to predefined standard deviation threshold values (see Table 2 and Hazan et al. (2016) for more details). Moreover, the water vapor content of the calibration gas is indirectly checked with a threshold on the difference between raw data and data corrected from water vapor effects. These thresholds are defined considering the instrument performances assessed by the ATC MLab and the station sampling setup and can be modified during the initial test period. For example, for a configuration without a drier, the typical humidity threshold for $CO_2$ will be 0.01ppm but

with a Nafion drier, as the dry cylinder gas will be humidified by the drying system, the threshold can be raised to 0.5ppm. The effect of long dry measure on wet air is discussed in the next section. Finally, for calibration cylinders and any others cylinders, it is advised to set the pressure on the regulators so that the pressure at the instrument inlet is slightly above the atmospheric pressure, thereby limiting a possible leakage contamination. However, the pressure should not be set at a too high or too low value in order to avoid a significant pressure jump while passing from cylinder to ambient air measurement

(which is usually done at an instrument pressure inlet below the atmospheric pressure). Indeed, laboratory tests have shown that transitory biases appear during step pressure change at the instrument inlet, the higher the step, the longer the return to equilibrium. In consequence, a large inlet pressure difference between ambient air and cylinder gas may result to an artifact, which will not have time to disappear over the time we are measuring the samples. During the initial test of the instrument, an acceptable range for the pressure is determined to help the PI set the regulators.

## 2.3 Quality control requirements

### 2.3.1 Target tank measurements

An important element of our quality control strategy for greenhouse gas measurements is to regularly measure a target gas of known concentration. On a daily basis we analyze air sampled from a short term target (every 7-10 hours during the initial test) and after each calibration we do the same with air from a long term target as shown in Figure 1. This ensures continuity

in the quality control as the long term target should last more than 10 years. Therefore, its chosen mixing ratio is relatively high (450ppm for $CO_2$ for background sites) compared to actual ambient air values in order to follow the increasing trend. It is recommended to send the long-term target, as well as the calibration set, for recalibration approximately every three years to CAL-FCL to investigate and take into account any possible composition changes in the gases, especially for CO.

Figure 1 shows the difference between the assigned and measured values over one month for the short-term (in green) and the long-term (in brown) targets. The instrument calibration dates are indicated at the bottom of the plot by the open orange circles. On this example, we can notice that after a calibration, the short-term target is significantly different from the other injections. Indeed after about six hours of dry air injection, the cavity is extremely dry compared to the usual injections after wet ambient air. This effect seen only on Cavity Ring-Down Spectroscopy (CRDS) analyzers (which however make the majority of the $CO_2$/$CH_4$/CO instruments in ICOS Atmosphere) is thought to be due to residual water on the pressure sensor in the instrument cavity (Reum et al., 2019). The extent of the effect is dependent on the analyzer, and thus this effect is important to assess as it allows to improve the bias estimate based on the target. Indeed, the mole fraction assigned by the CAL-FCL, as well as the target measured directly at the end of the calibration sequence in the field, are given in extremely dry conditions. While the instrument variability should be assessed with the short-term target measured regularly within ambient air, the measurement bias should be assessed only with the long term target and the short term target in extremely dry conditions, i.e. the target measured directly after calibration for an analyzer not equipped with an ambient air dryer. Depending on the instrument, this potential bias is more or less pronounced but do not exceed 0.05ppm for $CO_2$ and 0.4ppb for $CH_4$. It is part of the uncertainty of the water vapor correction estimated during the initial test at the Mlab. Moreover, one of the final test is to compare the tested instrument with the Mlab reference instrument whose samples (air and cylinder gas) are all dried. This allows to evaluate the weight of this bias.

### 2.3.2 Manual data quality control

All the data processed by ICOS ATC go through an automatic quality control (QC) based on various criteria (Hazan et al., 2016). However, as a second and final quality control step, the PI of the station has to review and validate the data on a regular basis using the logbook information from the station (e.g. contamination due to a maintenance on site). No data is flagged invalid without an objective reason.

To harmonize the quality control, the ATC provides dedicated software tools and organizes mandatory training that must be attended before beginning the operation at the station. The data validation is done with a software developed at the ATC directly on the server. On a daily basis, the station PI checks the ATC data products generated daily on the ATC website (https://icos-atc.lsce.ipsl.fr/dp) as an early detection of any issue related to the analyzers, sampling lines, data transmission or processing. On a weekly basis, raw greenhouse gas data have to be checked and (in)validated using ATC-QC software via a flagging scheme. Raw data are reviewed day by day. For valid data, we can choose additional information such as "Quality assurance operation" or " Non-background conditions" but this is not mandatory. Data have to be invalidated only for a objective reason which has to be chosen in a list to be able to carry on the QC. The reasons can be (non exhaustive list): "Calibration Issue", "Flushing period", "Maintenance with contamination", "Inlet leakage"... On a monthly basis, hourly average of greenhouse gas and meteorological data must also be verified.

During the initial test period, regular online meetings take place with the PI's and the ATC to review the data and assist the PI's. Their purpose are:

- To exchange expertise between ATC and the PI's, for example on how to QC local events (spikes) and how to interpret the data products. For example, for the spikes, a spike detection algorithm has been developed and is automatically applied on the data (El Yazidi et al., 2018),

- To make sure the data are regularly controlled,

- To benefit from local knowledge to explain patterns detected in the time series.

### 2.3.3   Intake line and water vapor correction tests

ICOS Atmosphere Station Specifications also require the station PI's to perform tests on the intake lines to investigate potential leaks and artifacts. These tests are extremely important because the target measurements, as described previously, do not make it possible to check for leaks on all parts of the air sampling lines. Consequently the PI's perform dedicated tests every six

months inside the measurement shelter on the different parts of the sampling system (including filters, valves..) and every year for the entire sampling lines running outside. Ideally, for this last test, a test gas should be injected through each sampling line on the outside structure (tower) in order to test the leakage and the inner surface artifacts. The test through the whole sampling line allows as well to calculate the sample residence time. However, for convenience, we propose to replace this test by a leak test of these outside sampling lines for lines younger than 10 years as for these lines, we expect that contamination like

bacterial build-up that could cause biases will remain rare and the leak test will suffice to identify cracks.

Dry air from a cylinder is measured first as close to the instrument as possible but without disconnecting the line to the instrument and this measure is considered the reference. Then, for the shelter test, the same gas is injected upstream of all the sampling parts in the shelter, at the outside sampling line connection point in the shelter, usually a filter as shown in Figure 2, which shows a typical set-up for an ICOS station. The two injection points are shown on the figure by the blue circles. During

this test, it is important to adjust the applied pressure of the test gas cylinder in order to reproduce the same pressure conditions than when ambient air is sampled. To avoid emptying the cylinder, the flushing pumps are off during the test upstream of all the sampling parts. If no significant bias is observed, we can consider that there are no leaks and that no component is causing an artifact. Significant means higher or very close to the WMO compatibility goals taking into account the water vapor uncertainty and the bias of the instrument determined at the ATC MLab. In the case of the entire lines, the test can either be done as per

the shelter test but with the gas injected through the whole line (usually by connecting the tops of the spare line and of the intake line and injecting the gas at the bottom of the spare line) or by closing the top of the intake line, creating a vacuum and then checking if this vacuum is holding over time. This last test only informs about leak presence but is easier to perform. The test through the whole lines is recommended for lines older than 10 years. In addition to the regular frequency, these leak tests must be carried out after any modification of the sampling lines.

Another source of uncertainty and possible biases is the water vapor effect on the measured gases. Tests at the MLab have shown that these effects can change over time and in a different way for each species and are visible on all instruments and technologies tested up to now (CRDS, Fourier Transform InfraRed spectroscopy (FTIR), Off-Axis Integrated Cavity Output spectroscopy (ICOS-OA)). If the instrument has not been tested at the MLab within the last year and no drying system is used, the PI needs to perform a new water vapor assessment to evaluate if the water vapor correction has indeed changed over time.

This test consists in injecting with a syringe at least three times a small droplet (0.2mL) directly in the inlet of the analyzer or through a filter when the analyzer does not have an internal filter to humidify a dry gas from a cylinder (with ambient air mixing ratios) and let it dry to obtain the profile of the trace gas versus the amount of water vapor (Rella et al., 2013).

## 2.4 Metrics for the station labeling

The decision taken by ICOS General Assembly to label a station or not has to be based on objective criteria, known in advance and common for all sites. During the initial test period, different metrics are thoroughly investigated to make sure the measurements meet the ICOS specifications and quality standards required. We detailed them below and illustrate each of them with a figure from one labeling period or one site that we used in the reports. As a result, the figures do not show all stations or the most recent period but are here to illustrate how the report content looks and what information can be derived. Most of the

figures are automatically generated regularly and are available on the ATC website (https://icos-atc.lsce.ipsl.fr/dp) but some are specifically produced for the labeling reports.

### 2.4.1 Percentage of data validated by station PI's

Quality control by the PI's is paramount to ensure the high quality of the ICOS dataset. When preparing the reports, we make sure that the quality control is done as detailed in the AS specifications, i.e weekly for the greenhouse gas air raw

data and monthly to every two months for the hourly means/injections of greenhouse gas (air and quality control gases) and meteorological data.

Figure 3 shows the status of hourly data validation at a given time for six stations. The hourly means are computed automatically using minute means which are themselves computed using raw data. If there is at least one valid raw data within a given minute, the corresponding minute mean is considered as valid. Similarly, if there is at least one valid minute mean

to compute an hourly mean, the hourly mean is consider valid. There is no automatic quality control criterion applied to the hourly means, the criteria are only applied on the raw data. Valid data are shown in green, and invalid data in red. Dark colors indicate automatic (by the ATC software) validation, prior to the manual data inspection by the PI (light colors). Dark red color will usually indicate flushing periods or instrument failure (hence no data while the database expects some) that are automatically flagged. For each station, the analyzers are identified by their unique ICOS ID attributed by the ATC. This is the number

shown in the second column in the graph. The third column shows the sampling heights. In this plot, however, we will mostly focus on the amount of manual validation to ensure that the data have been indeed controlled by the PI's. All intervention of the PI's to flag data are done through the ATC software and are recorded for traceability. When raw data are rejected, the PI has to select a reason for the problem within a predefined list of eleven issues (such as Flushing period, Instrument failure, Maintenance,...). The PI's validate the data on the raw level first then every one to two months on the hourly level. The second

validation on the hourly dataset aims to verify the longer term consistency of the time series. Every time a data flagging is performed either on the raw or hourly data, a reprocessing is automatically applied to the other aggregated levels (raw, minute, hour) for consistency.

### 2.4.2 Percentage of air measurement vs calibrations/target and flushing time

When switching from one sample to the other, there is some flushing period (defined by the PI in agreement with the ATC) that is automatically removed from the valid data. Figure 4 allows evaluating whether the time spent measuring "invalid" air is not too high compared to the time measuring "valid" air. It is mostly important for tall towers that switch between many levels and may end up spending too much time flushing. On the figure, when the percentage does not reach 100%, it means that the station has not yet provided data for the whole year. For example at Lindenberg (LIN), the analyzer for $CO_2$ and $CH_4$ was at least running since October 2017 but the $CO/N_2O$ analyzer was installed only in August 2018 thus showing only two months of data.

### 2.4.3 Optimized stabilization time to flush the sampling system

Related to the previous metric, to ensure the optimal time spent measuring ambient air and also to save calibration and target gases, the stabilization time needed for each gas tank is evaluated and optimized where necessary. If we observe that from one gas to the other, the stabilization period is significantly different, it can be a sign of leak or problem in the setup that will be reported to the PI. The sample is considered stable when the difference between a given minute averaged data point on the gas injection and the last injection point (after 30 minutes of measurement) is lower than 0.015ppm for $CO_2$, 0.25ppb for $CH_4$ and 1ppb for CO. These thresholds are determined considering the WMO recommendations (WMO, 2018) and expectations from the instrument performances (see Yver Kwok et al. (2015)).

On Figure 5, we show the average differences for all tanks over a period of six months at Monte Cimone (CMN). During that time, the short-term target has been injected 335 times, the long-term target 15 times and there are 240 injections for the calibrations (15 calibration sequences with four cylinders and all cycles taken into account, here four). For the calibrations, we show the average difference of all the calibration cylinders and cycles. Here, we see that the short term target and the calibration air stabilizes faster then the long term target, about 6 versus 18 minutes for $CO_2$. This can be a sign of leak or more likely be due to the fact that the long term target is measured only once every two weeks and in consequence the pressure regulator installed on the tank is flushed less often and requires a longer flushing time to reduce possible cumulative artifacts related to the pressure regulator inner parts in static mode.

### 2.4.4 Instrumental drift and optimization

The instrument response may drift over time, which is usually the case for some ICOS-compliant CRDS analyzers in $CH_4$ (Yver Kwok et al., 2015). This drift is corrected by the data processing using regular calibration sequences. Depending on the drift rate and its linearity, the frequency of the calibration may have to be adapted. Following the observed time evolution of the calibration gases allows tracking if one gas is behaving significantly differently than the others which could be caused by a drift in the cylinder or a leak in the setup (Figure 6). For some instruments, such as the ones measuring CO and $N_2O$, we also observe short-term drifts, in the scale of hours to days. In this case, we use a "short-term working standard" (STWS or reference) to correct for such drift. This standard is calibrated by the twice-monthly calibrations. By looking at the last

calibration injections, we can assess the number of cycles needed to reach the required stability and optimize this calibration sequence (Figure 7). The first cycle is always rejected by default as the samples are not yet well dried. The stability is estimated on the 2 to 3 following injections. In the examples presented on Figure 7, we see that after the first rejected injection, the spread between the other cycles is below 0.01ppm in $CO_2$. This shows that reducing the calibration sequence from four cycles to three is possible to save gas without reducing quality.

### 2.4.5 Temperature dependence of the instruments

A few of the instruments tested at the ATC MLab have shown a significant sensitivity of the GHG measurements to temperature. On site, the temperature variation is supposed to be small but in case of problem with the air conditioning, we can use the target gas measurement to evaluate the impact of the temperature changes on the measurements as seen in Figure 8. Currently, there is no correction derived from the MLab tests applied in the ICOS data processing. For instruments showing a significant variability due to the temperature or other parameters (i.e. leading to exceed the WMO goals within the observed range of temperature), it is recommended to use of a short term working standard, in order to correct the short term variability induced by these sensitivities. The target tank cannot be used for this or any correction as it is a quality control gas which is taken into account for data uncertainty assessment, contrary to the short term working standard.

### 2.4.6 Meteorological measurements

Meteorological parameters are mandatory as they are used to analyze the atmospheric signals measured at the station location, and associate them to regional or large scale processes. During the initial test period, the ATC checks that the sensors are compliant against the list from the AS specifications and the data are transmitted correctly to the AC data unit database for all mandatory levels (see Figure 9). The ATC checks the data availability and consistency. The ATC performs simple filtering on the raw data based on valid ranges (min/max values) for the five mandatory species which are pressure, temperature, relative humidity, wind speed and wind direction. Except for relative humidity, the data are also marked as invalid if the measurement is constant for more than X minutes in a row. X is set to ten for the wind variables and to 60 for the other species. This criterion is used to cope with blocked sensors. ATC is also working on a model versus measures comparison with the European Center for Medium-range Weather Forecasts (ECMWF) data to highlight potential drifts or outliers. In terms of instrumentation, ATC is working on instating a two-year recalibration for the humidity sensors that are the ones drifting the fastest over time. If meteorological data are sent to the database at the same time as the greenhouse gas data and not at the end of the test period, they can be used to understand the variability in the greenhouse gas data.

### 2.4.7 Diagnostic parameters

For the diagnostic parameters (room temperature, instrument and flushing pump flow rates), similarly to the meteorological parameters, the ATC checks that they are available and consistent. If they are present over the whole test period, they can be used to monitor that the room temperature is well controlled and that the instrument and flushing pump flow rates are as stable

as expected. Higher flow rates can indicate leaks whereas decreases over time will most probably indicate that filters are getting clogged and need maintenance or that there is an obstruction in the sampling line (see Figure 10 bottom panel). The measure of the flow rates is also important to estimate the time delay between the air sampling at the top of the sampling lines and the measurement in the analyzer. This delay can be significant for the highest level of a tall tower and needs to be known to correctly attribute a timestamp to the measured air. Finally, the instrument flow rates can be used to estimate the lifetime of the gas cylinders.

### 2.4.8   Time series and associated uncertainties

For most of the stations that enter labeling Step 2, data have already been collected before the initial test period which allows an analysis of the previous year (Figure 11), previous month (not shown) and to plot a wind rose (not shown) allocating the mixing ratios to the wind direction and intensity for the whole year and by season. This figure is of interest to evaluate the influence of different sources that can surround the site at a more or less large scale. On the yearly figure, ATC look for patterns in the target gases (biases, drifts), data gaps, outliers, whereas the ambient air signals are much more visible on shorter period's figures. To allocate mixing ratios to wind sectors or to compare two instruments measuring the same species, all instruments and sensors have to have access to a time server and update their clock regularly.

With the measurement of the target gases, uncertainties comparable to the ones estimated in the MLab during the initial test are calculated as well as bias to the CAL-FCL assigned values as shown in Figure 12. These values are compared to the MLab values and if very different can be a hint that the setup has introduced a problem that needs to be identified and solved. Details on the calculations of these uncertainties are found in Yver Kwok et al. (2015). Shortly, CMR stands for Continuous Measurement Repeatability and is calculated here using the the monthly average of the standard deviations of short-term target raw data over one minute intervals. LTR, Long-Term Repeatability is the standard deviation of the averaged short-term target measurement intervals over three days. Here we can see that before November 2017, the target variability was high leading to high and variable LTR and bias. After November 2017 and a change of parts in the sampling set-up that we discuss in section 4.6, the LTR and bias show a significant improvement.

## 3   Presentation of the 23 labeled stations

The 23 labeled stations described here passed Step 1 between 2016 and 2019 (see Table 3). For most of them, this was a straightforward step. For four of them (Ispra (IPR), Observatoire de l'Atmosphère du Maïdo (RUN), Lutjewad (LUT) and Karlsruhe (KIT)), additional documents and preliminary studies were requested mainly to address potential local contaminations. After Step 1 approval, they entered Step 2 up to two-and-a-half years later depending on the existing infrastructure and instrumentation. At the end of the initial test period, a scientific report summing up the setup done during this period and the resulting data was sent to the PI and the station was proposed for labeling. The stations were then approved by the ICOS General Assembly.

In November 2017, the first four ICOS atmosphere stations were labeled following this procedure. In May 2018, the next seven were approved. In November 2018, another six were labeled. In May and November 2019, two and four respectively were approved. Seven stations are located in Germany, three in mainland France, three in Sweden, three in Finland, two in Italy (with one operated by the Joint Research Centre of the European Commission), one in the Netherlands, one in Norway, one in Switzerland, one in Czech Republic and one in La Réunion Island, operated conjointly by the Belgian and French national networks. Ten countries plus the European Commission out of 12 ICOS RI member countries are represented.

The 23 stations cover the majority of western Europe with the most southerly in Italy and the furthest north on Svalbard in the Arctic Circle plus one located in the Indian Ocean in the Southern Hemisphere (see Figure 13 and Table 3). The first ICOS compliant data date from the end of 2015 for two German stations (which began measurements following ICOS procedure before the operational phase and so the Step 1 application) and the more recent stations have compliant data since September 2019. Fifteen stations are continental sites equipped with tall towers with up to six air sampling levels. Four are classified as mountain stations, two as coastal sites with one sampling level and two are remote sites.

The AS specifications also provides guidelines for sampling periphery such as regulators, sampling valves and tubing. This allows a high level of standardization while allowing flexibility for the PI's to design the setup. For the gas distribution to the analyzer, the required equipment is a rotary valve from Valco (model EMT2SD). Alternative options may be accepted after proving their suitability (dead volume, material compatibility, absence of leakages). A drier is recommended but not required.

Thirteen sites use the required rotary valve to switch between levels and quality control gases. Three use only solenoid valves while the last seven use the required valve for the quality control gases but solenoid valves to switch between levels. These valves have proven suitable during audits ran by the MobileLab on two of these sites or during the intake line tests run every six months.

Four sites (Hyltemossa (HTM), Norunda (NOR), Svartberget (SVB) and Hyytiälä (SMR) equipped with several sampling heights on tall towers use buffer volumes in order to have more hourly representative data at each sampling level. At the Swedish sites, buffer volumes of 8L are used with an integration time from 3.8min to 4.9min and a flushing rate between 1.6 to $2.1L\,min^{-1}$. At SMR, buffer volumes of 5.6L are flushed at $0.325L\,min^{-1}$ which gives an integration time of 16.9min. However, they lose the information about the short term variability of mixing ratios, which is essential for the application of the spike detection algorithm and therefore important for sites that often experience this type of signals. Of the nineteen sites that do not use buffer volumes, eight (Monte Cimone (CMN), Jungfraujoch (JFJ), Lutjewad (LUT), Pallas (PAL), Puy de Dôme (PUY), Observatoire de l'Atmosphère du Maïdo (RUN), Utö (UTO) and Zeppelin (ZEP) sample at a single height.

During the initial test period, two sites (Observatoire Pérenne de l'Environnement (OPE) and Křešín u Pacova (KRE)) were using cryogenic water traps to dry the air. IPR was using a compressor chiller set at a dew point of 5°C. ZEP was using a Nafion membrane into which all samples pass through and nineteen sites were not drying the air for their CRDS measurements. Out of these nineteen, four sites (Lindenberg (LIN), Karlsruhe (KIT), Ochsenkopf (OXK) and Steinkimmen (STE)) were also using OA-ICOS $N_2O$/CO instruments and using the recommended Nafion drier. In May 2020, seven sites (CMN, HTM, PUY, SVB, RUN, TRN (Trainou), JFJ) which were not equipped with a drier during the test period, had installed a Nafion drier for all their

samples while Hohenpeissenberg (HPB) and Gartow (GAT) had added a Nafion drier for their OA-ICOS $N_2O$/CO instruments. IPR stopped using its chiller in October 2018 then installed a Nafion drier in May 2020.

In Figure 14, the availability and distribution of data over the past year for $CO_2$, $CH_4$ and CO is shown for the 23 stations. Calibration is in red, target in blue and air in green/gray. Gray and darker color are invalid data. 100% is reached when data
are available for the full year. For some stations, two instruments are used to measure $CO_2$, $CH_4$ and CO. For other sites, due to instrumental failure, a new instrument has replaced the previous one and together the data availability reaches 100%. Table 4 shows for each station and species, aggregating instruments if needed, the percentage of ambient air and the percentage of rejected data (automatically and manually, including flushing for air and cylinders) over the past year or available period if shorter. Thus the percentages will look different than on the figure for stations that had not data for a full year.

**4   Some lessons learned from labeling 23 stations: data, troubleshooting and maintenance**

**4.1   Calibrations**

During the initial test period, all stations followed the recommendations and ran calibrations with 4 cycles of 30 minute injections every 2 weeks. At the end of this phase, it was noticed that for most of the stations, three cycles were enough to get precise results and thus proposed as a solution to save gas. By 2020, 9 out of 23 had opted to reduce their numbers of
calibration cycles. For most of the stations, it was recommended to keep a two week frequency, mostly to accommodate for the random variation of CO for the $CO_2$/$CH_4$/CO CRDS analyzer. Table 5 shows the instrumental drift observed during the test period over at least 6 months at each station for the different greenhouses gases. For the length of the injections (detailed in the next section), it was more variable: for 14 stations, the performances made it possible to reduce by five to ten minutes the injection length while for the other nine sites, it was advised to stay with the same schedule or even to increase the flushing
time from 10 to 15 minutes.

**4.2   Tank stabilization time**

Table 6 presents the average, minimum and maximum stabilization time observed at the stations for the different types of gases (short-term, long-term target and calibration) while the individual results for each station are shown in Figure 15. On average, all samples are stable after 10 minutes for all species with some being stable after only one minute. In the case of CO, the
high values at 30 minutes were not due to a problem in the setup of the station, but rather to the fact that the two instruments concerned were very noisy (as shown in the tests performed at the ATC MLab prior to installation at the station) and thus the values were systematically ranging above and below the threshold. For $CH_4$, the CRDS lines at STE showed an outlier with a short-term target only stable after 21 minutes. Another one (at SMR) had a long term target slightly above the others, stable after 15 minutes.
For $CO_2$, whose criterion is the more stringent, the time to reach stability is higher and with a larger spread over the stations. However, only five sites had samples needing more than 15 minutes to reach stability. For three sites (CMN, LIN, SVB), only

the long-term target was of concern, probably due to the fact that it is injected less often as discussed in section 2.4.3. A longer stabilization time was found in $CO_2$ for the calibration for GAT CRDS lines and for STE CRDS lines both in $CO_2$ and $CH_4$ for the short-term target. Leak tests were recommended to identify the problems. At GAT, the stabilization time is now equivalent for all types of samples between 10 and 12 minutes. For STE as well, the stabilization time is now reduced to about 12 minutes

for both $CO_2$ and $CH_4$.

## 4.3   Uncertainties

Figures 16, 17 and 18 show the uncertainties and bias of the short-term target for $CO_2$, $CH_4$ and CO as defined in Figure 12 and discussed in section 2.4.8. They are calculated using one year of data. For each station, we show two boxes and a dot. The left box (in pink when large enough to see) uses data from the year before the date of the end of the initial test period. The

right box (in blue when large enough to see) uses the last year from March 2019 to March 2020. The red dot shows the MLab initial test values. For example, for a site labeled in May 2018, we use data from April 15 2017 to April 15 2018. For the sites labeled last, the periods are almost the same. For some sites, instruments have changed since the labeling and thus there is one box per instrument.

For all species, the CMR and LTR are close to the values calculated at the Mlab and the bias for all sites is mostly within the

WMO recommendations. This shows that the setup of the station has not decreased the instrument performances seen at the MLab. Some of the sites show outliers in the left boxes (initial test period) due to problems that have been solved during the initial test period. At HTM, the instrument is known to be poor in CO. It was replaced between May 2018 and September 2019 by an instrument performing better (not shown). Unfortunately, due to a storm at the station, this instrument was damaged and the old one reinstalled. At JFJ, the reason for the $CO_2$ scatter is disccused in section 4.6. For the CO, the outlier in the LTR

is due to a single injection that was about $4ppb$ higher that the other injections. No particular reason could be identified but the subsequent injections went back to the normal values. Outliers at SVB in the recent period are related to two problems that have since been solved. During winter 2019/2020, the PI noticed a large temperature variation in the shelter. After inspection, they found holes in the walls and plugged them in February 2020. At SMR, the outliers are linked to a failure in the instrument pump, which finally broked down and was then changed.

## 25   4.4   Intake line tests

During the initial test period or within the next 6 months, 17 out of the 23 stations performed intake line tests. During this exercise, the bias between a reference measurement taken at a free valve port and measurements taken upstream all the ambient air sampling parts inside the shelter at the outside sampling line connection (see Figure 2) was calculated and is shown in Figure 19 as measured-reference. This value compared to the WMO compatibility goal and the instrument MLab performances helps

to determine if there is a leak or an artifact in the system. As seen in Figure 19, only three stations found a significant bias when testing their system. This highlights the quality of the work done during the setup as well as the right choices of parts that the ATC deemed compliant. This also demonstrates the importance of carrying out the intake line tests regularly as the target gas measurements alone will not show leakages or artifacts on the ambient air sampling system. Out of the three sites,

TOH experienced a positive bias (Measured – reference) hence a leak. However, the subsequent test showed that the leak was coming from the tubing attached to the test gas cylinder and not from the setup itself. KRE and TRN found a negative bias which implies a $CO_2$ absorption. In the case of KRE, it was attributed to a water trap. When changed, the bias disappeared. For TRN, it was due to a piece of stainless steel tubing that may have been contaminated. Even though the nature of the
contamination in the stainless steel tubing has not been clearly identified, it seems related to a water effect on the tubing inner surface. For both stations, only one sampling level was affected. Station PI's from all ICOS stations have been warned about such possible contamination with this material and advised to always use new tubings when doing modifications in the sampling system.

## 4.5  Water correction test

During the initial test period or within the next 6 months, 18 out of the 23 stations performed the water vapor correction assessment test (hereafter called droplet test) described earlier in section 2.3.3. Out of the five that did not, one is using a drying system and one had its instrument tested less than a year before the initial test period. Eight sites showed a drift in $CO_2$ up to $0.05 \mathrm{ppm}\,\mathrm{yr}^{-1}$ at 3% humidity but mostly insignificant at lower values. No significant drift was observed for $CH_4$. Most drifts were seen for CO. Eleven sites showed a drift from 2 to 7 $\mathrm{ppb}\,\mathrm{yr}^{-1}$ at 3% and still from 0.5 to 5 $\mathrm{ppb}\,\mathrm{yr}^{-1}$ at 1%
humidity. Only six sites had no data gaps in the water vapor profile.

   Two main conclusions can be drawn from the analysis of the water correction test reports: two-thirds of the sites did not manage to obtain a full water vapor profile covering all the water vapor levels. Often, levels around 0.5% and above 2% are data-depleted. Another 15-20 droplet tests have been performed on instruments not yet labeled and show the same pattern: it is difficult to perform a good droplet test despite following all the steps of the protocol. This can be due to factors such as
the atmospheric pressure, the room temperature, the filter at the analyzer inlet,... However, even with missing data points, the shape of the water vapor influence on $CO_2$, $CH_4$ and CO is still visible and gives us a qualitative information about the water correction drift over time. Here, we observe no significant drift in $CH_4$, about half of the analyzers drifting in $CO_2$ and two-thirds of the analyzers show a drift in CO. This CO drift is significant and can reach up to $7 \mathrm{ppb}\,\mathrm{yr}^{-1}$ at 3% humidity and still up to $5 \mathrm{ppb}\,\mathrm{yr}^{-1}$ at 1%. However, when looking at more recent droplet test, the CO drift seems to stabilize after a few years.
These tendencies are illustrated in Figure 20. Due to the still low number of tests and the known possible erratic behavior of the water vapor effect on CO (Zellweger et al., 2019), this tendency needs to be confirmed. In any case, this highlights the need for either a quantitative way of estimating the water correction on site or the need for drying. ICOS ATC strongly recommends the installation of a drier which can be a Nafion membrane (no maintenance, controlled artifact with a MD model from PermaPure, following the reflux method described in Welp et al. (2013)) or a cryogenic water trap (at least at -50°C, regular maintenance,
no artifact).

## 4.6  Troubleshooting

Thanks to the works of the PI's following good practice and the ICOS specifications from the installation of their stations, the initial test period did not highlight a high number of cases with major problems. However, for five stations, the labeling had to

be postponed by 6 months. In the case of JFJ, the labeling was postponed due the instability of the target gas measurements. For $CO_2$, the standard deviation reached 0.1ppm with a bias of 0.09ppm (see Figure 16. At that time, a Perma Pure PD-series polytube Nafion drier was used and degradation of it or the counterflow pump was suspected. Changing the pump did not solve the problem but as soon as the Nafion was removed, the $CO_2$ standard deviation dropped to 0.02ppm and a bias of 0.02ppm. This type of Nafion has subsequently been investigated by the ATC, and deemed unsuitable for ICOS use as the possible artifacts induced by polytube Nafion can be significant due to their large inner surface and can be problematic. ICOS ATC recommends the use of single tube Nafion instead.

Another problem was detected at HTM that potentially affected NOR and SVB as the three stations were using the same system on their tall towers. The lower sampling level with a low flow rate showed a positive bias in CO compared to the other sampling levels that could not be explained by the origin of air masses. The intakes in the tower were heated to avoid condensation or ice. This bias was caused by those heated intake cups. It is unclear whether the excess CO originated from the heater or from a plastic part. To be on the safe side, the heating was turned off and the plastic parts replaced by Teflon ones. After these modifications, the CO bias was no longer observed. About four months of data were invalid for this lower level at HTM as a result. For the fifth station, KRE, the delay was caused by problems in a faulty drying system as explained above.

## 5   Conclusions

In this paper we have presented the process used to label ICOS atmosphere stations. This process ensures that the produced data are of high-quality. For the 23 labeled sites, the PI's implemented the specifications and completed training at ATC before beginning the initial test phase or just after. This allowed for high-quality data with only few problems detected. The problems were handled and solutions were implemented to get good quality data. Other stations would benefit from the lessons learned during the certification process. From these data, we can conclude on different topics:

- Regular calibrations (twice a month) are important in particular to follow the CRDS analyzer's CO variability,

- Reducing the number of calibration injections (from four to three) in a calibration sequence is possible and allows to extend the calibration gas lifespan and maximize atmospheric measurement time,

- It is often possible to reduce the time of the target and calibration gas injections but it still has to be determined site by site depending on the instrument intrinsic performances and the stabilization time (mainly dependent of the upstream sampling system),

- Measuring the target in ultra-dry conditions (i.e. after the calibration cycles) is important to evaluate potential biases to the assigned values,

- The on-site water correction test does not deliver quantitative results for any species. A better way has to be devised to reevaluate the water vapor correction. Use of a drying system is recommended for all analyzers, either a Nafion membrane or a cryogenic water trap (at least at -50°C).

- Regular intake line tests are useful to detect artifacts and leaks.

**Table 1.** ICOS atmosphere station parameters (from the ICOS atmosphere station specifications ERIC (2020)). *not yet required for the labeling, see Section 2.1

| Category | Gases, continuous | Gases, flask sampling | Meteorology | Eddy Fluxes |
|---|---|---|---|---|
| **Class 1** mandatory parameters | **$CO_2$, $CH_4$, CO**: at each sampling height | **$CO_2$, $CH_4$, $N_2O$, $SF_6$, CO, $H_2$, $CO_2$ $^{13}$C, $^{18}$O and $^{14}$C:** weekly sampled at highest sampling height* <br> $^{14}$C (radiocarbon integrated samples): at highest sampling height | **Air temperature, relative humidity, wind direction, wind speed**: at highest and lowest sampling height <br> **Atmospheric pressure at the surface** <br><br> **Planetary boundary layer height*** | |
| **Class 2** mandatory parameters | **$CO_2$, $CH_4$**: at each sampling height | | **Air temperature, relative humidity, wind direction, wind speed**: at highest and lowest sampling height <br> **Atmospheric pressure at the surface** | |
| Recommended parameters | $^{222}$**Rn, $N_2O$, $O_2/N_2$ ratio** <br><br> **CO** for Class 2 stations | **$CH_4$ stable isotopes, $O_2/N_2$ ratio** for Class 1 stations: weekly sampled at highest sampling height | | **$CO_2$**: at one sampling height |

**Table 2.** ATC MLab typical thresholds for calibration quality control on standard deviation (sd). The minute data sd takes into consideration the sd of each minute of the injection. The injection average sd takes into consideration the sd of all the minutes of one injection and the cycle average sd takes into consideration the sd of the 2 to 3 injections.

| Species [unit] | Minute data sd | Injection average sd | Cycle average sd |
|---|---|---|---|
| $CO_2$ [ppm] | 0.08 | 0.06 | 0.05 |
| $CH_4$ [ppb] | 0.8 | 0.5 | 0.3 |
| CO [ppb] | 7 | 5 | 1 |

**Table 3.** Information about the 23 labeled stations: site name, three-letter acronym, coordinates, sampling level heights, station surroundings, class, date of the first ICOS compliant data and date of labeling

| Site | Acronym | Coordinates | Sampling levels | Type of station | Class | First ICOS data since | Labeled in |
|------|---------|-------------|-----------------|-----------------|-------|-----------------------|------------|
| Gartow, Germany | GAT | 53.0657°N, 11.4429°E, 70 masl | 30, 60, 132, 216 and 341 magl | Continental, flat with forests and fields | 1 | 10/05/2016, 04/04/2017 for CO | Nov 2017 |
| Hohenpeissenberg, Germany | HPB | 47.8011°N, 11.0246°E, 934 masl | 50, 93 and 131 magl | Continental, hilly, close to Alps, forests and meadows | 1 | 17/09/2015, 15/02/2017 for CO | Nov 2017 |
| Hyltemossa, Sweden | HTM | 56.0976°N, 13.4189°E, 115 masl | 30, 70 and150 magl | Continental, flat with forests | 1 | 16/04/2017 | May 2018 |
| Hyytiälä, Finland | SMR | 61.8474°N, 24.2947°E, 181 masl | 16.8, 67.2 and 125 magl | Continental, hilly with boreal forests | 1 | 04/04/2017 | Nov 2017 |
| Ispra, Europe | IPR | 45.8147°N, 8.6360°E, 210 masl | 40, 60 and 100 magl | Continental, close to local sources | 2 | 15/12/2017 | Nov 2018 |
| Jungfraujoch, Switzerland | JFJ | 46.5475°N, 7.9851°E, 3572 masl | 10 magl | Mountain, background | 1 | 12/12/2016 | May 2018 |
| Karlsruhe, Germany | KIT | 49.0915°N, 8.4249°E, 110 masl | 30, 60, 100 and 200 magl | Continental, close to local sources | 1 | 16/12/2016, 31/01/2019 for CO | Nov 2019 |
| Křešín u Pacova, Czech Republic | KRE | 49.5720°N, 15.0795°E, 534 masl | 10, 50, 125 and 250 magl | Continental, hilly with forests and fields | 1 | 12/04/2017 | May 2018 |
| Lindenberg, Germany | LIN | 52.1663°N, 14.1226°E, 73 masl | 2.5, 10, 40 and 98 magl | Continental, almost flat with forests and fields | 1 | 08/10/2015, 24/08/2018 for CO | Nov 2018 |
| Lutjewad, The Netherlands | LUT | 53.4036°N, 6.3528°E, 1 masl | 60 magl | Continental, on the seaside, flat rural landscape | 2 | 13/08/2018 | May 2019 |
| Monte Cimone, Italy | CMN | 44.1936°N, 10.6999°E, 2165 masl | 8 magl | Mountain, background | 2 | 03/05/2018 | Nov 2018 |
| Norunda, Sweden | NOR | 60.0864°N, 17.4794°E, 46 masl | 32, 58 and 100 magl | Continental, flat with forests | 1 | 04/04/2017 | May 2018 |

| Site | Acronym | Coordinates | Sampling levels | Type of station | Class | First ICOS data since | Labeled in |
|---|---|---|---|---|---|---|---|
| Observatoire de l'Atmosphère du Maïdo, France/Belgium | RUN | 21.0796°S, 55.3841°E, 2154 masl | 6 magl | South hemisphere background | 2 | 17/05/2018 | Nov 2019 |
| Observatoire Pérenne de l'Environnement, France | OPE | 48.5619°N, 5.5036°E, 390 masl | 10, 50 and 120 magl | Continental, flat with fields, pastures and forests | 1 | 18/08/2016 | Nov 2017 |
| Ochsenkopf, Germany | OXK | 50.0300°N, 11.8083°E, 1015 masl | 23, 90 and 163 magl | Continental, hilly with forests | 1 | 25/09/2019 | Nov 2019 |
| Pallas, Finland | PAL | 67.9733°N, 24.1157°E, 565 masl | 12 magl | Continental background | 1 | 16/09/2017 | Nov 2018 |
| Puy de Dôme, France | PUY | 45.7719°N, 2.9658°E, 1465 masl | 10 magl | Mountain, background | 2 | 01/05/2016 | May 2018 |
| Steinkimmen, Germany | STE | 53.0431°N, 8.4588°E, 29 masl | 32, 82, 127, 187 and 252 magl | Continental, flat with fields and forests | 1 | 22/07/2019 | Nov 2019 |
| Svartberget, Sweden | SVB | 64.2560°N, 19.7750°E, 267 masl | 35, 85 and 150 magl | Continental, hilly with forests | 1 | 01/06/2017 | May 2018 |
| Torfhaus, Germany | TOH | 51.8088°N, 10.5350°E, 801 masl | 10, 76, 110 and 147 magll | Continental, low mountain range with forests | 2 | 12/12/2017 | Nov 2018 |
| Trainou, France | TRN | 47.9647°N, 2.1125°E, 131 masl | 50, 100 and 180 magl | Continental, flat with fields and forests | 2 | 11/08/2016 | Nov 2018 |
| Utö, Finland | UTO | 59.7839°N, 21.3672°E, 8 masl | 57 magl | Continental, island | 2 | 09/03/2018 | May 2019 |
| Zeppelin, Norway | ZEP | 78.9072°N, 11.8867°E, 474 masl | 15 magl | Arctic background | 1 | 27/07/2017 | May 2018 |

**Table 4.** Percentage of ambient air and invalid data (manual and automatic, including flushing for air and cylinders) over the whole dataset at each station, aggregating instruments if more than one was used, over the period March 2019-March 2020 or over the available period within these dates if shorter (for stations labeled in November 2019 notably). CO percentage are indicated between brackets when different i.e. when measured with another analyzer than $CO_2$ and $CH_4$ and installed later.

| Station | Ambient air (%) | Invalid data (%) |
|---------|-----------------|------------------|
| GAT | 55 | 37 |
| HPB | 68 | 24 |
| HTM | 75 | 20 |
| SMR | 91 | 6 |
| IPR | 68 | 28 |
| JFJ | 91 | 5 |
| KIT | 70 | 24 |
| KRE | 69 (65) | 26 (30) |
| LIN | 70 (66) | 24 (26) |
| LUT | 94 | 3 |
| CMN | 89 | 7 |
| NOR | 78 | 19 |
| RUN | 88 | 7 |
| OPE | 73 | 25 |
| OXK | 61 (57) | 31 (35) |
| PAL | 93 | 4 |
| PUY | 89 (82) | 6 (13) |
| STE | 49 (62) | 45 (30) |
| SVB | 75 | 18 |
| TOH | 70 | 25 |
| TRN | 74 (68) | 20 (27) |
| UTO | 92 | 4 |
| ZEP | 94 | 3 |

**Table 5.** Instrumental drift calculated as the difference between the first and last data point divided by the number of years. NS: not significant. NA: Not applicable i.e. species not measured or not enough data to estimate a trend. In the case of CO CRDS, the calibration usually shows variability but no drift.

| Station | CO2 drift (ppm yr$^{-1}$) | CH4 drift (ppb yr$^{-1}$) | CO drift (ppb yr$^{-1}$) |
|---------|--------------------------|--------------------------|--------------------------|
| GAT | 0.1 | 3.5 | NS |
| HPB | 0.1 | 1 | 0.5 |
| HTM | NS | NS | NS |
| SMR | 0.1 | 1.5 | NS |
| IPR | 0.1 | 2.5 | 1 |
| JFJ | NS | NS | NS |
| KIT | NS | 1 | NS |
| KRE | NS | NS | 1.7 |
| LIN | NS | NS | NS |
| LUT | 0.1 | 2 | NS |
| CMN | 0.1 | 1.5 | NS |
| NOR | NS | NS | NS |
| RUN | NS | NS | NS |
| OPE | NS | 0.5 | NS |
| OXK | NA | NA | NA |
| PAL | 0.12 | 2.5 | NS |
| PUY | NS | 2 | NS |
| STE | 0.2 | 2 | NA |
| SVB | NS | NS | NS |
| TOH | 0.1 | 3 | NA |
| TRN | 0.15 | 3 | NS |
| UTO | 0.2 | 3 | NA |
| ZEP | NS | 2.4 | NS |

**Table 6.** Cylinder sample stabilization time estimated during the initial test period. The mean, minimum and maximum (mean;min;max) for the 23 stations are shown. *Too noisy for the algorithm to find the stabilization time.

|                  | CO2 (minutes) | CH4 (minutes) | CO (minutes) |
|------------------|---------------|---------------|--------------|
| Short-term target | 8 ; 3 ; 18   | 6 ; 1 ; 21    | 2 ; 1 ; 4    |
| Long-term target  | 9 ; 1 ; 20   | 3 ; 1 ; 14    | 5 ; 1 ; 29*  |
| Calibration       | 5 ; 1 ; 18   | 1 ; 1 ; 2     | 2 ; 1 ; 4    |

**Figure 1.** One month of target gas injections for $CO_2$ (ppm or $\mu$mol.mol$^{-1}$), shown as the difference of calculated vs. assigned mxing ratios. Short term target is plotted in green, whilst long term data are in brown. The calibration dates are shown by the light orange open circles. Cylinder number (D******), mean values ($\pm$ X), point-to-point variability (Ptp) and difference to the assigned value (Diff) are displayed above the figure.

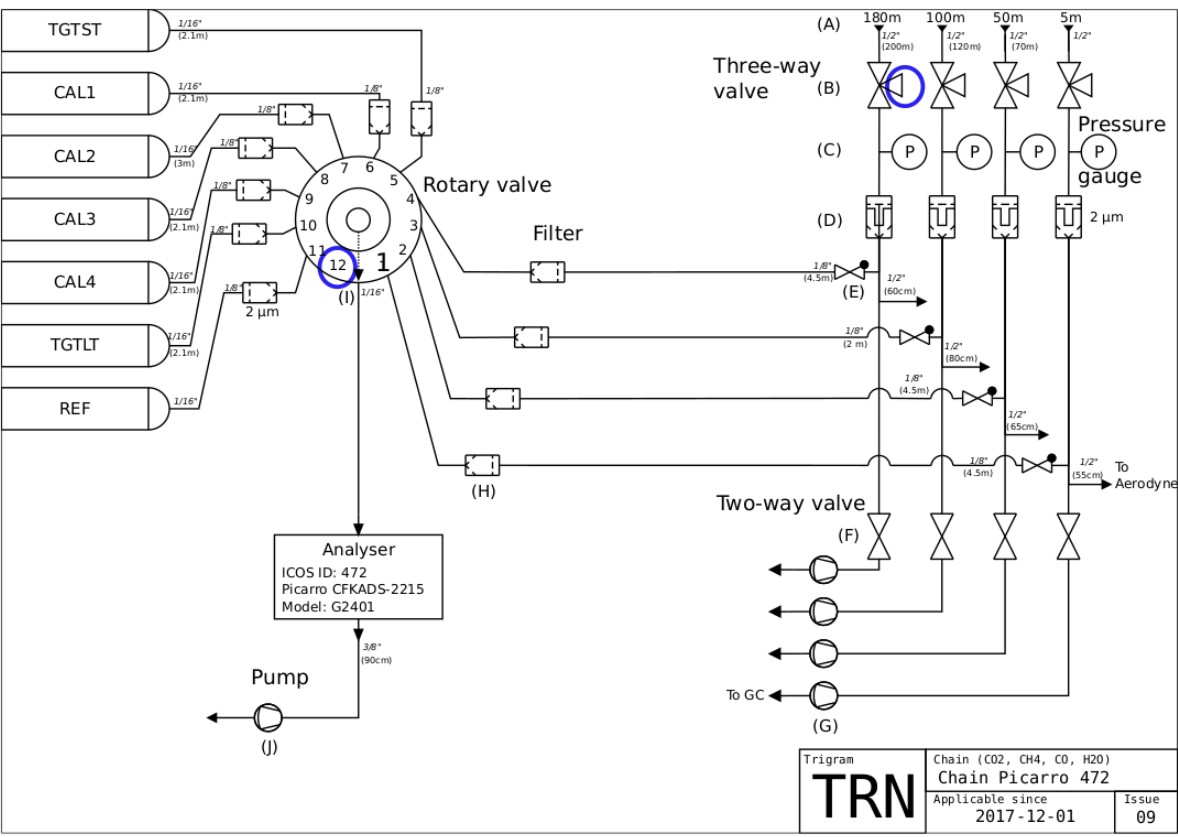

**Figure 2.** Station schematics with injection points in blue for the shelter test. Example of Trainou tower, France. Different parts include valves, filters, pressure gauges and pumps. One element of each type is legended for clarity.

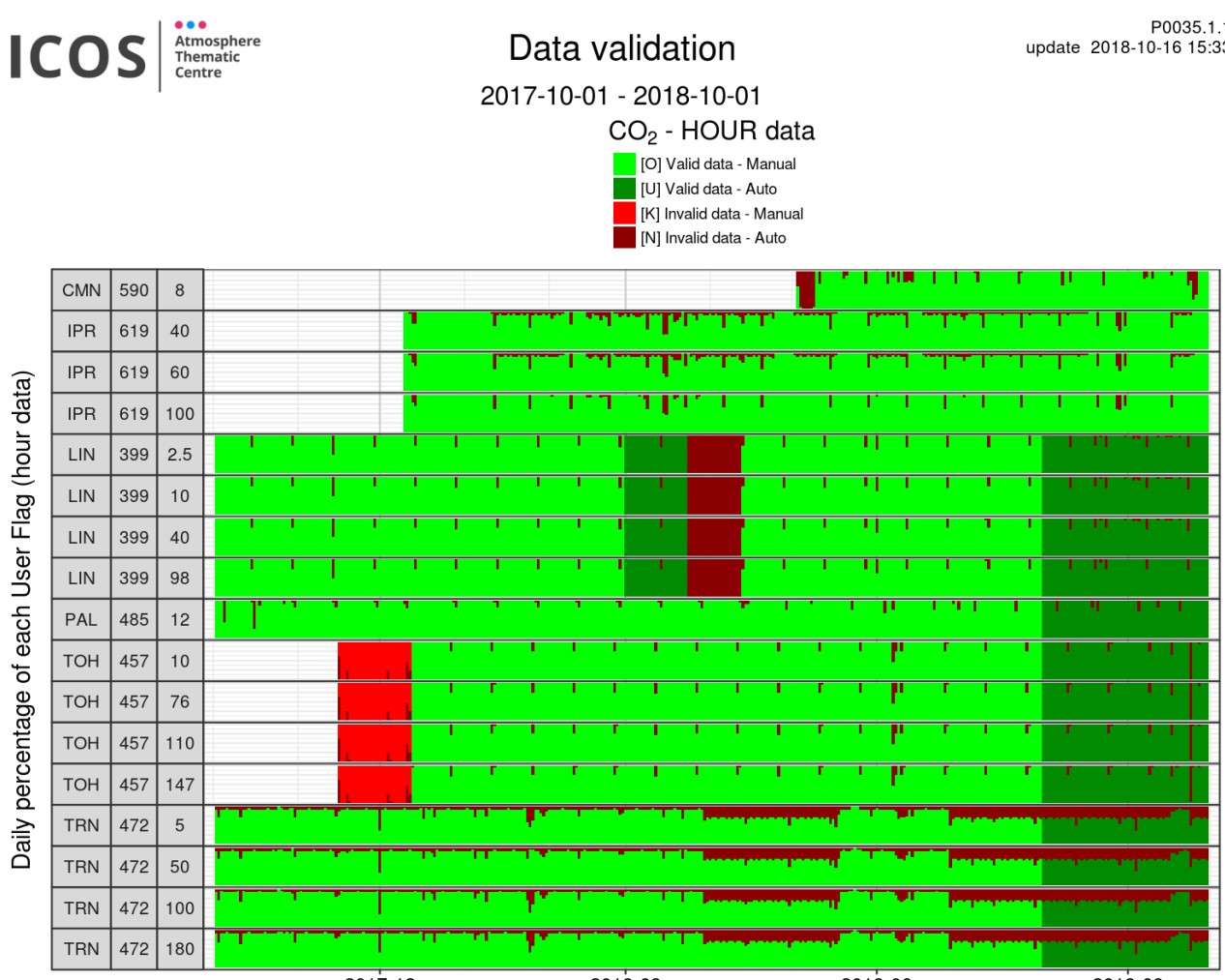

**Figure 3.** Example of data validation for six stations: CMN (Monte Cimone, Italy), IPR (Ispra, Italy), LIN (Lindenberg, Germany), PAL (Pallas, Finland), TOH (Torfhaus, Germany), TRN (Trainou, France). Dark colored data are data controlled by the software only. Light colored data are controlled by the PI. Green is valid, red invalid. On the left, the first column shows the station acronym, the second column the ICOS ID of the instrument and the third the sampling height.

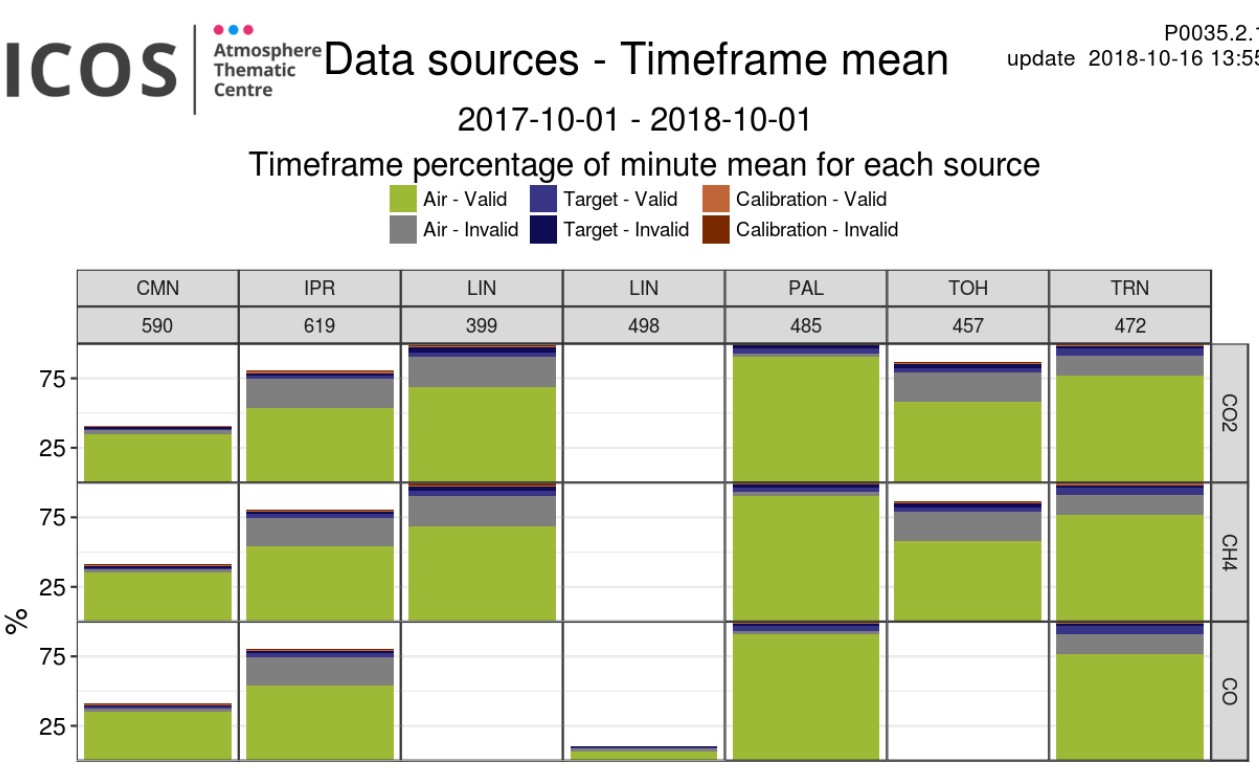

**Figure 4.** Data distribution between ambient air, target and calibration gases for the same 6 stations as Figure 3 for $CO_2$, $CH_4$, CO. Calibration is in red, target in blue and air in green/gray. Gray and darker color are invalid data. Less than 100% data availability means that the instrument was installed less than 12 months ago. The first line shows the station acronym and the second line the instrument ICOS ID. In the case of LIN, two instruments are used to measure $CO_2$, $CH_4$ and CO.

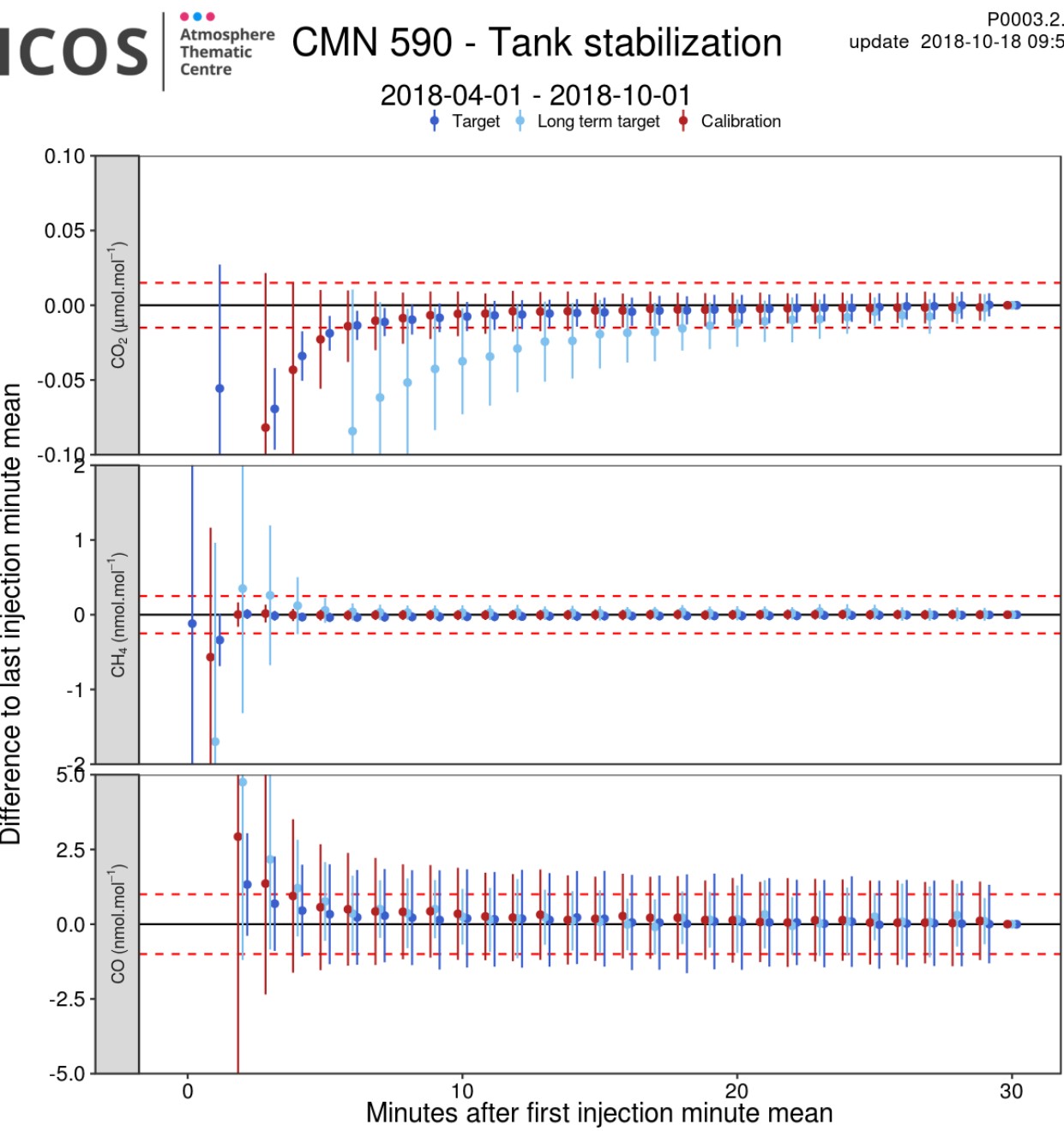

**Figure 5.** Difference between the last injection minute averaged data point and the rest of the injection for cylinder samples for Monte Cimone station (CMN) for instrument 590. All the injections over the last six months are averaged. Short-term target is in dark blue, long-term target in light blue and calibration in red. Red dashed lines show the thresholds. Vertical lines on each point show the minute standard deviation.

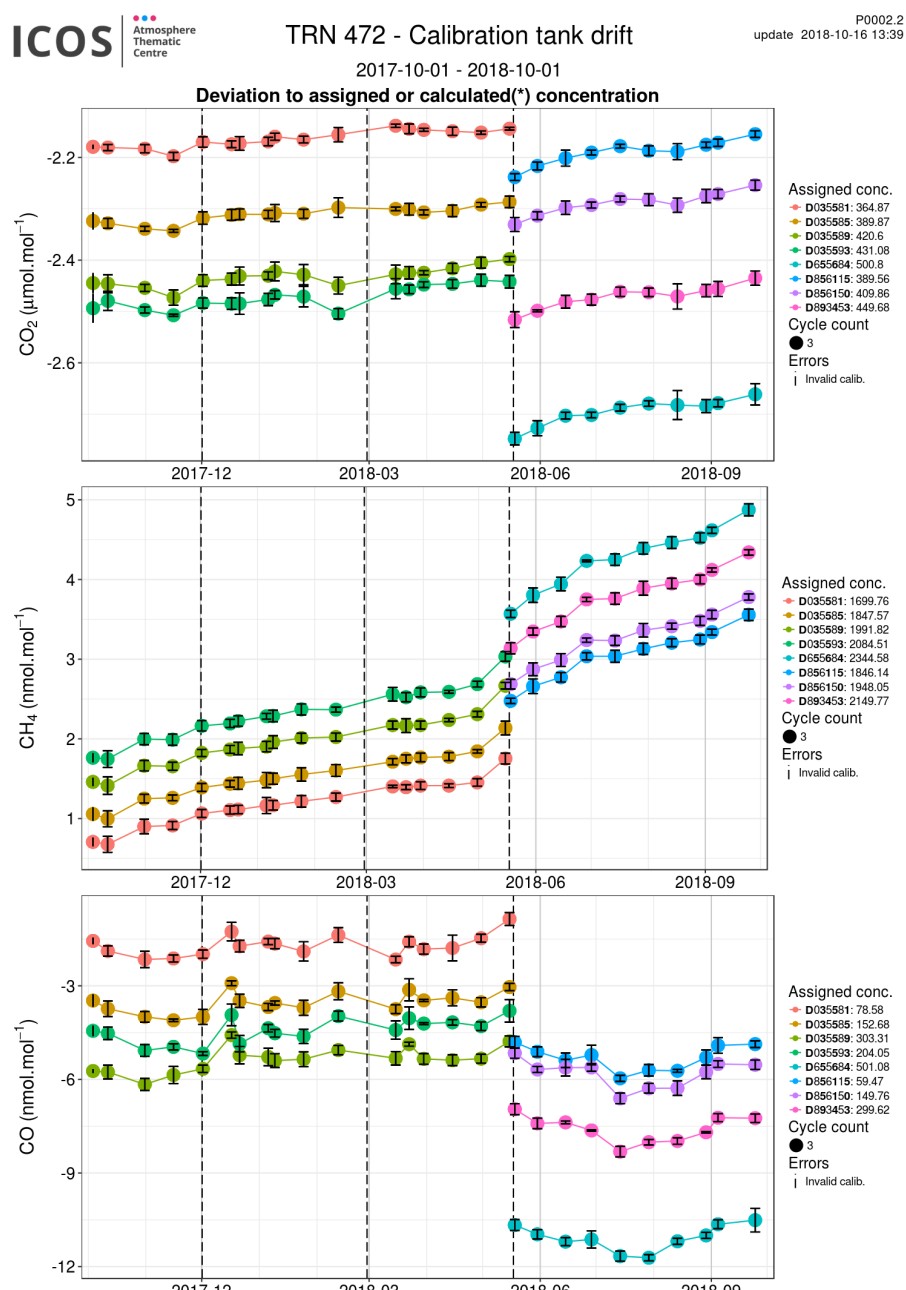

**Figure 6.** Evolution of the analyzer raw output when measuring different calibration gases with respect to the assigned values over a year at Trainou station (TRN) for instrument 472. Each calibration cylinder is shown with a different color. Assigned values are indicated on the right for each cylinder (D******).

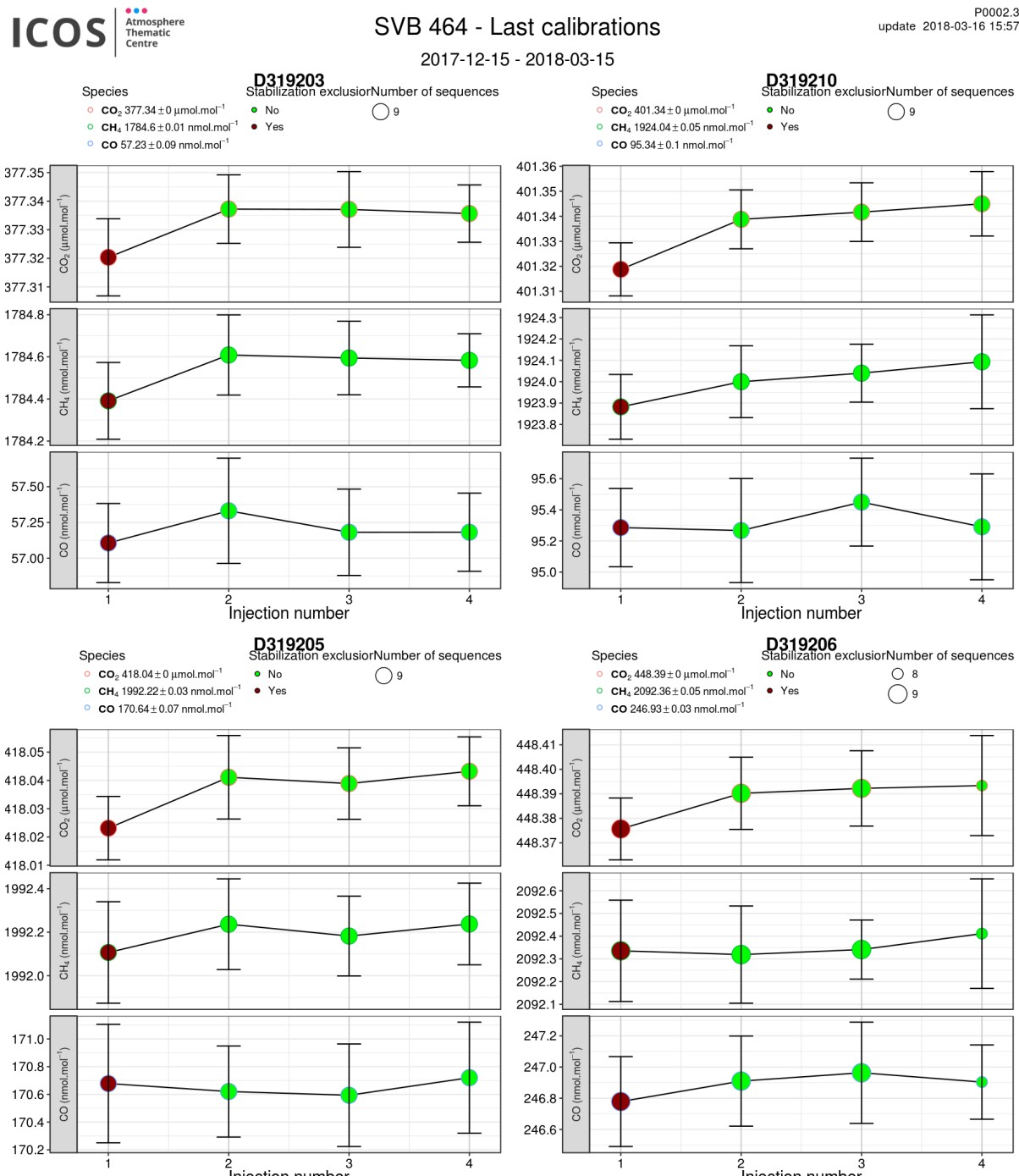

**Figure 7.** Average of each cycle injection for the last calibrations over three months at Svartberget station (SVB) for instrument 464. Green dots are data used for the calibration correction, red is rejected for stabilization. The number of calibrations is shown of the top right. Assigned values are on the top left. Cylinder number (D******) is shown at the top of each panel.

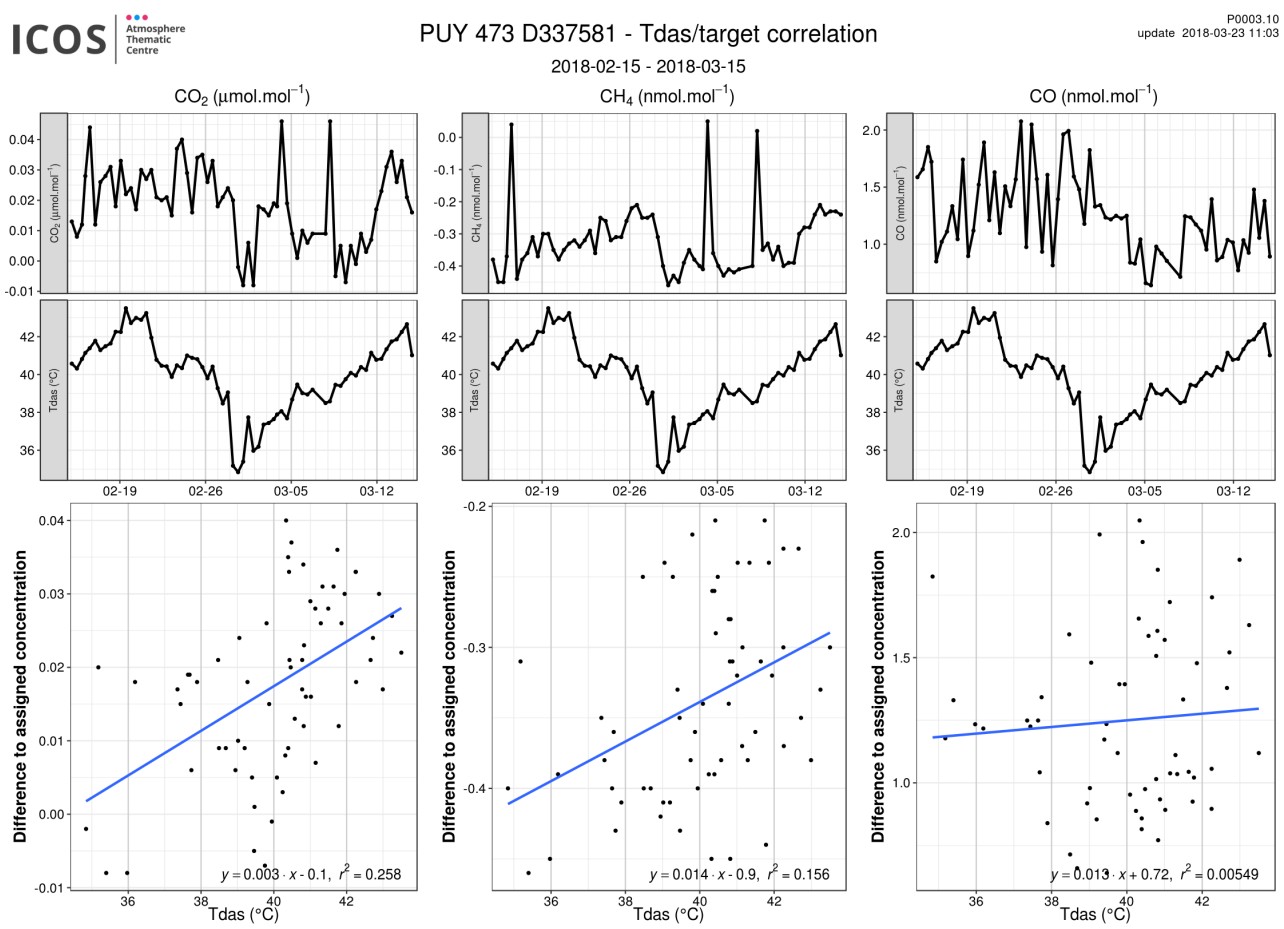

**Figure 8.** Temperature influence on the measurements at Puy de Dôme station (PUY) for the instrument 473 and the target cylinder D337581. On top, greenhouse gas and instrument temperature (Tdas) measurements against time. On the bottom, greenhouse gas measurements against instrument temperature. In most of the cases, no dependencies are seen.

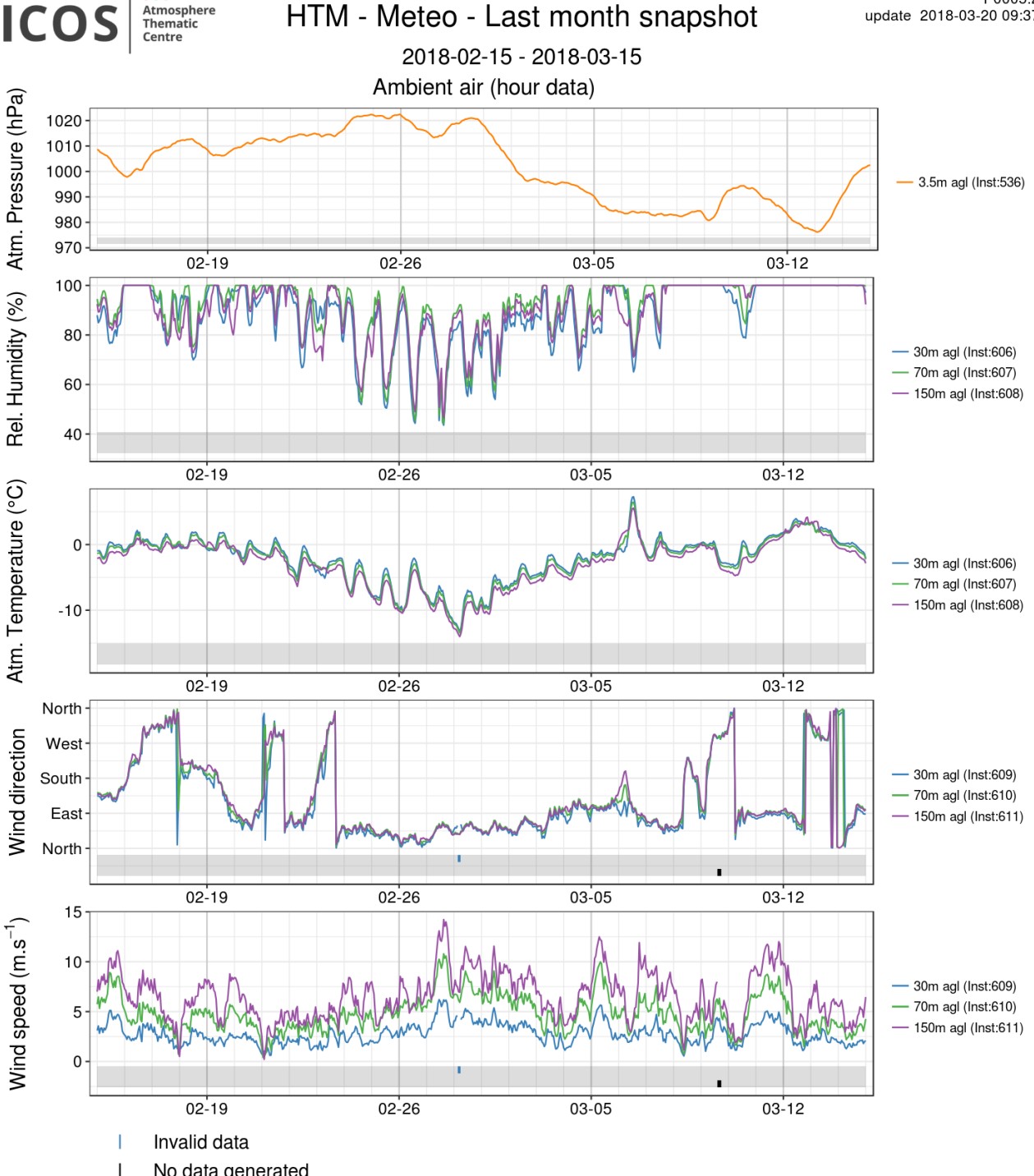

**Figure 9.** One month of meteorological parameters at Hyltemossa station (HTM). From top to bottom: atmospheric pressure, relative humidity, atmospheric temperature, wind direction and wind speed. The data at the different levels are plotted with different colors.

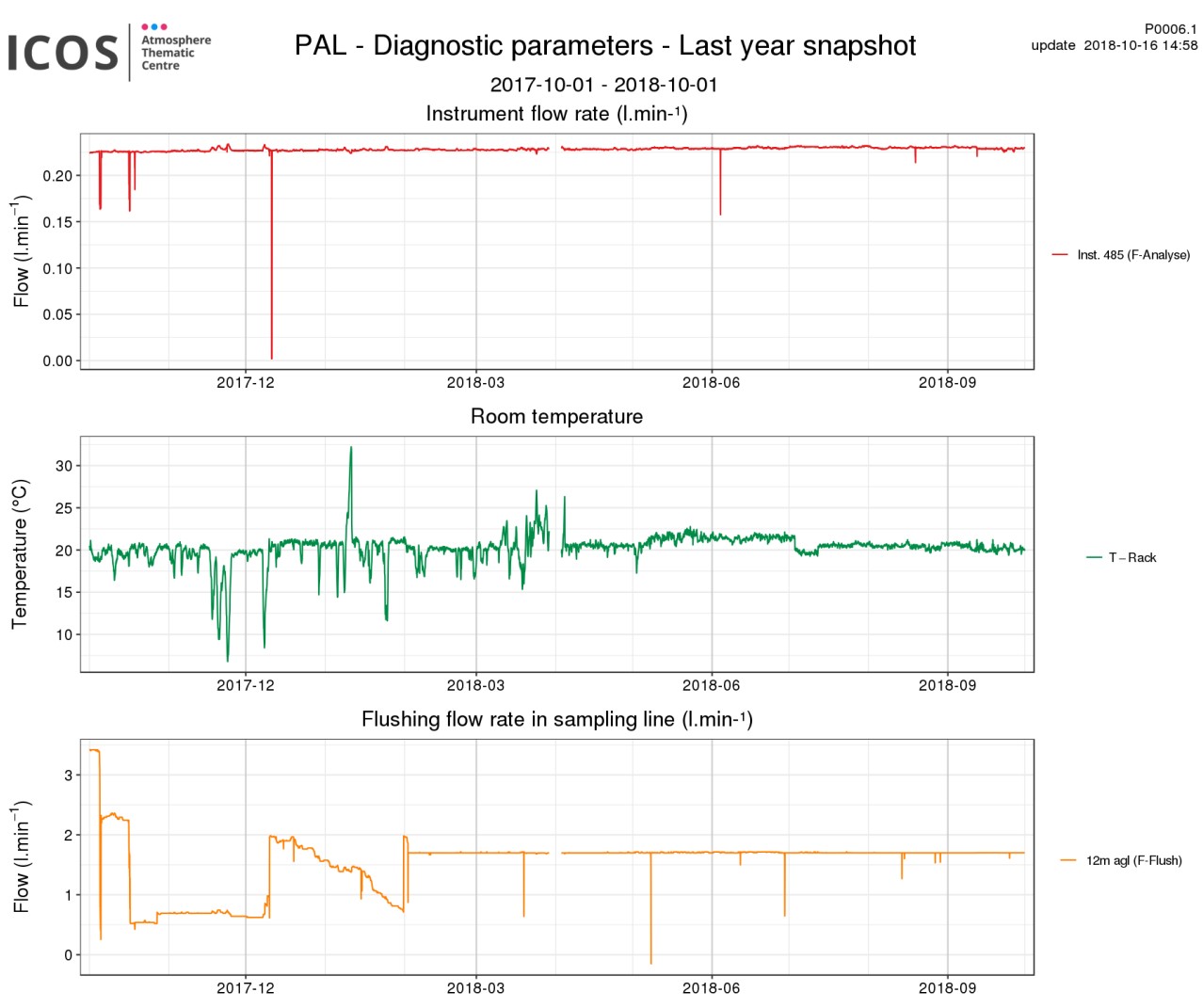

**Figure 10.** One year of diagnostic parameters at Pallas station (PAL). From top to bottom: instrument flow rate, room temperature and sampling line flushing flow rate.

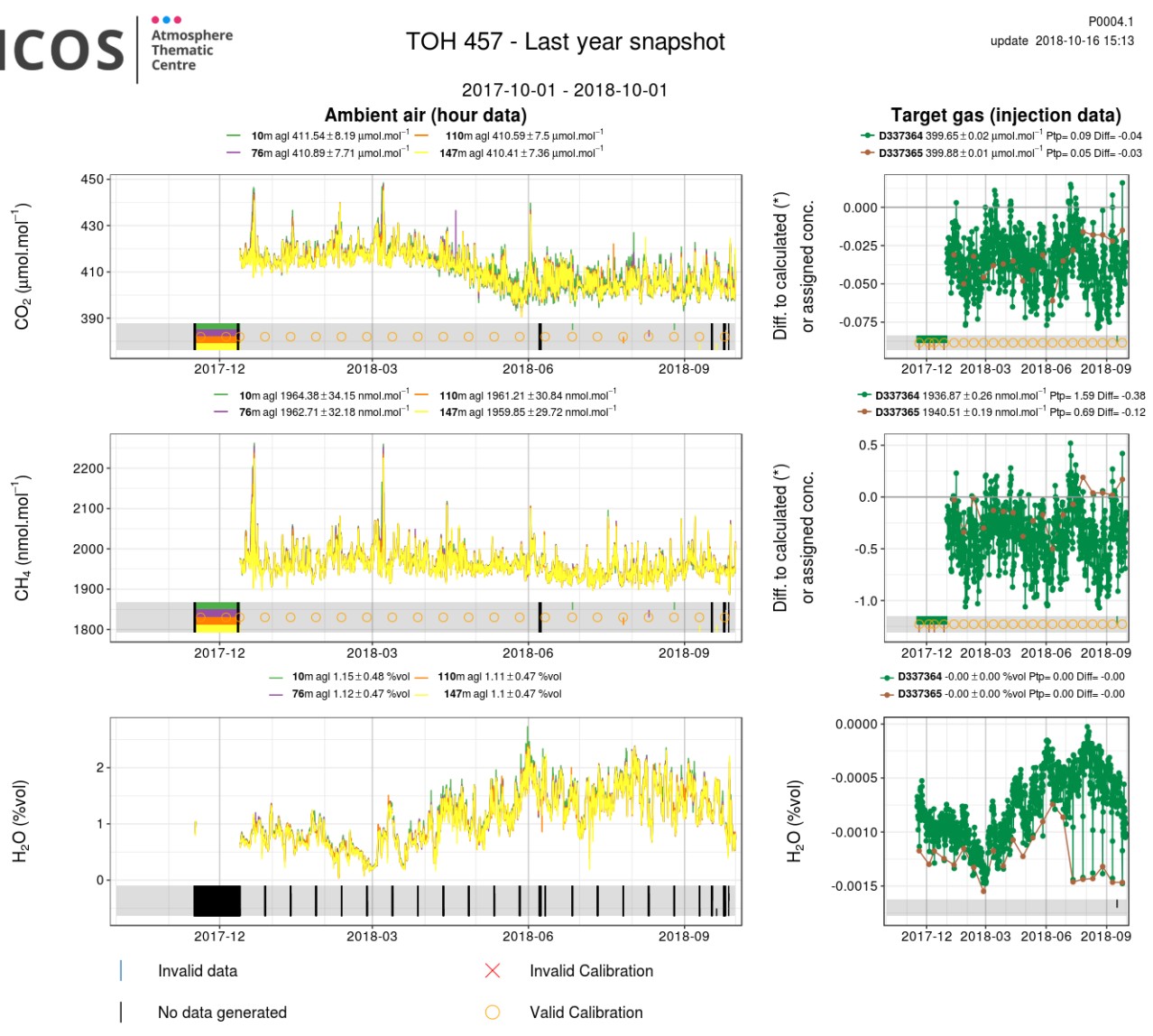

**Figure 11.** One year of hourly averaged greenhouse gas measurements at Torfhaus station (TOH) for instrument 457. The different levels and targets are plotted with different colors. Ambient air is plotted on the left, target measurements on the right. Calibration are shown with orange open circles. Invalid data are shown at the bottom of each plot. Cylinder number (D******), mean values (± X), point-to-point variability (Ptp) and difference to the assigned value (Diff) are displayed above the target gas plots. Measured GHG are shown on the different panels from top to bottom.

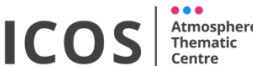

**Figure 12.** Last year of greenhouse gas measurements along with estimated uncertainties at Jungfraujoch station (JFJ) for instrument 225. Continuous measurement repeatability (CMR) and long-term repeatability (LTR) are calculated as in Yver Kwok et al. (2015). The short-term target bias is calculated as the difference between the hourly average of the short-term target injections and the value assigned by the FCL-CAL. On the top panel, the ambient air data are compared to the MHD (Mace Head, Ireland) Marine smooth curve, derived from atmospheric measurements made at Mace Head, an historical European background site. The smooth curve is calculated using NOAA's CCGCRV function (https://www.esrl.noaa.gov/gmd/ccgg/mbl/crvfit/crvfit.html, Thoning et al. (1989)).

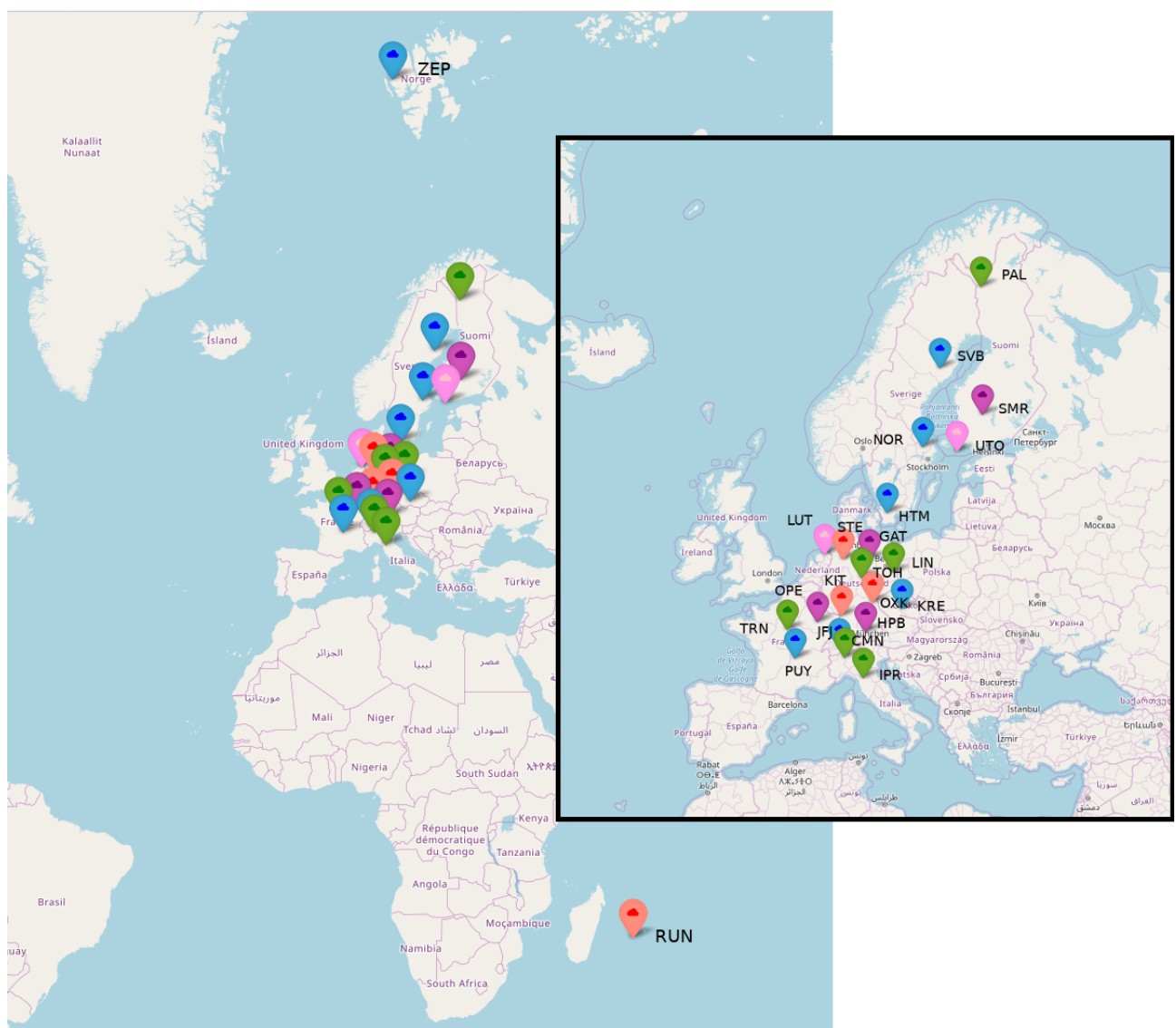

**Figure 13.** Map showing the 23 labeled stations before 2020. The colors show when the station was labeled. First, purple (11/2017), then blue (05/2018), green (11/2018), pink (05/2019) and red clouds (11/2019). On the right, a zoom shows the 21 labeled stations located in Europe mainland. Base map provided by OpenStreetMap contributors. See Table 3 for the acronyms and more details about each station.

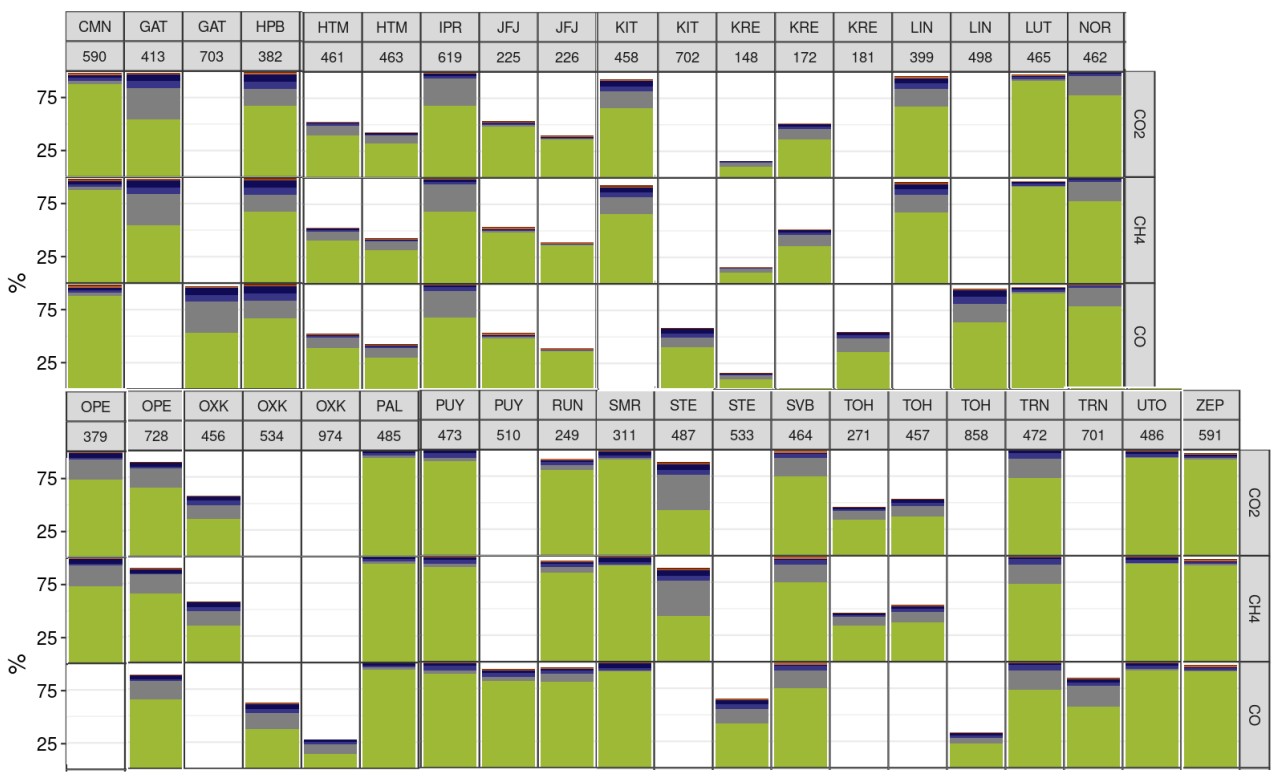

**Figure 14.** Data distribution between ambient air, target and calibration gases for the 23 stations for $CO_2$, $CH_4$ and CO over the past year. Calibration is in red, target in blue and air in green/gray. Gray and darker color are invalid data. 100% is reached when data are available for the full year. For some stations, two instruments are used to measure $CO_2$, $CH_4$ and CO (see their ICOS ID on the second line). For other, due to instrumental failure, a new instrument has replaced the previous one. See Table 3 for the acronyms and more details about each station.

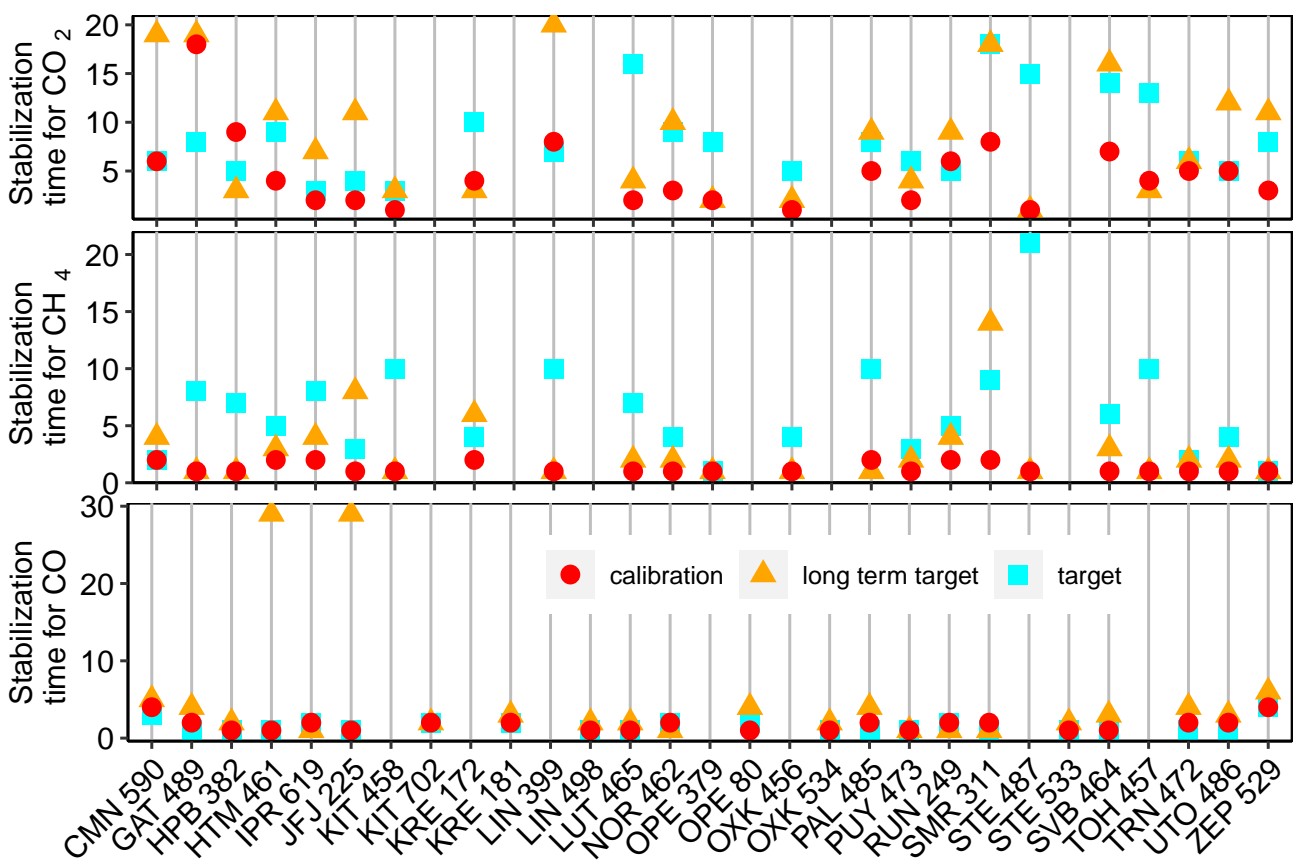

**Figure 15.** Stabilization time (minutes) for $CO_2$, $CH_4$ and CO at the 23 stations at the time of the labeling. Red show the calibration, green the long-term target and blue the short term-target. On the x-axis is shown the trigram of the station and the ICOS ID of the analyzer. Data from CRDS and OA-ICOS analyzers are shown.

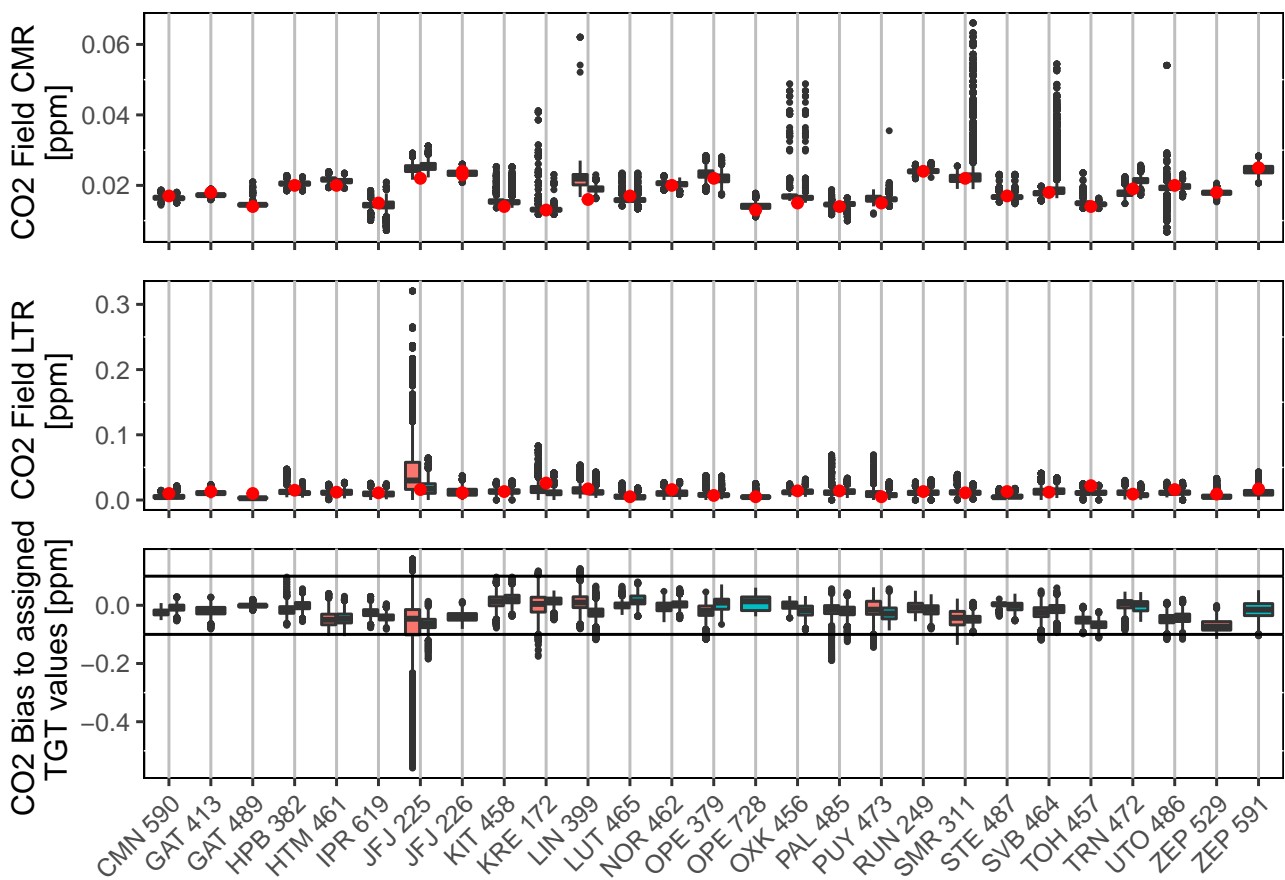

**Figure 16.** Uncertainties and bias to the short-term target for $CO_2$ defined as in Figure 12. The red dot shows the minute CMR and LTR from the MLab initial tests. The left box (pink) is calculated using data from the year prior to labeling. The right box (blue) is calculated using data from March 2019 to March 2020. For Gat, 489 was prior to 413. For JFJ, 226 has replaced 225. At OPE, 729 is running in parallel with 379. At ZEP, 529 was prior to 591. The x-axis shows the site trigram and ICOS ID of the analyzer. The black lines in the bias plot show the WMO compatibility goals.

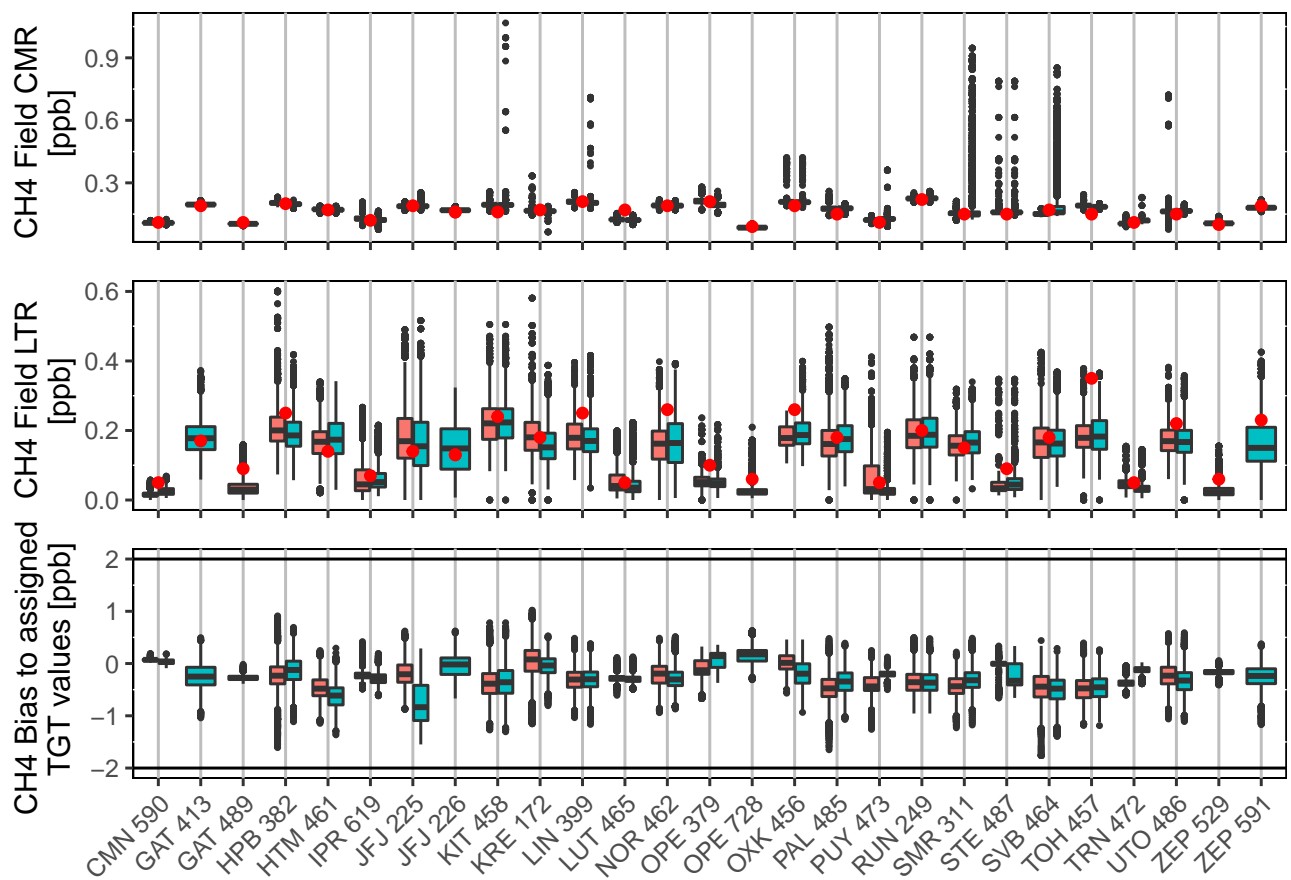

**Figure 17.** Uncertainties and bias to the short-term target for CH$_4$ defined as in Figure 12. The red dot shows the minute CMR and LTR from the MLab initial tests. The left box (pink) is calculated using data from the year prior to labeling. The right box (blue) is calculated using data from March 2019 to March 2020. For Gat, 489 was prior to 413. For JFJ, 226 has replaced 225. At OPE, 729 is running in parallel with 379. At ZEP, 529 was prior to 591. The x-axis shows the site trigram and ICOS ID of the analyzer. The black lines in the bias plot show the WMO compatibility goals.

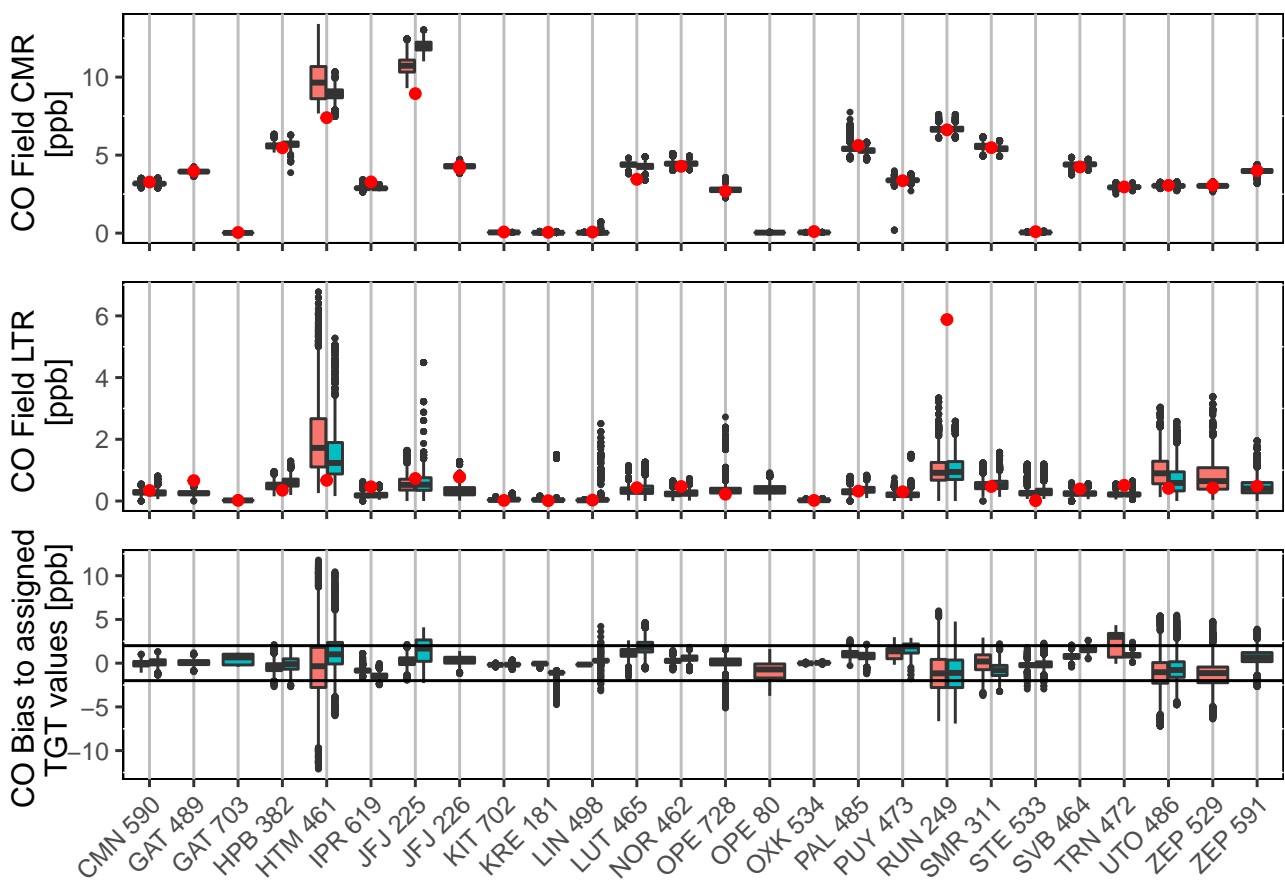

**Figure 18.** Uncertainties and bias to the short-term target for CO defined as in Figure 12. The red dot shows the minute CMR and LTR from the MLab initial tests. The left box (pink) is calculated using data from the year prior to labeling. The right box (blue) is calculated using data from March 2019 to March 2020. For Gat, 489 was prior to 413. For JFJ, 226 has replaced 225. At OPE, 729 is running in parallel with 379. At ZEP, 529 was prior to 591. The x-axis shows the site trigram and ICOS ID of the analyzer. The black lines in the bias plot show the WMO compatibility goals.

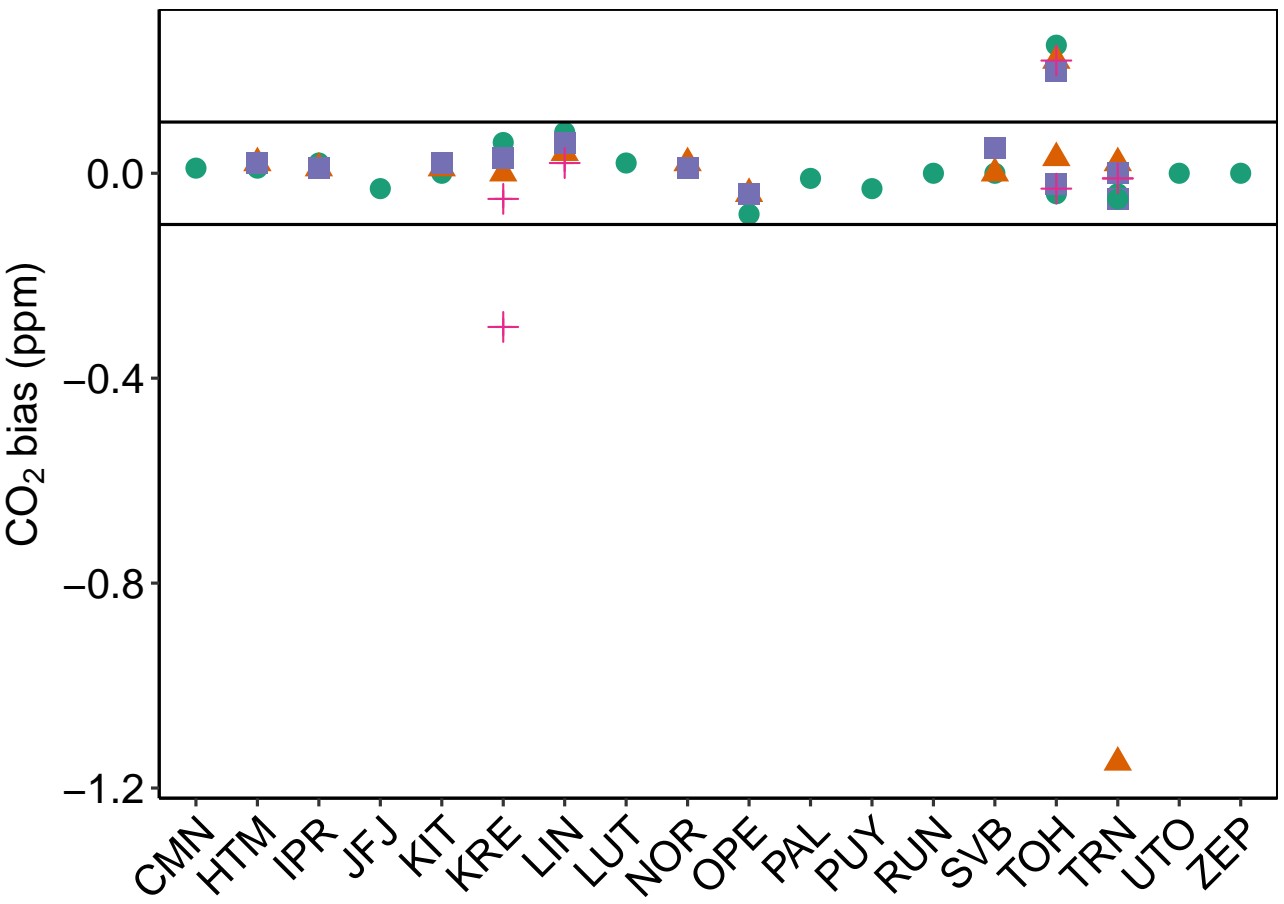

**Figure 19.** CO$_2$ difference between the reference injection and the injection before the shelter first element. The different colors and shape show the difference at each sampling levels (green dot: lower level, orange triangle: second level, purple square: third level, pink cross: fourth level). Same color and shape at one given site indicate that several tests have been performed (e.g. TOH). The black lines show the WMO compatibility goals.

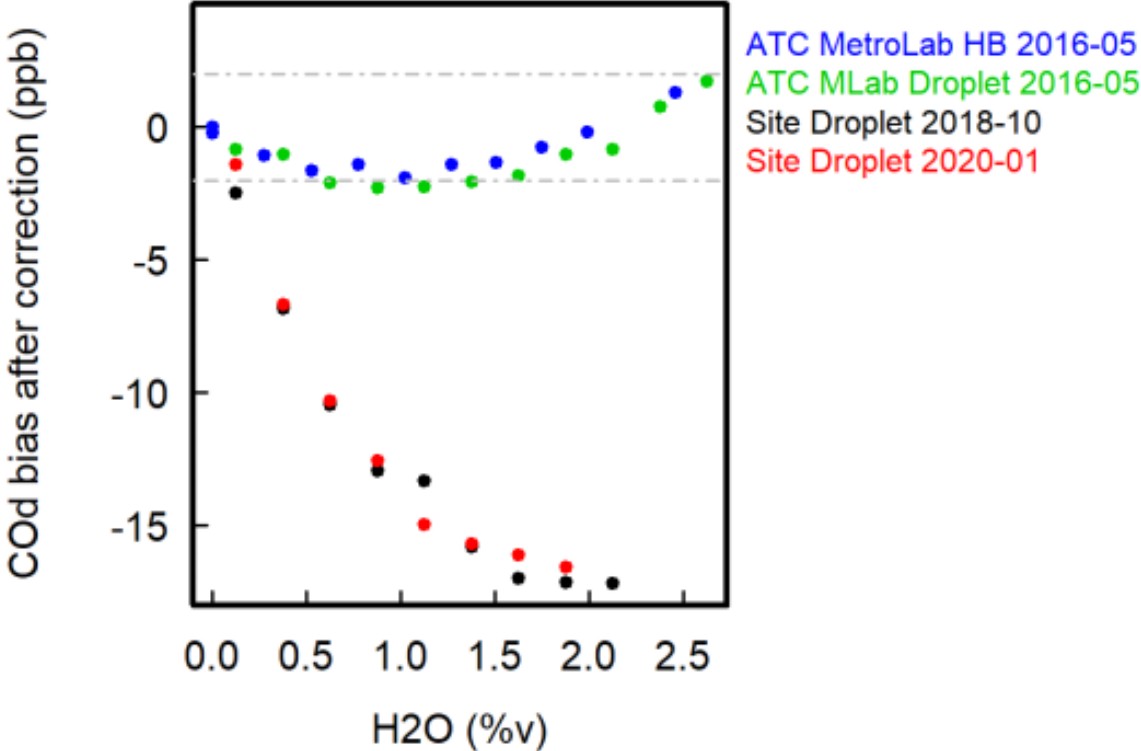

**Figure 20.** Example of water vapor correction assessment over time for CO. HB stands for humidity bench while Droplet is for the droplet test described in the text. The humidity bench allows a much more controlled and precise sensitivity test. The two sites droplets show a large drift of the water vapor correction compared to the initial tests.

*Code and data availability.* Data from the stations are available on the CarbonPortal, data from the initial tests and code are available on demand.

*Author contributions.* C Yver-Kwok wrote the manuscript and code to plot some of the figures. The other authors reviewed and improved the manuscript and as PI's produced the high-quality data presented here.

5  *Competing interests.* The authors declare no competing interests.

*Acknowledgements.* The French monitoring network acknowledges the long term support received as part of the Service National d'Observation program and thanks its technical staff. I.M. and J.L. thanks the SMR technical staff and funding from ICOS-Finland (281255). This work was supported by the Ministry of Education, Youth and Sports of CR within the CzeCOS program, grant number LM201812. ICOS-Switzerland is funded by the Swiss National Science Foundation (Phase I (2013-2017): 20FI21_148992; Phase II (2017-2021): 20FI20_173691) and

5   in-house contributions. ICOS Netherlands is substantially supported by the Dutch Research Council NWO through the Ruisdael large scale infrastructure project. ICOS labelling activities at CMN were started under the Project of National Interest NEXDATA funded by Italian Ministry for Education, University and Research (MIUR). CNR gratefully acknowledges the hospitality and the logistic support at CMN provided by CAMM - Italian Air Force. The current ICOS activity at CMN is supported by the JRU ICOS.Italia. The ICOS station Observatoire de l'Atmosphère du Maïdo (RUN) is a Belgian – France collaboration project and operated through a collaboration between the Royal

10  Belgian Institute for Space Aeronomy (BIRA-IASB) and the following French parthers: Commissiarat à l'Energie Atomique et aux Energies Alternatives (CEA), Centre National de la Recherche Scientifique(CNRS), Université de Versailes Saint Quentin-en-Yvelines (UVSQ), Université de La Réunion (UR). In Belgium it is financially supported since 2014 by the EU project ICOS-Inwire and the ministerial decree for ICOS (FR/35/IC1 to FR/35/C5). The authors are grateful to their colleagues M. De Mazière and C. Hermans (BIRA-IASB) and J.-M. Metzger (UR) for their contributions to the labelling process, daily operations and management of the station.

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
