# Peer review of "Evaluation and optimization of ICOS atmosphere station data as part of the labeling process"

_Atmospheric Measurement Techniques, 2020_

## Referee Comment (RC1) · Anonymous Referee #1 · 21 Aug 2020

This paper represents a substantial contribution to scientific progress in atmospheric measurement techniques of greenhouse gases. It describes in detail the labelling process that atmospheric greenhouse gas measurements sites have to go through to receive approval to join ICOS, as well as the quality controls used to assure high quality and precise scientific data. To date, there are few papers outlining the steps taken to harmonise trace gas measurements from large networks with multiple stakeholders. This paper bridges this gap, gives good lessons learnt during the process and provides great clarity on the ICOS atmospheric station labelling procedure that can sometimes be a little opaque to those outside of ICOS.

[Figure]

The paper is well written, with very good English and the scientific methods used are appropriate. The figures and tables are on the whole well-presented and add to the understanding of the manuscript. I thoroughly enjoyed reading and reviewing the manuscript.

Specific comments: There are a few inconsistencies within the manuscript that need to be addressed, including the mixing of units (e.g. ppm and $\mu$mol.mol-1) and whether species are written with subscripts or not (i.e. CO2 or CO2) within both the manuscript, tables and figures. Furthermore, there are a number of times where the authors have used atmospheric or atmosphere interchangeably to describe ICOS stations where atmospheric GHGs are measured (c.f. P3 L23 and P15 L20).

In addition, the use of abbreviations and their introduction into the manuscript aren't always consistent. For example, the abbreviation of greenhouse gas to GHG is used within the abstract whilst no abbreviation is used within the remaining sections of the manuscript. In contrast, station acronyms based on GAWSIS IDs for stations have been used within the manuscript but no full name has been given the first time that the station has been introduced in the text. For clarity, please include the full name, followed by an acronym, the first time it is used in the manuscript.

Furthermore, there are some formatting issues with some of the tables and figures that need attending to, such as the inconsistencies of table presentation and the boarders of column titles. Some figure captions are lacking in basic information (e.g. what abbreviations mean within the figure, keys for schematics, etc.) and mean that if a reader were to come and look at the figure only, it is difficult to comprehend (see technical corrections for specific figure comments). Additionally, some colour combinations are hard for people with colour-blindness to decipher. Please consider altering some combinations, e.g. not having red and green together.

Technical corrections: P1 L8: text does not read well, try – "... calibration gases are measured twice a month". P1 L8: change "controlled" to "verified". P2 L3: text does not

read well, try – "... in a calibration sequence is possible, saving gas and extending the calibration gas lifespan". P2 L10: text does not flow well, try – " Continuous, precise, greenhouse gas monitoring began in 1957 at the South Pole and in ...". P2 L21: remove "i.e." before "harmonized and high precision...". P3 L1: seeing as the location of the ATC is stated in P3 L6, this should also be done for the Flask and Calibration Laboratory and the Central Radiocarbon Laboratory. P3 L6-8: the location of the ATC is ambiguous, would be better to state outright that it is in France, i.e. "The Atmospheric Thematic Center (ATC, https://icos-atc.lsce.ipsl/fr/) is divided into three components; the metrology laboratory (MLab) responsible for instrument evaluation, protocol definition and PI support (located in France), the data centre (located in France) responsible for data processing, code development and graphical tools for PIs, ...". P3 L10: add "the" before ATC. P3 L15: it might be good to give an example of what elaborated products are available as those reading the manuscript who are not familiar with ICOS may not know what is offered. P3 L28-29: text does not read well, try - "... adhering to the WMO guidelines (WMO, 2018) for greenhouse gas observations but are elaborated on more in ATC and Laurent (2017) and presented in section 2.4. P4 L15: What is the frequency of routine data evaluation sessions? P4 L26-27: It is not clear here what it meant by the list of Class 1 parameters that aren't necessary for a station to be labelled. Is it that out of the three parameters (boundary layer height, GHGs and 14C values from flasks), two do not need to yet be in place for a site to be labelled; or should the sentence read three parameters? Additionally, could more clarity be given on what is meant by GHGs, seeing as in situ CO2 and CH4 measurements are mandatory and are GHGs. P5 L1: Please provide a reference for the specification document if publicly available. In addition, is the list of accepted analysers publicly available? P5 L15-18: the relative pronoun "whose" on L16, in the parentheses, should be replaced with "for which". P5 L25: Try to avoid double parentheses. P6 L6: What mixing ratios do you suggest for the long-term target to ensure that it is representative for more than 10 years? P6 L7-8: For clarity, change to "It is recommended to send the long-term target, as well as the calibration set, for recalibration approximately every 3 years to

the CAL-FCL to investigate and take into account any possible composition changes in the gases, especially for CO. P6 L10: To make the manuscript more accessible for colour-blind readers, change the end of the line to "The instrument calibration dates are included at the bottom of the plot by the open orange circles.". P6 L15: replace "depending" with "dependant". P6 L19: Include definite articles (the) in front of long-term and short-term target. P6 L22: replace "are going" with "go". P6 L23: To remove potential misunderstandings with the meaning of the word "control" in relation to station PI's reviewing data, replace "control" with "verify" or "review". P6 L28: Please state which server you are referring to at the beginning of the line. P6 L28: Where are the ATC data products located that station PIs check? Are they the plots shown on the ATC website (e.g. https://icos-atc.lsce.ipsl.fr/SAC)? If so, please state that these are publicly available for stations that have been labelled and give the link to the website. P6 L30: Please outline what is used as the flagging scheme. Is this already published in Hazan et al. (2016)? P6 L31: As for P6 L23, change "controlled" to "verified" or "reviewed" to remove misunderstandings. P7 L22-25: Establishing if your sample intake line has a leak or not is an important consideration. What is done at sites to establish if there are leaks if the mast cannot be climbed easily due to reasons outside of the control of the PIs and there is no infrastructure installed to alter intake configurations remotely? P8 L1-2: More information is needed for this test to be replicated by someone reading this paper and is not part of ICOS. It is not clear to me if a humidification loop is used, as in Stavert et al. (2019; https://doi.org/10.5194/amt-12-4495-2019) to conduct this test or is another method used within ICOS? In addition, what mixing ratios within the cylinder are suggested by ICOS for the test? P8 L9: Please provide a hyperlink for the automatically generated plots on the ATC website. P8 L14: Please alter "bi-monthly" to "twice-monthly" or "every 15 days" to remove ambiguity. P8 L30: Please include three letter identifier after Lindenberg to aid the reader in finding the example in Figure 4. P9 L8-9: Please include a space between numbers and units. P9 L18: Calibration drift is misleading as a title, as it suggests the drift of mole fractions within the calibration cylinders. This may be the case for some species and over longer periods of time

than 2 weeks; however, the authors are discussing the assessment of instrumental drift in this section. I would suggest changing the sub-heading to "Instrumental drift and calibration optimization". P9 L20: Is instrumental drift ever estimated over shorter timescales than 2 weeks? In my personal experience with optical instruments, some can drift quite significantly over periods of time less than 1 day. This is especially the case in environments where there is poor thermal stability (e.g. no air conditioning or too large thermal switches on air conditioning units). P9 L30: Are there specific types of analysers that are worse than others for temperature dependence, e.g. instruments with larger cavities? P10 L9-10: Please state what the meteorological sensors need to be complaint against? ATC specifications document? Also, are the meteorological data submitted to the same database as the GHG data? P11 L16: Replace "Belgium and France" with "Belgian and French". P11 L28-30: This sentence isn't clear and I cannot decipher what the authors mean. Please rephrase to give clarity. P11 L32-33: Has a study ever been done by ICOS on the representativeness of using buffer volumes? As I understand it, one of the main advantages of using optical instruments is the added information that can be gained from the short-term variability in mole fractions. Would it be better to not use buffer volumes to futureproof the data at these sites for when numerical models can ingest high frequency data and simply smooth data based on statistical filters, as used at other sites and networks? P13 L11-13: What were the sources of leaks for GAT and STE? It is not often that papers include lessons learned information, which is often very useful to other stations in diagnosing problems of their own. P13 L19: As American English has been used throughout the manuscript, please remove the th from after 15 (both occasions) as this is only included in British English. P14 L30: As far as I am aware, there are two methods for a Nafion counter purge: using a dry gas, such as zero air or N2, or the Welp et al. (2013; 10.5194/amt-6-1217-2013) method of reflux mode (i.e. taking a small portion of dried air post-Nafion and using it as the counter purge gas but ensuring it is at a lower partial pressure than the sample gas). Which method is currently recommended by ICOS when using a Nafion to dry samples? P15 L4: Full name used for Jungfraujoch instead of a trigram. Please

replace with JFJ to be consistent. P15 L20: Grammatical error, please change to – "In this paper we have presented the process used to label ICOS atmospheric stations.". P15 L28: Replace "oftentimes" with "often".

Table 1: Please include a reference to the ICOS atmospheric station specification document in the caption. In the table, it is not clear that the column "Gases, periodical" relates to flask measurements. Please add in more information in the column caption or in the table caption to clarify this. In addition, was atmospheric pressure at the highest inlet height ever discussed as a useful parameter for modellers to see the pressure differences between the top and bottom inlets of towers?

Table 2: I am assuming that the thresholds cited in this table are specific to a certain type of instrument, i.e. CRDS seeing as these instruments are mostly used within the network for $CO_2$ and $CH_4$ analysis. In addition, please could you include the $H_2O$ threshold used for calibration gases, as referred to on P5 L25-26.

Table 3: Please can you ensure that the writing of magl is consistent with figures (often written as m agl). This also applied to masl.

Table 4: The ambient air percentage for KRE looks erroneous (5069 %).

All figures: Please include lists of full site names, with acronyms in parentheses, of any site included in the figure. In addition, if any instrument number is included or a cylinder D number, please give an explanation. If species are mentioned, ensure that any numbers are subscripted, e.g. $CO_2$, not CO2.

Figure 1 caption: Suggest changing to = " One month of target gas injections for $CO_2$ (ppm), shown as the difference of calculated vs. assigned mixing ratios. Short term target data is plotted in green, whilst long term data is in brown. The calibration dates are shown by the light orange open circles. Cylinder number (D\*\*\*\*\*\*), mean values ($\pm$ X), point-to-point variability (Ptp) and difference to the assigned value (Diff) are displayed above the figure."

Figure 2 caption: Notwithstanding that the figure is there to show locations to connect a cylinder to run the leak tests outlined in section 2.3.3., a key for the different parts shown in the figure would be useful for reader comprehension.

Figure 3 caption: Please indicate what the values in the 2nd and 3rd columns represent.

Figure 4 caption: Please indicate why there are duplications of LIN (as explained in the manuscript text).

Figure 6 caption: please indicate how the deviation of measurement from assigned values is calculated.

Figure 12 caption: The MHD Marine Smooth Curve isn't explained, please include a brief description of the methods used to derive the curve.

Figure 13: To make the insert clearer, consider adding a black border around to isolate it from the main figure. In addition, add in a list of the sites, with the acronyms in parentheses, as well as giving the years for the colours of when sites were first labelled.

Figure 15: It is currently hard to read the site acronyms and instrument numbers, please reduce the text size slightly so that there are some gaps between each site, like in Figure 16.

Figure 17: There is no bottom tine to the top plot.

Figure 20: What is the difference between ATC MetroLab HB 2016-05 and ATC MLab Droplet?
* * *

---

## Referee Comment (RC2) · Anonymous Referee #2 · 29 Aug 2020

Accurate measurements of greenhouse gas (GHG) concentrations in the atmosphere is the first step to understanding and mitigating the impacts of climate change, and a full assessment of the temporal and spatial variabilities and trends in these concentrations requires a wide network of observations. The GHG measurement community has worked hard to ensure that the GHG measurements operated around the world are "compatible" (data meet quality standards necessary for addressing the scientific questions and can be compared under common calibration scales) through the efforts such as the recommendations of the WMO/IAEA GGMT meetings (most recent report at; https://library.wmo.int/index.php?lvl=notice_display&id=21758#.Xzxuv1QzaUk). But more focused efforts such as the Integrated Carbon Observation System (ICOS) present an opportunity to improve on these practices and derive a more coherent dataset required for the regional focus of such projects. As such, this current work, documenting the key aspects of the ICOS station labeling process and data quality control, is a significant contribution to the field, and suitable for publication in this journal.

Overall, the manuscript is well-written and major revisions aren't necessary. However, I would like to make some suggestions that, when properly addressed, I feel the community will benefit in knowing, especially given the extensive and high-quality dataset that this work can derive it's conclusions from.

[P5L13-29] Cylinder calibration requirements
I feel that the authors can further elaborate on the following questions.
- Line 14: Can the authors specify what range of calibration frequencies are typically tested in the initial test period?
- Are there any requirements for the order in which the 3-4 cylinders are measured in a sequence?
- Line 18: I would prefer that the authors clarify that the 30 minutes is the run time per cylinder, even if that becomes implied when reading through other parts of the text. Similarly, I think it's worth further clarifying that the calibration cylinders will be run in series, leading to an estimated length of some continuous hours of calibration per sequence.
- When stating the total length of the calibration sequence, I feel it would be more coherent if a short reference is made to the discussions in 2.3.1 regarding the how the instrument will dry out during the long calibration sequence.
- Figure 2 indicates a REF tank which isn't explained in the text. Is this part of the ICOS requirements, and if so, can an explanation be provided for the role of the REF tank?
- Figure 2 suggests that all the samples go through the selection valve and straight into the instrument, which means that the inlet pressure will likely vary between samples, for example the air inlets will likely be slightly sub-ambient, while the calibration cylinders will be dependent on the regulator settings for each tank, and this design is dependent on the instrument's internal cell pressure control to compensate accordingly. Is this correct? Is there any guidance on the regulator pressure settings?
- Regarding regulators, I feel the community would greatly benefit from a list of recommended regulator models, similarly to how the the authors have stated a specific recommendation for the sample switching rotary valve later in the text.

[P6L9-20] Moisture effects on the short-term target measurements
- The effect of moisture on the short-term target is very interesting. How long does it take for the moisture level to return to steady state once the air measurements resume, and is there an attempt to flag out the air measurements while the moisture levels are still low?
- Also, what are the implications of calibrating the instrument in a dry state while air measurements are done wet, especially in cases where a dryer is not used? Does what's shown for the short-term target measurements indicate there could there be a (small) bias between the calibration and air measurements?

[P7L15-27] Inlet line tests
- Just making sure I understand this process correctly: For the instrument test, do I understand correctly that there's an electronically actuated 12-port switching valve, and that one of the ports (#12 in this case) is being left open to facilitate direct measurement of the test tank? Is this port normally plugged?
- For the shelter test, I presume the downstream overflow pump for the inlet line is kept on, then the pressure of the test tank connected to the inlet line is adjusted to the point that the slight sub-ambient pressures in the inlet lines are reproduced? If these presumptions are correct, I would advise that these details be included in the manuscript.

[P8L12-24] PI data validation
- Can the authors clarify the temporal resolution in which the PI's perform data validation? I presume either raw or 1-min level?
- What are the criteria that go into validating the data at hourly timescales as shown in Figure 3? For example, is there a % or minimum data amount requirement to flagging the hourly data as valid/invalid?

[P9L19-28] Calibration drift due to regulator effects
- I find Figure 5 fascinating, in terms of understanding regulator contamination effects. As expected, the short-term target, which is measured more frequently, shows faster stabilization time. On the other hand, for the long-term target and calibration tanks, all measured at the same 15-day cycles (as far as I understand), regulator flushing problems are much more severe in the the long-term target. My guess is that the calibration runs are an average of the 3~4 cycles, hence the variability is substantially reduced by the 2nd~4th runs. Is this correct? If so, comparing the first calibration runs against the long-term target run may provide a more meaningful comparison of regulator contamination effects. Also, it would be interesting to know if stability is reached much more quickly in the 2nd~4th calibration runs.
- Do all the cylinders use the same model of regulators? If not, could that be a consideration in interpreting the results shown in Figure 5?

- I might expect that each of the calibration tanks would show a somewhat different contamination effect based on its concentration, but have the authors looked as this?
- I find it interesting that each compound shows a different pattern in reaching steady state. Looking at the long-term target where the signals are most amplified, $CO_2$ is initially very low, $CH_4$ starts low and slightly overshoots prior to reaching steady state, while CO is initially very high. Do the authors have an explanation for these results?
- Some of these discussions may be more suitable in section 4.2.

[P9L19-28] Calibration drift
- Have the authors estimated whether the instrument drift is "linear"? I.e. When looking at the drift rates of the 4 calibration cylinders at different concentrations, is there indication that the drift rate is similar/different at each of the concentrations? Looking at Figure 6, it certainly seems like the 4 tanks drift at a similar rate, but it's hard to be quantitive. I would love to see an assessment of instrument linearity changes quantified and compared across instruments, and the vast dataset accumulated in this ICOS experiment would be an excellent for this.

[P19L7-12] Meteorological measurements
- Can the authors specify what checks the ATC performs on the meteorological measurements? My understanding is that verifying the instrumentation and checking the accuracy of the meteorological data is quite challenging, and it would be interesting to know the ATC's specific procedures.

[P10L22] Alignment of timestamps
- One thing that hasn't been mentioned here is whether there are procedures to ensure that the timestamps of the various data streams on site are matched. For example, when combining the data stream from the CRDS and the meteorology measurements, is there a process to ensure that the timestamps for these two data streams are matched? Or similarly in case there are multiple CRDS instruments on site?

[P12L29] Section on tank stabilization time
- Again, can the authors confirm that the stabilization time for the calibration combines the data for all 3~4 cycles? I would imagine that the 1st cycle shows the longest stabilization time, while the others would be relatively short, and averaging these together would lead to a relatively short stabilization time. In fact, I wonder if the 1-min stabilization time mentioned in page 13 line1 refers to this special case of 2nd~4th cycles in a calibration run?
- Please refer to questions above on "Calibration drift due to regulator effects" on some questions I hope the authors can address, perhaps in this section.
- It is interesting that looking at $CH_4$ in Figure 15, the short-term target stabilization times are higher than the long-term target stabilization times at many sites. I'm a little

surprised by this since the short-term target regulator would be flushed more frequently and hence I would expect it to show less regulator artifacts. Do the authors have an explanation for this?

Minor comments (Denoted by Page# and Line#)

P2L10-19 Introductory paragraph: This introductory paragraph is a summary of the long history of greenhouse gas measurements, which understandably is not an easy task. Here are some thoughts:
- "Continuous"?: First, it is somewhat vague what "continuous" refers to in this case. I presume the authors mean measurements of very high frequency, but technically speaking, no measurement is truly "continuous". While I understand the authors' emphasis on measurements made at high-frequency, I do feel that the significant scientific contributions from flask based measurements of GHGs deserves to be acknowledged. Also, I would note that the AGAGE measurements (referenced in the Prinn et al.) are GC based and I'm not sure that one would describe them as "continuous" (although I believe some prior publications have described the measurements as "quasi-continuous").
- Also, I find it odd that the more recent megacities efforts are not referenced here, nor are any of the flux measurement networks, as both are high-frequency measurements, and the authors note the importance of networks to address regional and local fluxes (line 17).
- Given these comments, I would ask that the authors reconsider the framing of this introductory sentence, and rewrite as necessary.
- Line 13, Prinn et al. 2000: Please update to Prinn et al 2018. Prinn, R., Weiss, R., Arduini, J., Arnold, T., DeWitt, H., Fraser, P., Ganesan, A., Gasore, J., Harth, C., Hermansen, O., Kim, J., Krummel, P., Li, S., Loh, Z., Lunder, C., Maione, M., Manning, A., Miller, B., Mitrevski, B., Mühle, J., O'doherty, S., Park, S., Reimann, S., Rigby, M., Saito, T., Salameh, P., Schmidt, R., Simmonds, P., Steele, L., Vollmer, M., Wang, R., Yao, B., Yokouchi, Y., Young, D., Zhou, L. (2018). History of chemically and radiatively important atmospheric gases from the Advanced Global Atmospheric Gases Experiment (AGAGE) Earth System Science Data 10(2), 985 - 1018. https://dx.doi.org/10.5194/essd-10-985-2018

P3L8: Does the "data center" have any special acronym like MLab or MobileLab? At first reading, the data center didn't seem distinguished enough as the second component of ATC.

P3L11 "ATC and Laurent (2017)": The notation of this reference seems a bit strange to me. Looking at the original document, I see that it is edited by Laurent with many contributors listed. This metadata page (https://meta.icos-cp.eu/objects/_fDB4nDzrYYG9uu6fPsvfiG9) asks that the citation be for "Laurent, O., ICOS Atmosphere Monitoring Station Assembly and ICOS Atmosphere Thematic Centre (ATC)". My suggestion would be for "Laurent et al." or "ATC" with Laurent noted as an

editor in the reference, but I ask the authors to confirm the appropriate way to reference this report. Also note that the full reference for this item seems to include some typographical errors.

P3L13 "Downstream": This usage seems a little awkward, as I would associate the term with a physical connected process (as in instrumentation), and not for a separate task carried out at a later point in time. Perhaps something like "Afterwards" or "Once the labeling process has been approved,"?

P3L30-33: Should the first word after the ":"s be capitalized?

P4L4 "exchanges first with" -> "first contacts"?

P4L6 "in a limited number" -> "a limited number of"?

P4L10 "sites" -> "site"?

P4L15 "thanks to" -> "through"?

P4L33 "class 1" -> "Class 1"?

P5L14 "a lot of" -> "more frequent"?

P6L15 "depending" -> "dependent"?

P6L15 "really important" -> "important"?

P6L18-20: "If the instrument variability should be assessed with…": I'm not exactly sure what the authors refer to in this sentence, please clarify.

P6L22 "are going through" -> "go through"?

P6L22 "different criteria" -> "various criteria"?

P7L1-5: Capitalize each "to"?

P8L6 "are meeting" -> "meet"?

P8L27 "evaluating that" -> "evaluating whether"

P10L16 "flow rates": Can the authors clarify what specific flow rate is being referenced here, and where this flow rate measured?

P11L20 "who" -> "which"?

P11L28-30 "The other ten use either solenoid values either for…" : The meaning of this sentence seems unclear, with the two "either"s. Please revise.

P12L2 "experience often" -> "often experience"?

P12L2 "do no use" -> "do not use"

P13L9 "was concerned" -> "was of concern"

P13L9 "as said in" -> "as discussed in"

P13L9-10 "GAT CRDS lines showed a long stabilization time" -> "Long stabilization times were found for the GAT CRDS lines"?

P14L21-22 "This can be due to … analyzer inlet,…" -> "This can be due to factors such as … and the filter at the analyzer inlet."

P14L30 "either a cryogenic" -> "or a cryogenic"?

P15L11: Use of acronyms vs full names for the sites?

P15L16 "no CO bias was observed anymore" -> "CO bias was no longer observed"?

P15L25: Should the first letter of each bullet item be capitalized?

Table 1. Based on P4L27, is the "*" in [Class 1 ; Gases, periodical] applied to the wrong item? I.e., shouldn't the $^{14}C$ item be starred?

Table 3. Coordinates: I would find it helpful if the lat/lon coordinates could be given with more significant digits to allow for pin-pointing site locations in interactive mapping tools such as Google Earth.

Table 4. KRE Ambient air (%) is 5069?

All figures: Some figures missing specific description of how the error bars were calculated. Also, please note the differing concentration units in the plots/text (ex. ppm vs $\mu$mol/mol), and any missing subscripts in $CO_2$ and $CH_4$.

Figure 2. Perhaps arrows would be more apt in identifying the specific injection points? Also, the plot is difficult to interpret without legends for the symbols.

Figure 15. Can the font size of the x-axis labels be adjusted for better legibility?

Figure 16-18: It seems like the bottom Bias plot in each figure has 26 ticks on the x-axis, while the CMR/LTR plots above have 27 ticks. Is there a station/instrument missing in the bottom Bias plot? Also, the pink/blue boxes are very hard to distinguish for many of the box plots, is there a way to improve legibility?

---

## Referee Comment (RC3) · Anonymous Referee #3 · 9 Sep 2020

This work describes the specific workflow and quality assurance processes within the ICOS network. The authors did a wonderful job composing a well written, comprehensive paper about the difficulties and challenges faced in the world of high-quality greenhouse gas observations. I think this work is a substantial contribution to the knowledge base and scientifically very important. Everyone in the field relies on inter-comparable high-quality long-term observations but is rare to see papers that clearly describe and outline the hard work that lies behind them. I especially appreciate that the examples given in the paper include times where problems were found and solved, as well the realistic description of timeframes when setting up a new site. It is the lived reality in the field and will hopefully be educational to both users of the ICOS data as

well as groups interested in setting up long-term observation stations. While the quality of the paper is generally high, there are some inconsistencies in the use of language mainly the interchangeable use of site names and abbreviations. Some of the parameters mentioned in the text and/or in figures have no clear explanation of how they were derived. The figures contain a lot of vocabulary and abbreviations that are probably useful within ICOS but can be confusing for the casual reader (tank and instrument numbers) without additional explanation. A lot of the figures also use very small fonts that can only be read by zooming and will not be legible in print outs at all. The colour in the figures is generally not suitable for colour-blind individuals (a lot of red and green with the same saturation levels). The descriptions of the figures in both the text and figure legends are very perfunctory even for some of the more complex figures

Specific Comments: While the introduction gives a brief overview of greenhouse gas measurements and observation networks what the data is used for but lacks information about why we need such high-quality greenhouse gas observations and the benefits of the labelling process for the end-user of the data product.

In 2.4 general requirements and table 1, the different parameters mandatory and recommended for the different classes of station are given but not much information about why they were deemed mandatory or recommended. Two parameters in table 1, the mandatory atmospheric pressure observation at ground level and the recommended eddy covariance flux for CO2 are not mentioned in the text at all.

In 2.2 greenhouse gas calibration requirement: Add an example or range for the automatically filtered data.

2.3.3 Intake line and water vapour correction tests: The text is a bit unclear, it mentions a shelter test every 6 months and testing the outside lines every year, then later in the text it mentions testing of the whole line is recommended at sites where lines are older than 10 years. Is the yearly testing of the outside line just to tests from the base of the tower to the shelter or is the yearly testing of the whole outside line only recommended

for lines older than 10 years? I assume that since ICOS focusses on tall tower sites, yearly full line tests would be very expensive (hiring climbers to climb the tower twice for each test to connect and disconnect the lines).

2.4.4 Calibration drift and optimization: In this section and in figure 6 the word calibration drift is used for the instrument drift that is then corrected and optimized with the calibration, this is very confusing as there is also such a thing as a drift in the concentration of calibration gases.

2.4.7 Diagnostic parameters, Page 10, line 7: A low flow rate within the line could also be indicative of an obstruction in the line (damage to sampling line or blockage).

3. Presentation of the 23 labelled stations Page 11 line33: The paper generally describes all the site setups in great detail, but there is no description of the 4 sites with buffer volumes. I appreciate that everyone in the community has strong opinions about the usage of buffer volumes, but regardless of their merits or lack thereof, a more detailed description of the buffer volume setups (integration volume, flow rate and in integration time) would be helpful.

4.1 Calibrations: Is the drift described this chapter and in table 5 the same instrument response drift described in 2.4.4 and figure 6? The naming is ambiguous as it implies it is the calibration that is drifting not the instrument response. There is also no description of how the drift was calculated.

4.3 Uncertainties: Figures 16-18 contain a lot of information, while some of it is explained here in detail, other artefacts are left unmentioned or are mentioned later in the text and then not referenced. For example in Figure 16: For JFJ, The continuous instrument repeatability (CMR) is good but the long-term repeatability (LTR) is high, later in paragraph

4.6 it is explained that this is was due to a polytube Nafion, but the text does not reference figure 16. Then there is OPE that seems to have had a bias in the target

value for both CH4 and CO2 for a while. On page13, line 21 it is mentioned that two sites show values outside of the WMO targets for CO, but it is not specified which of the subplots this refers to. 3 additional sites show larger biases after the initial test period (IPR, SMR and NOR) but they are not mentioned in this section. Later in the troubleshooting section, there is a mention of the issue with CO that was related with the use of heated inlet cups but figure 18 is not mentioned in that section and the sites are referred to with their full name.

Figure 11: This figure is not easy to read and within the text, it is just casually referred too, does it add any value? How does it help evaluate the influence of different sources ( is it because it shows the different inlet heights?) whatever information it is supposed to convey is lost in the sheer amount of data (1 year 4 heights 3 compounds, plus quality assurance subplots). I could see the value of a plot like this online where you can zoom in to it. The short-term long-term target stability on the right-hand side is interesting but is not even mentioned in the text or legend of the figure and the short and long-term targets are also shown in figure 1.

Technical comments: Page 3, line 11: First mention of WMO compatibility goals but no information what they are for the gases discussed in the paper.

Page 5, line 1: No reference for the ICOS specification document.

Page 5, line 25 mentions that Table 2. Contains the raw minute and cycle but then Table 2 contains the minute, injection, and cycle data. Is an injection not the same as a cycle? The words are interchangeably used throughout the paper for example in figure 7.

Page 7 line 22: What is meant with the intrinsic bias of the instrument?

Page 7, line 31: Rephrase to clarify that the onsite water test needs to be performed if the last instrument test at MLab was longer than a year ago.

Page 10, line 20-21: Clarify that the instrument flow rate can be used to estimate the

lifetime of cylinders.

Page 11, line 1: LTR, is defined as the Long-Term Repeatability which is I understand to be the 3 day average of the standard deviation of the short-term target measurement. The text does not specify that it is based on the short-term target although it must be as the long-term target would not be measured often enough for this. The naming is confusing as the short-term target makes up the long-term repeatability but then there is also a long-term target.

Page 13, line 15: Figure 12, not 13 describes the bias calculation, the text would be easier to follow it the calculation was described within the text of the chapter and or the relevant figure description.

Page 13, line 18 change to: The red dot.

Page 13, line 29-Page 14, line 4: Restructure this part to make it easier to read. The text references that the description of figure 19 contains how the bias was calculated but figure 19 contains no calculations. Later in the paragraph in page 14, line 4 it is clarified that the bias is (measured-reference) I assume what is described as the bias is just the difference between the concentration measured in the line vs. directly in the tank but the roundabout description makes it harder to understand than necessary. Maybe just rephrase the second sentence to make it clear it is measured - reference as currently, it is the other way around.

Page 14, line 13: Reference water droplet test protocol used.

Page 14, line 30: If available add part numbers and manufacturer for both the recommended Nafion membrane and cryogenic water trap.

Table 1: is missing a reference to the ICOS specification document.

Table 4: 5069% in KRE?.

Figure 2: No labelling and description of parts and existing text font size too small.
* * *
Interactive
comment

Figure 11: The colour choice is not ideal (yellow on white background, and red and green lines and circles for both the target values). The numbers above the target plots are not explained anywhere, one I assume is the targets assigned concentration but Ptp is not explained anywhere. For the $H_2O$ % the target numbers are also present but empty (0.00 for all) please remove.

Figure 15-19: It is hard to visually align the legend at the bottom with the two figures above, especially in figure 16-18 where there are 2 box and whisker plots for each instrument. Adding the legend to the other 2 plots or some kind of shading or line might help.

Figure 16: Has some random signs at below the figure legend # # \.

---

## Author Comment (AC1) · 2 Oct 2020

**Author comment on "Evaluation and optimization of ICOS atmospheric station data as part of the labeling process" by Camille Yver-Kwok et al.**

**Anonymous Referee #1**

*We thank the reviewer for their positive review and their constructive and helpful comments. We answer them below, highlighted in bold and italic.*

This paper represents a substantial contribution to scientific progress in atmospheric measurement techniques of greenhouse gases. It describes in detail the labelling process that atmospheric greenhouse gas measurements sites have to go through to receive approval to join ICOS, as well as the quality controls used to assure high quality and precise scientific data. To date, there are few papers outlining the steps taken to harmonise trace gas measurements from large networks with multiple stakeholders.
This paper bridges this gap, gives good lessons learnt during the process and provides great clarity on the ICOS atmospheric station labelling procedure that can sometimes be a little opaque to those outside of ICOS.

The paper is well written, with very good English and the scientific methods used are appropriate. The figures and tables are on the whole well-presented and add to the understanding of the manuscript. I thoroughly enjoyed reading and reviewing the manuscript.

Specific comments: There are a few inconsistencies within the manuscript that need to be addressed, including the mixing of units (e.g. ppm and µmol.mol-1) and whether species are written with subscripts or not (i.e. CO2 or CO2) within both the manuscript, tables and figures. Furthermore, there are a number of times where the authors have used atmospheric or atmosphere interchangeably to describe ICOS stations where atmospheric GHGs are measured (c.f. P3 L23 and P15 L20).
In addition, the use of abbreviations and their introduction into the manuscript aren't always consistent. For example, the abbreviation of greenhouse gas to GHG is used within the abstract whilst no abbreviation is used within the remaining sections of the manuscript. In contrast, station acronyms based on GAWSIS IDs for stations have been used within the manuscript but no full name has been given the first time that the station has been introduced in the text. For clarity, please include the full name, followed by an acronym, the first time it is used in the manuscript.
*We will pay attention to this and correct accordingly.*

Furthermore, there are some formatting issues with some of the tables and figures that need attending to, such as the inconsistencies of table presentation and the boarders of column titles. Some figure captions are lacking in basic information (e.g. what abbreviations mean within the figure, keys for schematics, etc.) and mean that if a reader were to come and look at the figure only, it is difficult to comprehend (see technical corrections for specific figure comments). Additionally, some colour combinations are hard for people with colour-blindness to decipher. Please consider altering some combinations, e.g. not having red and green together.
*We will work on the captions to help the tables and figures be understandable as standalone. When possible, we will change the colors to address color-blindness. However the figures in the first part were made for the Step2 reports and are not easily reproduced. Thanks to your remark, we are working to improve the color code for the figures that are produced daily and are visible on the ATC website.*

*For the technical corrections, we will apply them. Specific answers can be find below.*
Technical corrections:

P1 L8: text does not read well, try – ". . . calibration gases are
measured twice a month".
P1 L8: change "controlled" to "verified".
P2 L3: text does not read well, try – ". . . in a calibration sequence is possible, saving gas and
extending the calibration gas lifespan".
P2 L10: text does not flow well, try – " Continuous, precise, greenhouse gas monitoring began in
1957 at the South Pole and in . . .".
P2 L21: remove "i.e." before "harmonized and high precision. . .".
P3 L1: seeing as the location of the ATC is stated in P3 L6, this should also be done for the Flask
and Calibration Laboratory and the Central Radiocarbon Laboratory.
P3 L6-8: the location of the ATC is ambiguous, would be better to state outright that it is in France,
i.e. "The Atmospheric Thematic Center (ATC, https://icos-atc.lsce.ipsl/fr/) is divided into three
components; the metrology laboratory (MLab) responsible for instrument evaluation, protocol
definition and PI support (located in France), the data centre (located in France) responsible
for data processing, code development and graphical tools for PIs, . . .".
***It is in fact located in France for the Mlab and data center and in Finland for the MobileLab. We
will reformulate for accuracy.***

P3 L10: add "the" before ATC.
P3 L15: it might be good to give an example of what elaborated products are available as those
reading the manuscript who are not familiar with ICOS may not know what is offered.
P3 L28-29: text does not read well, try - ". . . adhering to the WMO guidelines (WMO, 2018) for
greenhouse gas observations but are elaborated on more in ATC and Laurent (2017) and presented
in section 2.4.
P4 L15: What is the frequency of routine data evaluation sessions?
***We usually go for once a month.***

P4 L26-27: It is not clear here what it meant by the list of Class 1 parameters that aren't necessary
for a station to be labelled. Is it that out of the three parameters (boundary layer height, GHGs and
14C values from flasks), two do not need to yet be in place for a site to be labelled; or should the
sentence read three parameters? Additionally, could more clarity be given on what is meant by
GHGs, seeing as in situ CO2 and CH4 measurements are mandatory and are GHGs.
***We will clarify. There are only two parameters: BLH and flask measurements. For the flask we
measure GHG (the one we sample quasi-continuously and some we don't such as $SF_6$ and $H_2$)
and radiocarbon. For the flasks, there were delay with the development of the automatic flask
sampler.***

P5 L1: Please provide a reference for the specification document if publicly available. In addition,
is the list of accepted analysers publicly available?
***The list is in the specification document which is publicly available (ICOS RI (2020): ICOS
Atmosphere Station Specifications V2.0 (editor: O. Laurent). ICOS ERIC.
https://doi.org/10.18160/GK28-2188)***

P5 L15-18: the relative pronoun "whose" on L16, in the parentheses, should be replaced with "for
which".
P5 L25: Try to avoid double parentheses.
P6 L6: What mixing ratios do you suggest for the long-term target to ensure that it is representative
for more than 10 years?
***In the specifications, the recommended long-term target mixing ratio is 450ppm for $CO_2$ for
background stations, 470 ppm for peri-urban sites. You can find the values for the other species
in the specification document. Of course this value will pass from a relatively high to a low value
over time compared to most of the ambient air measurements. However, since this tank is really***

*intended to check the long term consistency of the measurements we could not find a better option to cope with the trend.*

P6 L7-8: For clarity, change to "It is recommended to send the long-term target, as well as the calibration set, for recalibration approximately every 3 years to the CAL-FCL to investigate and take into account any possible composition changes in the gases, especially for CO.

P6 L10: To make the manuscript more accessible for colour-blind readers, change the end of the line to "The instrument calibration dates are included at the bottom of the plot by the open orange circles.".

P6 L15: replace "depending" with "dependant".

P6 L19: Include definite articles (the) in front of long-term and short-term target.

P6 L22: replace "are going" with "go".

P6 L23: To remove potential misunderstandings with the meaning of the word "control" in relation to station PI's reviewing data, replace "control" with "verify" or "review".

P6 L28: Please state which server you are referring to at the beginning of the line.

P6 L28: Where are the ATC data products located that station PIs check? Are they the plots shown on the ATC website (e.g. https://icos-atc.lsce.ipsl.fr/SAC)? If so, please state that these are publicly available for stations that have been labelled and give the link to the website.

*Yes, indeed, they are located on the website. We will add the link there.*

P6 L30: Please outline what is used as the flagging scheme. Is this already published in Hazan et al. (2016)?

*In Hazan et al. (2016), the automatic flagging scheme is described but not the user part of it. Raw data are controlled day by day. For valid data, we can choose additional information such as "Maintenance", Quality assurance operation" or " Non-background conditions" but this is not mandatory. Data have to be invalidated only for a objective reason which has to be chosen in a list to carry on the QC. The reasons can be (non exhaustive list): "Calibration Issue", "Flushing period", "Maintenance with contamination", "Inlet leakage"...   We will add this short description.*

P6 L31: As for P6 L23, change "controlled" to "verified" or "reviewed" to remove misunderstandings.

P7 L22-25: Establishing if your sample intake line has a leak or not is an important consideration. What is done at sites to establish if there are leaks if the mast cannot be climbed easily due to reasons outside of the control of the PIs and there is no infrastructure installed to alter intake configurations remotely?

*Up to now, all Pis of the labeled stations have managed to perform the intake line at least once. Also, to minimize leaks,  the specifications recommends to have sampling lines in one part, without connections.*

*One more point we are promoting is a check of the consistency of the vertical profiles of $CO_2$, $CH_4$ and CO during mid-afternoon periods in well mixed conditions (see Matthias Lindauer presentation during the 2020 ICOS conference, "Vertical gradients of greenhouse gases at 8 German atmospheric ICOS Stations").*

P8 L1-2: More information is needed for this test to be replicated by someone reading this paper and is not part of ICOS. It is not clear to me if a humidification loop is used, as in Stavert et al. (2019; https://doi.org/10.5194/amt-12-4495-2019) to conduct this test or is another method used within ICOS? In addition, what mixing ratios within the cylinder are suggested by ICOS for the test?

*We will rephrase for clarity. No humidification loop is used. We inject from a syringe directly in the inlet of the instrument in the case of the CRDS (they have an internal filter) and through a filter for the OA-ICOS. The cylinder used is with ambient air mixing ratios.*

P8 L9: Please provide a hyperlink for the automatically generated plots on the ATC website.

P8 L14: Please alter "bi-monthly" to "twice-monthly" or "every 15 days" to remove ambiguity.

P8 L30: Please include three letter identifier after Lindenberg to aid the reader in finding the example in Figure 4.

P9L8-9: Please include a space between numbers and units.

P9 L18: Calibration drift is misleading as a title, as it suggests the drift of mole fractions within the calibration cylinders. This may be the case for some species and over longer periods of time than 2 weeks; however, the authors are discussing the assessment of instrumental drift in this section. I would suggest changing the sub-heading to "Instrumental drift and calibration optimization".

*Thanks for this comment, we will do so.*

P9 L20: Is instrumental drift ever estimated over shorter timescales than 2 weeks? In my personal experience with optical instruments, some can drift quite significantly over periods of time less than 1 day. This is especially the case in environments where there is poor thermal stability (e.g. no air conditioning or too large thermal switches on air conditioning units).

*For some instruments, such as the ones measuring CO and $N_2O$, we definitively observe short-term drifts, in the scale of hours to days. In this case, we use a "short-term working standard" to correct for such drift. This standard is calibrated by the twice-monthly calibrations.*

P9 L30: Are there specific types of analysers that are worse than others for temperature dependence, e.g. instruments with larger cavities?

*We can expect that with a larger cell, there will be more inertia and hence a larger temperature dependence. However, within one type of instruments, we also observe some instruments with a significantly larger dependence than the others.*

P10 L9-10: Please state what the meteorological sensors need to be compliant against? ATC specifications document? Also, are the meteorological data submitted to the same database as the GHG data?

*The meteorological sensors need to comply with the list in the specification documents and indeed the meteorological data as well as the diagnostic data (flow rate, room temperature) are submitted to the same database.*

P11 L16: Replace "Belgium and France" with "Belgian and French".

P11 L28-30: This sentence isn't clear and I cannot decipher what the authors mean. Please rephrase to give clarity.

*We will rephrase for clarity. We wanted to say that some sites use two systems: the rotary valves to switch between the calibration and target gases and solenoid valves to choose between air at different levels and cylinder gases.*

P11 L32-33: Has a study ever been done by ICOS on the representativeness of using buffer volumes? As I understand it, one of the main advantages of using optical instruments is the added information that can be gained from the short-term variability in mole fractions. Would it be better to not use buffer volumes to futureproof the data at these sites for when numerical models can ingest high frequency data and simply smooth data based on statistical filters, as used at other sites and networks?

*We have studied the representativeness of using buffer volumes. The results have been presented at the Monitoring Station Assembly meetings.*

*The advantage of buffer is to have a better hourly representativeness while using discrete data for a multisampling height site.*

*As of now, only fours sites have chosen to use the buffer volumes.*

P13 L11-13: What were the sources of leaks for GAT and STE? It is not often that papers include lessons learned information, which is often very useful to other stations in diagnosing problems of their own.
***Unfortunately, we do not know the source of the longer stabilization time. At both sites, several interventions happened over time. This is also a lesson to learn on the ATC side, that we must make sure that the PIs are aware of the Step2 report recommendations and that they are followed up through.***

P13 L19: As American English has been used throughout the manuscript, please remove the th from after 15 (both occasions) as this is only included in British English.
P14 L30: As far as I am aware, there are two methods for a Nafion counter purge: using a dry gas, such as zero air or N2, or the Welp et al. (2013; 10.5194/amt-6-1217- 2013) method of reflux mode (i.e. taking a small portion of dried air post-Nafion and using it as the counter purge gas but ensuring it is at a lower partial pressure than the sample gas). Which method is currently recommended by ICOS when using a Nafion to dry samples?
***We recommend to use the reflux method described in Welp et al (2013) which avoid using consumable gas.***

P15 L4: Full name used for Jungfraujoch instead of a trigram. Please replace with JFJ to be consistent.
P15 L20: Grammatical error, please change to – "In this paper we have presented the process used to label ICOS atmospheric stations.".
P15 L28: Replace "oftentimes" with "often".

Table 1: Please include a reference to the ICOS atmospheric station specification document in the caption. In the table, it is not clear that the column "Gases, periodical" relates to flask measurements. Please add in more information in the column caption or in the table caption to clarify this. In addition, was atmospheric pressure at the highest inlet height ever discussed as a useful parameter for modellers to see the pressure differences between the top and bottom inlets of towers?
***No added value was found in having different measures of the atmospheric pressure. For modelers, we do measure the temperature gradients, useful to evaluate the inversion.***

Table 2: I am assuming that the thresholds cited in this table are specific to a certain type of instrument, i.e. CRDS seeing as these instruments are mostly used within the network for CO2 and CH4 analysis. In addition, please could you include the H2O threshold used for calibration gases, as referred to on P5 L25-26.
***The thresholds here are instrument specific and can change within the same type of instruments but indeed the thresholds written here work well for most of the CRDS instruments. For CO, they work as well for OA-ICOS, being even very conservative.***

Table 3: Please can you ensure that the writing of magl is consistent with figures (often written as m agl). This also applied to masl.
Table 4: The ambient air percentage for KRE looks erroneous (5069 %).

All figures: Please include lists of full site names, with acronyms in parentheses, of any site included in the figure. In addition, if any instrument number is included or a cylinder D number, please give an explanation. If species are mentioned, ensure that any numbers are subscripted, e.g. CO2, not CO2.
Figure 1 caption: Suggest changing to = " One month of target gas injections for CO2 (ppm), shown as the difference of calculated vs. assigned mixing ratios. Short term target data is plotted in green, whilst long term data is in brown. The calibration dates are shown by the light orange open circles.

Cylinder number (D******), mean values (± X), point-to-point variability (Ptp) and difference to the assigned value (Diff) are displayed above the figure."

Figure 2 caption: Notwithstanding that the figure is there to show locations to connect a cylinder to run the leak tests outlined in section 2.3.3., a key for the different parts shown in the figure would be useful for reader comprehension.

Figure 3 caption: Please indicate what the values in the 2nd and 3rd columns represent.

Figure 4 caption: Please indicate why there are duplications of LIN (as explained in the manuscript text).

Figure 6 caption: please indicate how the deviation of measurement from assigned values is calculated.

Figure 12 caption: The MHD Marine Smooth Curve isn't explained, please include a brief description of the methods used to derive the curve.

Figure 13: To make the insert clearer, consider adding a black border around to isolate it from the main figure. In addition, add in a list of the sites, with the acronyms in parentheses, as well as giving the years for the colours of when sites were first labelled.

Figure 15: It is currently hard to read the site acronyms and instrument numbers, please reduce the text size slightly so that there are some gaps between each site,like in Figure 16.

Figure 17: There is no bottom tine to the top plot.

Figure 20: What is the difference between ATC MetroLab HB 2016-05 and ATC Mlab Droplet? *HB stands for humidity bench while the Droplet is for the droplet test described in the text. The humidity bench allows a much more controlled and precise sensitivity test. We will detail the caption to give more clarity.*

---

## Author Comment (AC3) · 2 Oct 2020

**Author comment on "Evaluation and optimization of ICOS atmospheric station data as part of the labeling process" by Camille Yver-Kwok et al.**

**Anonymous Referee #3**

*We thank the reviewer for their positive review and their constructive and helpful comments. We answer them below, highlighted in bold and italic.*

This work describes the specific workflow and quality assurance processes within the ICOS network. The authors did a wonderful job composing a well written, compre hensive paper about the difficulties and challenges faced in the world of high-quality greenhouse gas observations. I think this work is a substantial contribution to the knowledge base and scientifically very important. Everyone in the field relies on intercomparable high-quality long-term observations but is rare to see papers that clearly describe and outline the hard work that lies behind them. I especially appreciate that the examples given in the paper include times where problems were found and solved, as well the realistic description of timeframes when setting up a new site. It is the lived reality in the field and will hopefully be educational to both users of the ICOS data as well as groups interested in setting up long-term observation stations. While the quality of the paper is generally high, there are some inconsistencies in the use of language mainly the interchangeable use of site names and abbreviations. Some of the parameters mentioned in the text and/or in figures have no clear explanation of how they were derived. The figures contain a lot of vocabulary and abbreviations that are probably useful within ICOS but can be confusing for the casual reader (tank and instrument numbers) without additional explanation. A lot of the figures also use very small fonts that can only be read by zooming and will not be legible in print outs at all. The colour in the figures is generally not suitable for colour-blind individuals (a lot of red and green with the same saturation levels). The descriptions of the figures in both the text and figure legends are very perfunctory even for some of the more complex figures

Specific Comments:
While the introduction gives a brief overview of greenhouse gas measurements and observation networks what the data is used for but lacks information about why we need such high-quality greenhouse gas observations and the benefits of the labelling process for the end-user of the data product.

*ICOS was first designed to serve as a backbone network to monitor fluxes away from main anthropogenic sources. There the concentration gradients between European sites is typically of only a few ppm on seasonal time scales. It is this signal that is used to make atmospheric inversion where from atmospheric gradients, using atmospheric transport models, one can deduce surface emission fluxes. To correctly capture this signal of a few ppms, a high precision and integrated network is needed. As a precise example, Ramonet et al. 2020, show that a strong drought in Europe like the one seen in summer 2018 produces an atmospheric signal of only 1 to 2 ppm.*

*The labeling process is very useful for new stations coming into the network to ensure proper setting; good measurement practice and in the end be able to reach the precision and stability requirement of ICOS. The whole process is lead by the ICOS ATC that offers expert support to the PIs of the stations. For the end-user, the labeling process is a guarantee of high quality measurement with proper metadata description and associated traceability along the data processing.*

In 2.4 general requirements and table 1, the different parameters mandatory and recommended for the different classes of station are given but not much information about why they were deemed mandatory or recommended. Two parameters in table 1, the mandatory atmospheric pressure observation at ground level and the recommended eddy covariance flux for CO2 are not mentioned in the text at all.

*We will add a few sentences about why these parameters were chosen.*

*ICOS atmospheric network aims to provide high precision measurements of greenhouse gases, and in priority $CO_2$ and $CH_4$ which represent the main anthropogenic GHG. In-situ measurements of $N_2O$, the third contributor to the additional radiative forcing, was not required in the initial phase of ICOS due to the difficulty to find at this time reliable instruments enable to provide the expected precision (Lebegue et al. 2016). This gas will be added in the new phase of ICOS. Flask sampling is required at class 1 stations for quality control of in-situ measurements, and to provide additional trace gases measurements like $N_2O$, $H_2$, $CO_2$ isotopes (Levin et al., 2020). Other parameters are required in order to support the interpretation of the GHG variabilities, like CO as a tracer of combustions, and meteorological parameters to characterize the local winds, vertical stability along tall towers and weather conditions (P, T, RH). The eddy covariance flux have been selected in this list with the idea to characterize the local surface fluxes either from biogenic and/or anthropogenic activities and to monitor possible long term changes around the ICOS sites. So far this parameter is not required for the labeling process due to logistical difficulties to install such measurements at several atmospheric sites.*

In 2.2 greenhouse gas calibration requirement: Add an example or range for the automatically filtered data.

*We will add an example of such range. For example, for the cavity pressure, the range for the CRDS instruments is usually 139..8 to 140.2 Torr. These ranges are also given in Hazan et al (2016). Of course, those parameters are depending on the type of analyzer.*

2.3.3 Intake line and water vapour correction tests: The text is a bit unclear, it mentions a shelter test every 6 months and testing the outside lines every year, then later in the text it mentions testing of the whole line is recommended at sites where lines are older than 10 years. Is the yearly testing of the outside line just to tests from the base of the tower to the shelter or is the yearly testing of the whole outside line only recommended for lines older than 10 years? I assume that since ICOS focusses on tall tower sites, yearly full line tests would be very expensive (hiring climbers to climb the tower twice for each test to connect and disconnect the lines).

*We will clarify this section. Every 6 months, there is a shelter test and then every year the intake line test. This test can be either by injecting a target gas from the top of the lines (that we recommend for older lines only as it it complicated to implement) or by maintaining a depression in the line. For almost all towers, PIs can climb themselves so it is not a problem.*

2.4.4 Calibration drift and optimization: In this section and in figure 6 the word calibration drift is used for the instrument drift that is then corrected and optimized with the calibration, this is very confusing as there is also such a thing as a drift in the concentration of calibration gases.

*We will clarify the text and reformulate to talk about instrument drift instead.*

2.4.7 Diagnostic parameters, Page 10, line 7: A low flow rate within the line could also be indicative of an obstruction in the line (damage to sampling line or blockage).

*Thanks for that addition included in the manuscript.*

3. Presentation of the 23 labelled stations Page 11 line33: The paper generally describes all the site setups in great detail, but there is no description of the 4 sites with buffer volumes. I appreciate that

everyone in the community has strong opinions about the usage of buffer volumes, but regardless of their merits or lack thereof, a more detailed description of the buffer volume setups (integration volume, flow rate and in integration time) would be helpful.

***We will add some details about the buffer volumes. At the Swedish sites, buffer volumes of 8L are used with an integration time from 3.8min to 4.9min and a flushing rate between 1.6 to 2.1M/min. At SMR, buffer volumes of 5.6L are flushed at 0.325 L/min which gives an integration time of 16.9min.***

4.1 Calibrations: Is the drift described this chapter and in table 5 the same instrument response drift described in 2.4.4 and figure 6? The naming is ambiguous as it implies it is the calibration that is drifting not the instrument response. There is also no description of how the drift was calculated.

***Yes, we are still talking about instrument drift. As above, we will clarify and explain how we calculated the drift. When the drift looked linear enough, we used the difference between the first and last data points and divided by the number of months.***

4.3 Uncertainties: Figures 16-18 contain a lot of information, while some of it is explained here in detail, other artefacts are left unmentioned or are mentioned later in the text and then not referenced. For example in Figure 16: For JFJ, The continuous instrument repeatability (CMR) is good but the long-term repeatability (LTR) is high, later in paragraph 4.6 it is explained that this is was due to a polytube Nafion, but the text does not reference figure 16. Then there is OPE that seems to have had a bias in the target value for both CH4 and CO2 for a while. On page13, line 21 it is mentioned that two sites show values outside of the WMO targets for CO, but it is not specified which of the subplots this refers to. 3 additional sites show larger biases after the initial test period (IPR, SMR and NOR) but they are not mentioned in this section. Later in the troubleshooting section, there is a mention of the issue with CO that was related with the use of heated inlet cups but figure 18 is not mentioned in that section and the sites are referred to with their full name.

***We will comment in more details these figures and reference them in a better way in the text. In the case of OPE, it is most probably due to the manual QC that has not been done in time. It may disappear as all data have been carefully quality control for the recent ICOS release in September 2020. For CO, on Figure 18, bias to the assigned values, the black lines shows the WMO compatibility goals. In the text, we talk about GAT and HTM (next sentences). We see that HTM, IPR, NOR and SMR are slightly below the threshold over the last year while during the test period, GAT was exceeding the goals and HTM showed very noisy data. The data from the intake cups were QC invalid, moreover they were not concerning the target data that are used to produce Figure 18 only the air data.***

Figure 11: This figure is not easy to read and within the text, it is just casually referred too, does it add any value? How does it help evaluate the influence of different sources ( is it because it shows the different inlet heights?) whatever information it is supposed to convey is lost in the sheer amount of data (1 year 4 heights 3 compounds, plus quality assurance subplots). I could see the value of a plot like this online where you can zoom in to it. The short-term long-term target stability on the right-hand side is interesting but is not even mentioned in the text or legend of the figure and the short and long-term targets are also shown in figure 1.

***We will comment this figure in more details. This figure is available on the ATC web site in daily, 10 days, monthly and yearly versions (updated every day). Indeed, on the yearly figure, we mostly look for patterns in the targets, data gaps, outliers, whereas the ambient air signals are much more visible on shorter period's figures. ...***

*We will clarify or correct as proposed in the technical comments. If needed, we add a more detailed answer to the comments.*

Technical comments:

Page 3, line 11: First mention of WMO compatibility goals but no information what they are for the gases discussed in the paper.

Page 5, line 1: No reference for the ICOS specification document.

Page 5, line 25 mentions that Table 2. Contains the raw minute and cycle but then Table 2 contains the minute, injection, and cycle data. Is an injection not the same as a cycle? The words are interchangeably used throughout the paper for example in figure 7.

*We will clarify: an injection is one sampling of 30 minutes, a cycle is the suite of the injection of each calibration cylinders (so 4 injections of different mixing ratios), the calibration is then a suite of 3 to 4 cycles.*

Page 7 line 22: What is meant with the intrinsic bias of the instrument?

*What is called "intrinsic bias" in the Mlab initial test report is the bias observed in the final test when the tested instrument with dry air is compared to reference instrument with dry air. We will simplify in the manuscript.*

Page 7, line 31: Rephrase to clarify that the onsite water test needs to be performed if the last instrument test at MLab was longer than a year ago.

Page 10, line 20-21: Clarify that the instrument flow rate can be used to estimate the lifetime of cylinders.

Page 11, line 1: LTR, is defined as the Long-Term Repeatability which is I understand to be the 3 day average of the standard deviation of the short-term target measurement.

The text does not specify that it is based on the short-term target although it must be as the long-term target would not be measured often enough for this. The naming is confusing as the short-term target makes up the long-term repeatability but then there is also a long-term target.

*Sorry about the confusion, we will clarify in the text but cannot change the term. The LTR is in opposition with the short-term repeatability calculated over 10 injections within a few hours (see Yver et al, 2014).*

Page 13, line 15: Figure 12, not 13 describes the bias calculation, the text would be easier to follow it the calculation was described within the text of the chapter and or the relevant figure description.

*We will correct and add some details.*

Page 13, line 18 change to: The red dot.

Page 13, line 29-Page 14, line 4: Restructure this part to make it easier to read. The text references that the description of figure 19 contains how the bias was calculated but figure 19 contains no calculations. Later in the paragraph in page 14, line 4 it is clarified that the bias is (measured-reference) I assume what is described as the bias is just the difference between the concentration measured in the line vs. directly in the tank but the roundabout description makes it harder to understand than necessary.

Maybe just rephrase the second sentence to make it clear it is measured – reference as currently, it is the other way around.

Page 14, line 13: Reference water droplet test protocol used.

*This test is shortly described in 2.3.3. The protocol used is not available publicly.*

Page 14, line 30: If available add part numbers and manufacturer for both the recommended Nafion membrane and cryogenic water trap.

*The recommended Nafion membrane is the model MD from Perma Pure. For the cryogenic water trap, no model is recommended except one that goes at least at -50°C.*

Table 1: is missing a reference to the ICOS specification document.
Table 4: 5069% in KRE?.
***We will correct this typo, the right value is 69%.***

Figure 2: No labelling and description of parts and existing text font size too small.
Figure 11: The colour choice is not ideal (yellow on white background, and red and green lines and circles for both the target values). The numbers above the target plots are not explained anywhere, one I assume is the targets assigned concentration but Ptp is not explained anywhere. For the H2O % the target numbers are also present but empty (0.00 for all) please remove.
***This figure is produced automatically (once a week) and stand as an example of what is provided in the reports. It was taken from the report for the site Torfhaus (TOH). We will add comments about the different numbers.***

Figure 15-19: It is hard to visually align the legend at the bottom with the two figures above, especially in figure 16-18 where there are 2 box and whisker plots for each instrument. Adding the legend to the other 2 plots or some kind of shading or line might help.
Figure 16: Has some random signs at below the figure legend # # \.
***This is due to writing the paper in R+Latex, hopefully it will disappear once processed by the editor.***

---

## Author Comment (AC4) · 12 Oct 2020

We would like to correct our answer for one of the comments:

P13 L11-13: What were the sources of leaks for GAT and STE? It is not often that papers include lessons learned information, which is often very useful to other stations in diagnosing problems of their own.

There seems to be a misunderstanding in regards to potential leakages at GAT and STE stations. During the period with longer stabilisation times, leakages were neither confirmed nor disproven due to missing leak tests. Leakages affecting the stabilisation

time have to be located between the calibration cylinder, the rotary valve and the instrument. Therefore ambient air measurements are not affected. However, the potential leak would have been minor resulting in valid calibrations. This was independently confirmed by the agreement of the target tank value within the WMO compatibility goal for CO2 and CH4. The implementation of the first ICOS stations and their labelling process showed that the communication between ATC and stations PIs needs to be improved. Therefore, a better feedback mechanism will be established to track the fulfilment of the ATC's recommendations by the stations.
* * *

---

## Author Comment (AC5) · 12 Oct 2020

We would like to correct the answer to two comments that were switched around:

What are the criteria that go into validating the data at hourly timescales as shown in Figure 3? For example, is there a % or minimum data amount requirement to flagging the hourly data as valid/invalid?

The hourly means are computed automatically using minutes means which are themselves computed using raw data. If there is at least one valid raw data within a given minute, the corresponding minute mean is considered as valid. Similarly, if there is at

least one valid minute mean to compute an hourly mean, the hourly mean is consider valid. There is no automatic quality control criterion applied to the hourly means, only the criteria are applied on the raw data.

[P19L7-12] Meteorological measurements - Can the authors specify what checks the ATC performs on the meteorological measurements? My understanding is that verifying the instrumentation and checking the accuracy of the meteorological data is quite challenging, and it would be interesting to know the ATC's specific procedures.

The ATC performs simple filtering on the raw data based on valid ranges (min/max values) for the five mandatory species which are pressure, temperature, relative humidity, wind speed and wind direction. Except for relative humidity, the data are also marked as invalid if the measurement is constant for more than X minutes in a row. X is set to ten for the wind variables and to 60 for the other species. The second criterion is used to cope with blocked sensors. We are working on a comparison with the ECMWF data to highlight potential drifts or outliers. In terms of instrumentation, we are working on instating a two-year recalibration for the humidity sensors that are the one drifting the fastest over time.

---

## Author Comment (AC2)

**Author comment on "Evaluation and optimization of ICOS atmospheric station data as part of the labeling process" by Camille Yver-Kwok et al.**

**Anonymous Referee #2**

***We thank the reviewer for their positive review and their constructive and helpful comments. We answer them below, highlighted in bold and italic.***

Accurate measurements of greenhouse gas (GHG) concentrations in the atmosphere is the first step to understanding and mitigating the impacts of climate change, and a full assessment of the temporal and spatial variabilities and trends in these concentrations requires a wide network of observations. The GHG measurement community has worked hard to ensure that the GHG measurements operated around the world are "compatible" (data meet quality standards necessary for addressing the scientific questions and can be compared under common calibration scales) through the efforts such as the recommendations of the WMO/IAEA GGMT meetings (most recent report at; https://library.wmo.int/index.php?lvl=notice_display&id=21758#.Xzxuv1QzaUk). But more focused efforts such as the Integrated Carbon Observation System (ICOS) present an opportunity to improve on these practices and derive a more coherent dataset required for the regional focus of such projects. As such, this current work, documenting the key aspects of the ICOS station labeling process and data quality control, is a significant contribution to the field, and suitable for publication in this journal.

Overall, the manuscript is well-written and major revisions aren't necessary. However, I would like to make some suggestions that, when properly addressed, I feel the community will benefit in knowing, especially given the extensive and high-quality dataset that this work can derive it's conclusions from.

[P5L13-29] Cylinder calibration requirements
I feel that the authors can further elaborate on the following questions.
- Line 14: Can the authors specify what range of calibration frequencies are typically tested in the initial test period?
***During the initial test period, a minimal frequency of 15 days is applied. Depending on the stability of one calibration to the other, we can determine if the frequency can be reduced but at the present time we require at least one calibration sequence per month. In addition we recommend to perform a calibration also after any power cut. In such a case the protocol is to let the instrument measuring ambient air for several hour after the restart of the instrument, before starting the calibration sequence. According to the stability of the analyzers for $CO_2$ and $CH_4$ we could consider relaxing the frequency. However, the lifetime of the calibration scales at stations is already long (at least 10 years) so the cost of the calibrations is not very high. Also the CO measurements performed by the same analyzers (CRDS picarro model G2401) or by other analyzers (OA-ICOS or CRDS for $N_2O$ and CO) are generally less stable in time. In such cases we add to the calibration protocol one tank (called REF or short-term working standard) measured every few hours in order to correct for short term drifts often observed for CO and $N_2O$ measurements.***

- Are there any requirements for the order in which the 3-4 cylinders are measured in a sequence?
***No order is required but in most sites, the calibration goes from lower to higher mixing ratio.***

- Line 18: I would prefer that the authors clarify that the 30 minutes is the run time per cylinder, even if that becomes implied when reading through other parts of the text.
Similarly, I think it's worth further clarifying that the calibration cylinders will be run in series, leading to an estimated length of some continuous hours of calibration per sequence.

- When stating the total length of the calibration sequence, I feel it would be more coherent if a short reference is made to the discussions in 2.3.1 regarding the how the instrument will dry out during the long calibration sequence.
*We will clarify these points.*

- Figure 2 indicates a REF tank which isn't explained in the text. Is this part of the ICOS requirements, and if so, can an explanation be provided for the role of the REF tank?
*We will add an explanation. A REF or short-term working standard is strongly recommended for instruments showing a short-term drift in the order of hour/day such as the instruments measuring $N_2O/CO$. It is measured as often or more than the target and is used to correct the short-term drift (see previous point on calibration frequency).*

- Figure 2 suggests that all the samples go through the selection valve and straight into the instrument, which means that the inlet pressure will likely vary between samples, for example the air inlets will likely be slightly sub-ambient, while the calibration cylinders will be dependent on the regulator settings for each tank, and this design is dependent on the instrument's internal cell pressure control to compensate accordingly. Is this correct? Is there any guidance on the regulator pressure settings?
*It is correct. We advise to set the pressure on the regulators so that the pressure and flow are slightly higher than the air measurements. For example, on CRDS instrument, this is done by comparing the outletvalve value. During the initial test of the instrument, an acceptable range for the pressure, and outletvalve is determined to help the PI set the regulators. During the initial tests at the MLab, biases are mostly seen when changing the pressure then over time at the same pressure they decrease, this means if the difference between air and cylinder is too large then it will take too long to disappear over the time we are measuring the samples. This useful comment will be added in the manuscript.*

- Regarding regulators, I feel the community would greatly benefit from a list of recommended regulator models, similarly to how the the authors have stated a specific recommendation for the sample switching rotary valve later in the text.
*The recommendation is listed in the ICOS specification document (SCOTT MODEL 14 M-14C (or -14B) Nickel-plated brass or TESCOM Serie 64-3400 Stainless steel electropolish with PCTFE valve seat and the use of Stainless steel High purity gas pressure gauge (e.g. Bourdon Haenni UPG2)). We may add this information in the paper, at least mentioning that it is provided in the ICOS specifications which is a public document (see the point about this reference below).*

[P6L9-20] Moisture effects on the short-term target measurements
- The effect of moisture on the short-term target is very interesting. How long does it take for the moisture level to return to steady state once the air measurements resume, and is there an attempt to flag out the air measurements while the moisture levels are still low?
*The first minutes of each sample are flagged out, the length is determined individually. Usually, without a Nafion drier (which was the case in this study for all CRDS at the time of the labeling), the humidity levels are back to steady state almost instantaneously. With a Nafion drier, it can take up to 2 hours but the humidity influence is corrected.*

- Also, what are the implications of calibrating the instrument in a dry state while air measurements are done wet, especially in cases where a dryer is not used? Does what's shown for the short-term target measurements indicate there could there be a (small) bias between the calibration and air measurements?
*Indeed, for this type of instrument, measuring in super dry conditions leads to a bias. This is why we ask the PIs to measure the target tank just after the calibration to be able to estimate this bias. Depending on the instrument, this bias is more or less pronounced but do not exceed 0.05 ppm*

*for $CO_2$ and 0.4ppb for $CH_4$. It is also part of the uncertainty of the water vapor correction estimated during the initial test at the Mlab. Moreover, one of the final test is to compare the instrument with the Mlab reference instrument whose samples (air and cylinder gas) are all dried.*

[P7L15-27] Inlet line tests
- Just making sure I understand this process correctly: For the instrument test, do I understand correctly that there's an electronically actuated 12-port switching valve, and that one of the ports (#12 in this case) is being left open to facilitate direct measurement of the test tank? Is this port normally plugged?
*Usually, we use 16 port valve so even with 7 cylinders and up to 5 levels, there are some ports that are free. We can use any free port, it does not need to be the same all the time. They are usually closed when not used.*

- For the shelter test, I presume the downstream overflow pump for the inlet line is kept on, then the pressure of the test tank connected to the inlet line is adjusted to the point that the slight sub-ambient pressures in the inlet lines are reproduced? If these presumptions are correct, I would advise that these details be included in the manuscript.
*We turn the flushing pump off and adjust the regulator to reproduce the ambient pressures. We will add theses details in the text.*

[P8L12-24] PI data validation
- Can the authors clarify the temporal resolution in which the PI's perform data validation? I presume either raw or 1-min level?
*The PIs validate the data on the raw level first then every one to two months on the hourly level. The second validation on the hourly dataset aims to verify the longer term consistency of the time series. Every time a data flagging is performed either on the raw or hourly data, a reprocessing is automatically applied to the other aggregated levels (raw, minute, hour) for consistency.*

- What are the criteria that go into validating the data at hourly timescales as shown in Figure 3? For example, is there a % or minimum data amount requirement to flagging the hourly data as valid/invalid?
*The ATC performs simple filtering on the raw data based on valid ranges (min/max values) for the five mandatory species which are pressure, temperature, relative humidity, wind speed and wind direction. Except for relative humidity, the data are also marked as invalid if the measurement is constant for more than X minutes in a row. X is set to ten for the wind variables and to 60 for the other species. The second criterion is used to cope with blocked sensors.*

[P9L19-28] Calibration drift due to regulator effects
- I find Figure 5 fascinating, in terms of understanding regulator contamination effects. As expected, the short-term target, which is measured more frequently, shows faster stabilization time. On the other hand, for the long-term target and calibration tanks, all measured at the same 15-day cycles (as far as I understand), regulator flushing problems are much more severe in the the long-term target. My guess is that the calibration runs are an average of the 3~4 cycles, hence the variability is substantially reduced by the 2nd~4th runs. Is this correct? If so, comparing the first calibration runs against the long-term target run may provide a more meaningful comparison of regulator contamination effects. Also, it would be interesting to know if stability is reached much more quickly in the 2nd~4th calibration runs.
*It is correct, however we have focused in this plot in providing a stabilization time for each sample to potentially save gas without losing quality. This is something we could look at in the future.*

- Do all the cylinders use the same model of regulators? If not, could that be a consideration in interpreting the results shown in Figure 5?- I might expect that each of the calibration tanks would show a somewhat different contamination effect based on its concentration, but have the authors looked as this?

*All cylinders should use the regulators deemed compliant in the specification document. Only two types were selected: SCOTT MODEL 14 M-14C (or -14B) Nickel-plated brass or TESCOM Serie 64-3400 Stainless steel electropolish with PCTFE valve seat and the use of Stainless steel High purity gas pressure gauge (e.g. Bourdon Haenni UPG2) after study presented during MSA meetings.*

- I find it interesting that each compound shows a different pattern in reaching steady-state. Looking at the long-term target where the signals are most amplified, $CO_2$ is initially very low, $CH_4$ starts low and slightly overshoots prior to reaching steady state, while CO is initially very high. Do the authors have an explanation for these results? Some of these discussions may be more suitable in section 4.2.

*We do not have an explanation for this but this at least is partly instrument specific.*

[P9L19-28] Calibration drift
- Have the authors estimated whether the instrument drift is "linear"? I.e. When looking at the drift rates of the 4 calibration cylinders at different concentrations, is there indication that the drift rate is similar/different at each of the concentrations? Looking at Figure 6, it certainly seems like the 4 tanks drift at a similar rate, but it's hard to be quantitive. I would love to see an assessment of instrument linearity changes quantified and compared across instruments, and the vast dataset accumulated in this ICOS experiment would be an excellent for this.

*From the linearity tests performed at the Mlab and the figures like Figure 6 made for each site, it indeed seems that the drift is linear. However the drift can change every time the instrument is restarted which makes the analysis challenging. We think it will be interesting to perform such analysis when all the sites will have a longer period to investigate (some sites only had a few months of data).*

[P19L7-12] Meteorological measurements
- Can the authors specify what checks the ATC performs on the meteorological measurements? My understanding is that verifying the instrumentation and checking the accuracy of the meteorological data is quite challenging, and it would be interesting to know the ATC's specific procedures.

*The hourly means are computed automatically using minutes means which are themselves computed using raw data. If there is at least one valid raw data within a given minute, the corresponding minute mean is considered as valid. Similarly, if there is at least one valid minute mean to compute an hourly mean, the hourly mean is consider valid.*
*There is no automatic quality control criterion applied to the hourly means, only the criteria are applied on the raw data.*
*We are working on a comparison with the ECMWF data to highlight potential drifts or outliers. In terms of instrumentation, we are working on instating a two-year recalibration for the humidity sensors that are the one drifting the fastest over time.*

[P10L22] Alignment of timestamps
- One thing that hasn't been mentioned here is whether there are procedures to ensure that the timestamps of the various data streams on site are matched. For example, when combining the data stream from the CRDS and the meteorology measurements, is there a process to ensure that the timestamps for these two data streams are matched? Or similarly in case there are multiple CRDS instruments on site?

*All instruments have access to internet and a time server that is updated at least once a day. This information will be added to the manuscript.*

[P12L29] Section on tank stabilization time
- Again, can the authors confirm that the stabilization time for the calibration combines the data for all 3~4 cycles? I would imagine that the 1st cycle shows the longest stabilization time, while the others would be relatively short, and averaging these together would lead to a relatively short stabilization time. In fact, I wonder if the 1-min stabilization time mentioned in page 13 line1 refers to this special case of 2nd~4th cycles in a calibration run?
*Indeed, we combine the data from all cycles, including the first one. As the first one is not used to calculate the calibration coefficients, its weight on the stabilization time is less important.*

- Please refer to questions above on "Calibration drift due to regulator effects" on some questions I hope the authors can address, perhaps in this section.
- It is interesting that looking at $CH_4$ in Figure 15, the short-term target stabilization times are higher than the long-term target stabilization times at many sites. I'm a little surprised by this since the short-term target regulator would be flushed more frequently and hence I would expect it to show less regulator artifacts. Do the authors have an explanation for this?
*We do not have a definitive explanation for this. However, in most cases, the difference is small. As both target flushing time is the same in the database, we choose the largest value to be conservative.*

Minor comments (Denoted by Page# and Line#)
P2L10-19 Introductory paragraph: This introductory paragraph is a summary of the long history of greenhouse gas measurements, which understandably is not an easy task.
Here are some thoughts:
- "Continuous"?: First, it is somewhat vague what "continuous" refers to in this case. I presume the authors mean measurements of very high frequency, but technically speaking, no measurement is truly "continuous". While I understand the authors' emphasis on measurements made at high-frequency, I do feel that the significant scientific contributions from flask based measurements of GHGs deserves to be acknowledged. Also, I would note that the AGAGE measurements (referenced in the Prinn et al.) are GC based and I'm not sure that one would describe them as "continuous" (although I believe some prior publications have described the measurements as "quasi-continuous").
- Also, I find it odd that the more recent megacities efforts are not referenced here, nor are any of the flux measurement networks, as both are high-frequency measurements, and the authors note the importance of networks to address regional and local fluxes (line 17).
- Given these comments, I would ask that the authors reconsider the framing of this introductory sentence, and rewrite as necessary.
*We will rephrase to acknowledge the different frequencies and type of sampling as well as different networks.*

- Line 13, Prinn et al. 2000: Please update to Prinn et al 2018. Prinn, R., Weiss, R., Arduini, J., Arnold, T., DeWitt, H., Fraser, P., Ganesan, A., Gasore, J., Harth, C., Hermansen, O., Kim, J., Krummel, P., Li, S., Loh, Z., Lunder, C., Maione, M., Manning, A., Miller, B., Mitrevski, B., Mühle, J., O'doherty, S., Park, S., Reimann, S., Rigby, M., Saito, T., Salameh, P., Schmidt, R., Simmonds, P., Steele, L., Vollmer, M., Wang, R., Yao, B., Yokouchi, Y., Young, D., Zhou, L. (2018). History of chemically and radiatively important atmospheric gases from the Advanced Global AtmosphericGases Experiment (AGAGE) Earth System Science Data 10(2), 985 - 1018. https://dx.doi.org/10.5194/essd-10-985-2018
P3L8: Does the "data center" have any special acronym like MLab or MobileLab? At first reading, the data center didn't seem distinguished enough as the second component of ATC.
*The data center is definitively as distinguished as the MLab but it does not have a special acronym. The need for such acronym never seemed to come.*

P3L11 "ATC and Laurent (2017)": The notation of this reference seems a bit strange to me. Looking at the original document, I see that it is edited by Laurent with many contributors listed. This metadata page (https://meta.icos-cp.eu/objects/_fDB4nDzrYYG9uu6fPsvfiG9) asks that the citation be for "Laurent, O., ICOSAtmosphere Monitoring Station Assembly and ICOS Atmosphere Thematic Centre (ATC)". My suggestion would be for "Laurent et al." or "ATC" with Laurent noted as aneditor in the reference, but I ask the authors to confirm the appropriate way to reference this report. Also note that the full reference for this item seems to include some typographical errors.
***A DOI has recently been attributed to this document and the reference will be updated in the manuscript:***
***ICOS RI (2020): ICOS Atmosphere Station Specifications V2.0 (editor: O. Laurent). ICOS ERIC. [https://doi.org/10.18160/GK28-2188](https://doi.org/10.18160/GK28-2188)***

***For the technical corrections, we will apply them.***
P3L13 "Downstream": This usage seems a little awkward, as I would associate the term with a physical connected process (as in instrumentation), and not for a separate task carried out at a later point in time. Perhaps something like "Afterwards" or "Once the labeling process has been approved,"?
P3L30-33: Should the first word after the ":"s be capitalized?
P4L4 "exchanges first with" -> "first contacts"?
P4L6 "in a limited number" -> "a limited number of"?
P4L10 "sites" -> "site"?
P4L15 "thanks to" -> "through"?
P4L33 "class 1" -> "Class 1"?
P5L14 "a lot of" -> "more frequent"?
P6L15 "depending" -> "dependent"?
P6L15 "really important" -> "important"?
P6L18-20: "If the instrument variability should be assessed with...": I'm not exactly sure what the authors refer to in this sentence, please clarify.
P6L22 "are going through" -> "go through"?
P6L22 "different criteria" -> "various criteria"?
P7L1-5: Capitalize each "to"?
P8L6 "are meeting" -> "meet"?
P8L27 "evaluating that" -> "evaluating whether"
P10L16 "flow rates": Can the authors clarify what specific flow rate is being referenced here, and where this flow rate measured?
P11L20 "who" -> "which"?P11L28-30 "The other ten use either solenoid values either for..." : The meaning of this
sentence seems unclear, with the two "either"s. Please revise.
P12L2 "experience often" -> "often experience"?
P12L2 "do no use" -> "do not use"
P13L9 "was concerned" -> "was of concern"
P13L9 "as said in" -> "as discussed in"
P13L9-10 "GAT CRDS lines showed a long stabilization time" -> "Long stabilization times were found for the GAT CRDS lines"?
P14L21-22 "This can be due to ... analyzer inlet,..." -> "This can be due to factors such as ... and the filter at the analyzer inlet."
P14L30 "either a cryogenic" -> "or a cryogenic"?
P15L11: Use of acronyms vs full names for the sites?
P15L16 "no CO bias was observed anymore" -> "CO bias was no longer observed"?
P15L25: Should the first letter of each bullet item be capitalized?

Table 1. Based on P4L27, is the "*" in [Class 1 ; Gases, periodical] applied to the wrong item? I.e., shouldn't the 14 C item be starred?

Table 3. Coordinates: I would find it helpful if the lat/lon coordinates could be given with more significant digits to allow for pin-pointing site locations in interactive mapping tools such as Google Earth.

***We used the values provided by the PIs during the Step1 on the CarbonPortal, we will check them for their accuracy.***

Table 4. KRE Ambient air (%) is 5069?

***This is of course a typo, it will be corrected.***

All figures: Some figures missing specific description of how the error bars were calculated. Also, please note the differing concentration units in the plots/text (ex. Ppm vs µmol/mol), and any missing subscripts in CO 2 and CH 4 .

Figure 2. Perhaps arrows would be more apt in identifying the specific injection points? Also, the plot is difficult to interpret without legends for the symbols.

Figure 15. Can the font size of the x-axis labels be adjusted for better legibility?

Figure 16-18: It seems like the bottom Bias plot in each figure has 26 ticks on the x-axis, while the CMR/LTR plots above have 27 ticks. Is there a station/instrument missing in the bottom Bias plot? Also, the pink/blue boxes are very hard to distinguish for many of the box plots, is there a way to improve legibility?

***We will improve the figures according to comments. In the case of the box plots, their size is due to the small range (which show the good quality of data). We can try to show some outliers differently to reduce the overall range.***

---

## Editor Decision (ED1)

Many thanks for working on the thorough revisions to this manuscript based on the reviewers' comments. This is clearly a substantial piece of work and will serve as a highly beneficial resource for the atmospheric monitoring community.

I have some minor additional comments that the authors might want to consider in preparing the final manuscript for publication.

- Prinn et al. 2000 should be updated to Prinn et al., 2018 (https://doi.org/10.5194/essd-10-985-2018)
- Page 2 line 25: 'surface emission fluxes' to 'surface fluxes'
- Page 3 line 20: suggest 'For the end-user, the labeling process guarantees high quality observations with full metadata description and traceable data processing.'
- The labelling document at the bottom of page 3 has a poor link. Please can you state exactly the title
  of the document, rather than just 'labelling document' this will help find the document should that
  link be lost at some point. A better link would be fantastic if possible, or a higher tier website that is
  more likely to maintain this document at some point in the future. This might also be the case
  elsewhere e.g. bottom of page 5
- Page 5 line 9: '14C values', should this be ' $\Delta^{14}$ CO2 values'?
- Page 5 line 26: rather, 'due to poor performance'
- 'Indeed, the mole fraction assigned by the CAL-FCL are given in extremely dry conditions as well as the target measured directly at the end of the calibration sequence in the field.' I think the following might be better if I understand correctly: 'Indeed, the mole fraction assigned by the CAL-FCL, as well as the target measured directly at the end of the calibration sequence in the field, are given in extremely dry conditions.'
- Page 8 line 27. Is there any reason 10 years is recommended? At least some reason to recommend this would be good
- Page 8 line 32: 'If the instrument has NOT..'?
- Page 9 line 27: Worth listing all the issue if not too long.
- Page 10 line 19: I couldn't quite make sense of '(15 times four cylinders and all cycles taken into account, here four).'
- Page 11: Spell our ECMWF and state model data comparison.
- Page 16/17. There are a lot of different dimensions/models of Nafion is there any more detail as to what the 'polytube' Nafion is?
- Page 17 line 4: what needs to be 'well controlled'?
- Figure 4 can you just mention what reason could be for white space (e.g. instrument doesn't measure something, or down for maintenance)?
- Figure captions 16, 17 and 18 refer to Figure 13. Is this correct? I don't think it makes sense. Please double check that all Table and Figure references are correct throughout.

Fantastic job. Thanks to the ICOS team for their continuing hard work!